# Massively parallel reporter assay for mapping gene-specific regulatory regions at single-nucleotide resolution

**Alastair J Tulloch[1,2†], Ryan Nicholas Delgado[1,2], Rinaldo Catta-Preta[1,2‡, §], Constance L Cepko[1,2,3]\***

[1]Department of Genetics, Harvard Medical School, Boston, United States; [2]Howard Hughes Medical Institute, Harvard Medical School, Chevy Chase, United States; [3]Department of Ophthalmology, Harvard Medical School, Boston, United States

**\*For correspondence:** cepko@genetics.med.harvard.edu

**Present address:** [†]Neurobiology, Neurodegeneration and Repair Laboratory, National Eye Institute, National Institutes of Health, Bethesda, Maryland, United States; [‡]Ocular Genomics Institute, Mass Eye and Ear, Boston, United States; [§]Department of Ophthalmology, Harvard Medical School, Boston, United States

**Competing interest:** The authors declare that no competing interests exist.

## eLife Assessment

This manuscript presents a **valuable** methodological approach to investigating context-dependent activity of cis-regulatory activity within defined genomic loci. The authors combine a locus-specific massively parallel reporter assay, enabling unbiased and high-coverage profiling of enhancer activity across large genomic regions, with a degenerate reporter assay to identify nucleotides critical for enhancer function. The data supporting the conclusions are **solid**, highlighted by successful identification and characterization of both previously known and new regulatory elements across multiple developmental stages, cell types, and species. While the approach has inherent limitations in sensitivity, and indirect assignment of regulatory elements to target genes, it provides a flexible platform for nominating candidate cis-regulatory elements across defined loci.

**Abstract** Precise gene regulation is essential for tissue development and function, yet mapping cis-regulatory modules (CRMs) at high resolution and in specific cell types remains challenging. We introduce two complementary strategies—a locus-specific massively parallel reporter assay (LS-MPRA) and a degenerate MPRA (d-MPRA)—designed to overcome limitations in throughput, resolution, and prior knowledge requirements. LS-MPRA uses BAC-based libraries to densely sample genomic regions, enabling unbiased interrogation of millions of DNA fragments for CRM activity. D-MPRA applies systematic mutagenesis to resolve CRM architecture at single-nucleotide resolution, nominating essential bases that may function as TF binding sites or other regulatory elements. We applied these methods to retinal genes expressed in mature rods and bipolar interneurons using in vivo and ex vivo mouse (*Mus musculus*) tissue. LS-MPRA recapitulated known CRMs and identified previously uncharacterized CRMs, including those embedded in neighboring genes. Applied to *Olig2*, a dynamically expressed gene in retinal progenitors, LS-MPRA identified three CRM regions, which d-MPRA and motif analyses further dissected. CUT&RUN confirmed direct binding of candidate TFs. Extending LS-MPRA to chick (*Gallus gallus*) retina and spinal cord demonstrated cross-species and cross-tissue applicability. Together, these approaches provide a rapid, scalable, inexpensive, and accessible platform for CRM discovery that can be carried out without prior element annotation and with tunable (small) fragment sizes.

## Introduction

The precise regulation of gene expression is fundamental to the development and function of complex tissues. Intricate transcriptional programs govern cell fate decisions and ensure the proper spatial and

temporal expression of genes. Central to these programs are *cis*-regulatory modules (CRMs), which integrate combinatorial inputs from transcription factors (TFs) and other regulatory mechanisms to orchestrate gene expression in a context-dependent manner. Despite extensive research identifying individual enhancers and promoters, a comprehensive understanding of the regulatory landscapes that dictate cellular diversity remains elusive. This gap is largely attributable to limitations in conventional assays regarding throughput, resolution, and the reliance on cultured cells.

Massively parallel reporter assays (MPRAs) have emerged as powerful platforms for systematically interrogating enhancer activity (*Agarwal et al., 2025*; *Arnold et al., 2013*; *Inoue and Ahituv, 2015*; *Lagunas et al., 2023*; *McAfee et al., 2022*; *Melnikov et al., 2012*; *Shen et al., 2016b*). Most MPRAs rely on predetermined sets of DNA fragments, identified through chromatin, RNA, and/or bioinformatic analyses. They are often broad in scale, in some cases, extending to genome-wide analyses. Constructing such libraries is both time-consuming and resource-intensive, and the selection process can be inherently biased. When an investigator aims to identify CRMs for only one or a few genes, most MPRAs are more complex and expensive than is required, and genome-wide MPRA libraries may lack the resolution needed to detect CRMs for genes that are transiently expressed or limited to small cell populations. CRMs for specific genes might not only lead to an understanding of gene regulation, but they can be used for cell type-specific gene expression for perturbations in specific settings and/or for clinical applications. To provide methods for these latter applications, we developed a locus-specific MPRA (LS-MPRA) that leverages bacterial artificial chromosomes (BACs) to generate highly complex libraries for the unbiased interrogation of millions of fragments surrounding genes of interest (*Rabe, 2020*). Fragment size can be chosen by the investigator, which is especially useful when a small fragment might be required, e.g., for cell type-specific expression in a viral vector. The LS-MPRA library approach not only identifies potential CRMs under homeostatic conditions but can be run in a perturbed condition to identify CRMs that might direct a response to e.g., drug treatment. In addition, we devised a degenerate MPRA (d-MPRA) technique to resolve the functional architecture of CRMs at single-nucleotide resolution. By systematically introducing random point mutations into fragments identified in the LS-MPRA, d-MPRA nominates sites required for repression and/or activation of CRMs. Integration of these data with de novo motif discovery, single-cell RNA sequencing (scRNA-seq) data, and CUT&RUN assays can be used to further analyze the connection between CRM activity and specific transcriptional regulators.

In this study, we first determined that these methods can identify CRMs for stably expressed, cell type-specific genes, previously identified by our laboratory and others. These assays were successfully carried out in the mouse in vivo and in explant cultures. We then investigated the regulation of more challenging genes, *Olig2* and *Ngn2*, that are dynamically expressed within subsets of retinal progenitor cells (RPCs) during development. CRMs for Olig2 were run through the d-MPRA method, where binding sites for known and previously unreported TF regulators of retinal development were identified. Analysis of the patterns of Olig2 RNA expression and those of these predicted TF regulators, within single cells, ruled in the possibility that the suggested TFs might regulate Olig2 repression and activation. Binding of the suggested TFs was then carried out and was validated by CUT&RUN assays.

These methods can be launched without any prior knowledge of a gene's architecture, and without synthesizing fragments or producing a library of specific PCR products. They offer a modular framework that can be readily adapted to diverse model systems. The methods are straightforward, requiring only basic molecular biology skills. They are rapid, with library preparation taking 1–2 weeks. Moreover, the methods can be run in a multiplexed fashion, allowing one to combine libraries from several BACs.

## Results
### LS-MPRA for stably expressed cell type-specific genes

The overall LS-MPRA protocol is shown in *Figure 1A*. Fragment libraries from large contiguous genomic regions (~150–250 kb) within a BAC encoding a locus of interest are generated. Essentially, random DNA fragments are generated by digestion by the NEBNext Ultra II FS DNA Module enzyme, followed by size selection on an agarose gel. The fragments are then cloned into a barcode library with a synthetic, minimal promoter (TATAA) and GFP, with an intron in the GFP open reading frame (ORF), to allow selection of GFP cDNA vs contaminating plasmid DNA, for some applications. The

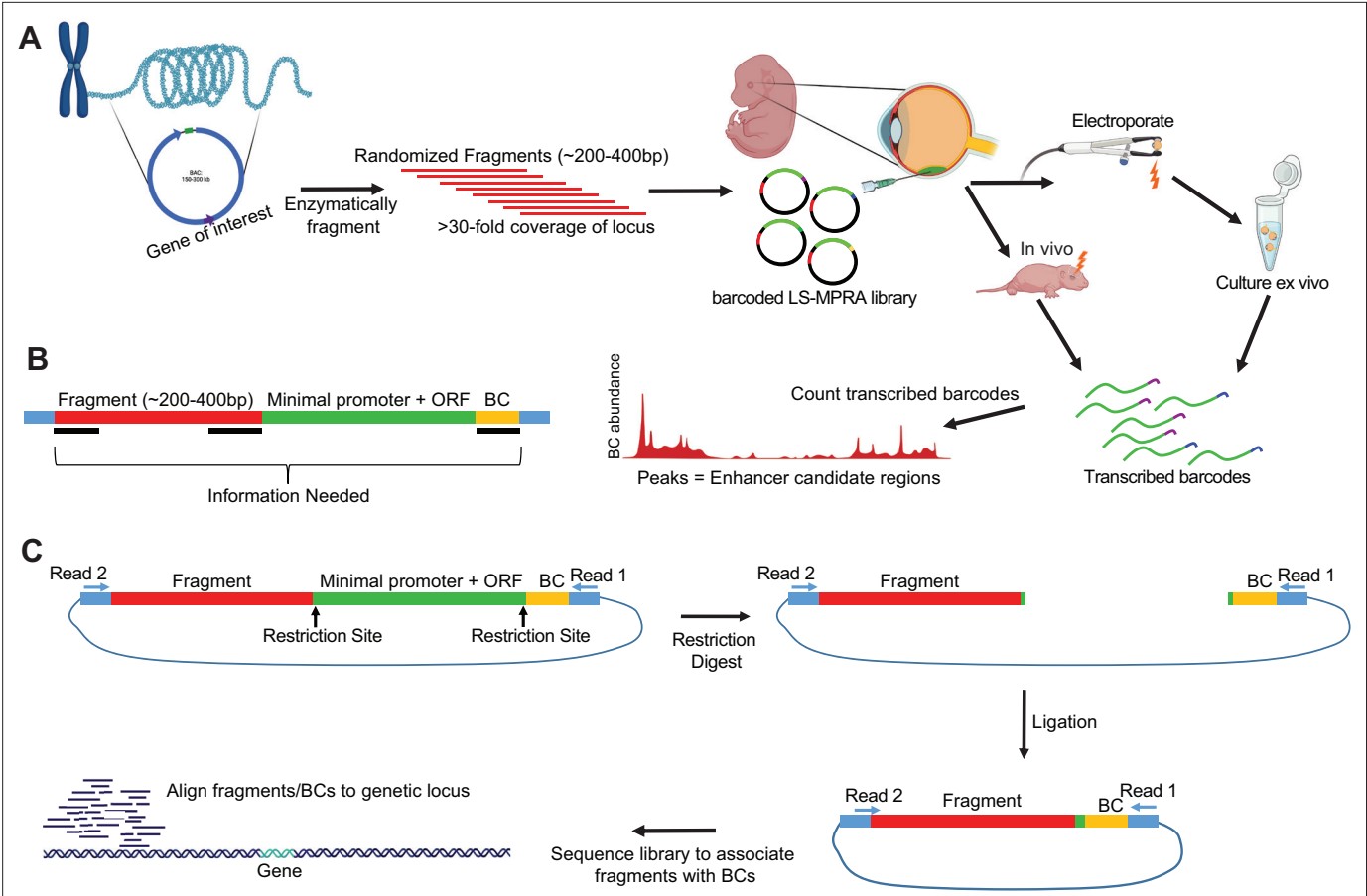

**Figure 1.** Overview of the locus-specific massively parallel reporter assay (LS-MPRA). (**A**) Schematic representation of LS-MPRA. Bacterial artificial chromosomes (BACs) containing genomic regions of interest are enzymatically fragmented to generate a high-complexity DNA library. Size-selected fragments are cloned into a vector containing a minimal promoter that will drive expression of GFP and a barcode positioned within the 3' untranslated region (UTR). The LS-MPRA library is then electroporated into retinal cells, where *cis*-regulatory module (CRM) activity is inferred by quantifying barcode enrichment in transcribed mRNA. (**B**) Illustration of fragment-barcoding strategy and necessary elements to sequence, wherein each DNA fragment is uniquely barcoded. (**C**) Preparation and sequencing of the plasmid library, associating fragment-barcode pairs, and establishing a baseline for barcode abundance, which is used to normalize barcode counts after mapping of barcode-labeled fragments to genomic coordinates. ORF, open reading frame; BC, barcode. Partially created with BioRender.com.

barcodes are located in the 3'UTR and were used to track DNA fragment activity. Because there are many overlapping fragments from a given region, active genomic regions appear as peaks, where barcode signals from each fragment accumulate. Libraries of at least $6.2\times10^6$ and as many as $1.3\times10^7$ fragments were made (***Source data 1***), with each nucleotide in the BAC residing in a different position within at least 5000 fragments and as many as 11,000 fragments. BAC libraries are delivered to the cells of interest, in our case by electroporation into the retina. After incubation, a cDNA library is generated using a RT primer that enables capture of barcoded transcripts, followed by PCR amplification. A gel is then run for isolation of the appropriate sized fragments. The cDNA-derived barcodes are sequenced and mapped to the fragment library. Similar to other MPRAs, CRM activity is inferred from cDNA barcode ratio relative to the barcode frequency in the electroporated library.

In order to associate a given DNA fragment with a particular barcode (***Figure 1B***), an association library, derived from the original LS-MPRA library, is created and sequenced. This association library is made by excision of the minimal promoter and GFP, followed by ligation (***Figure 1C***). This is followed by paired-end sequencing, from the barcode at the 3' end and the fragment with a potential CRM from the 5' end. This allows for the association of each fragment with a specific barcode. This baseline sequencing serves two additional purposes. One is the normalization of the barcode reads recovered from electroporated tissues or cells to their frequency within the electroporated library, ensuring that

barcode enrichments reflect true CRM activity rather than variability in fragment representation. The second is that it allows mapping of each fragment to the BAC and an assessment of library coverage across the BAC.

As an initial test of the LS-MPRA approach, BAC clones containing genes previously analyzed for CRM activity were selected. These genes are stably expressed in specific cell types of the mature retina. They were *Rho* (rhodopsin), expressed in rod photoreceptors, *Cabp5* (calcium-binding protein 5), expressed in subsets of cone bipolar cells and in rod bipolar cells (*Haeseleer et al., 2000*; *Shekhar et al., 2016*), *Grm6* (metabotropic glutamate receptor 6), expressed in all ON-bipolar cells (*Ueda et al., 1997*), and *Vsx2* (visual system homeobox 2; homolog of *Chx10*), expressed in all bipolar cells and some Müller glial cells (*Hatakeyama et al., 2001*; *Liu et al., 1994*; *Rowan and Cepko, 2004*). LS-MPRA libraries for each of the BACs encoding these genes were made as described above (library characteristics in *Source data 1*). The libraries were pooled and electroporated into postnatal day 3 (P3) retinas in vivo or ex vivo, with sacrifice and/or tissue collection 7 days later. This developmental window was chosen as it is the period when rod photoreceptors and bipolar cells are generated from RPCs, and RPCs are readily electroporated, while postmitotic cells are not. The Rho library was present at 5% the level of the other libraries as rods are the predominant cell type produced at this time (*Young, 1985*), and *Rho* is a highly expressed gene. This and all LS-MPRA experiments in this study were performed using three replicates, with each replicate generated by pooling retinas from

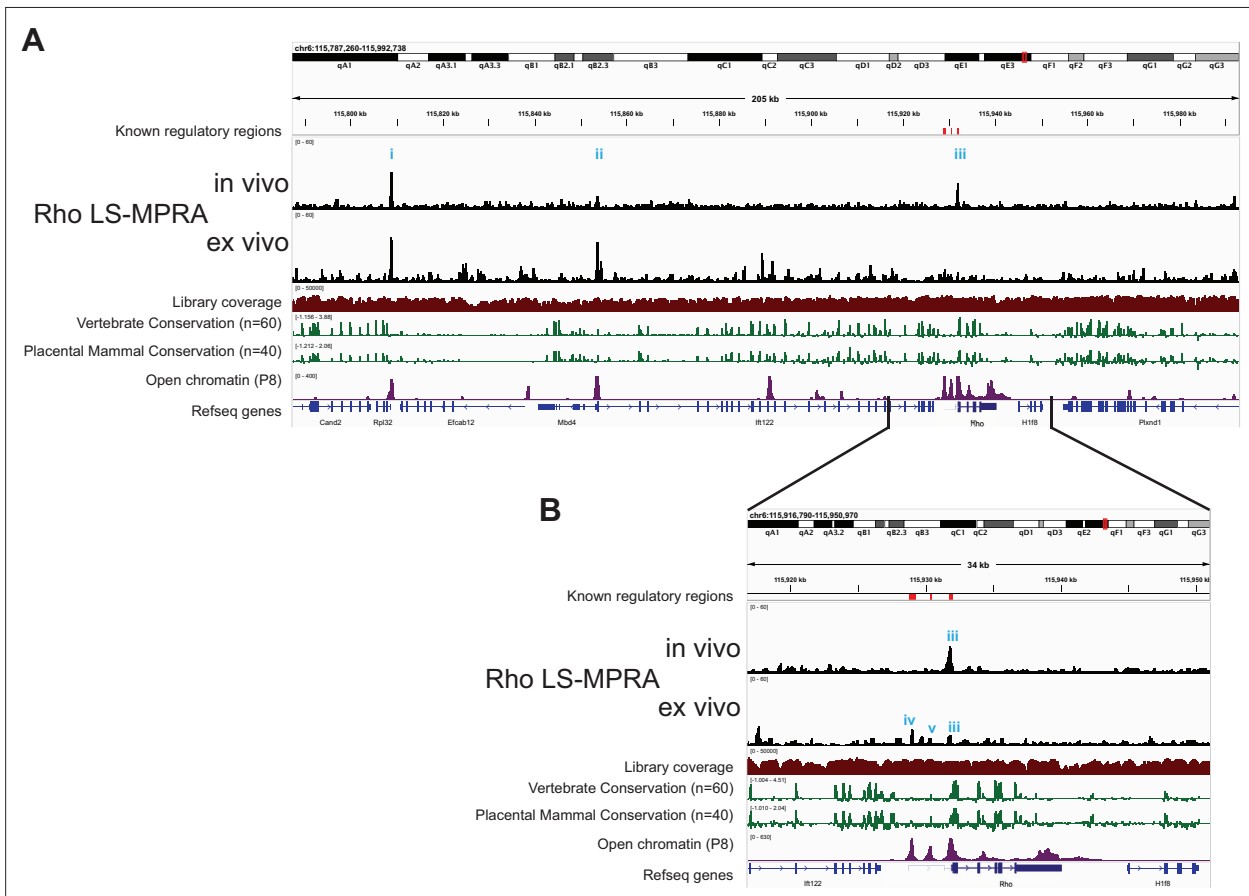

**Figure 2.** Rho locus-specific massively parallel reporter assay (LS-MPRA) to identify *cis*-regulatory modules (CRMs) in the neonatal mouse retina. (**A**) LS-MPRA barcode enrichment plots for the Rho locus aligned to a genome browser track from in vivo and ex vivo experiments (N=3 experimental replicates, combined), and annotated with: (i) known *Rho* regulatory regions, (ii) the coverage of the barcode–fragment association library across the locus, (iii) $\log_2$-transformed base conservation among 60 vertebrate and 40 placental mammal species, (iv) regions of open chromatin in the P8 mouse retina, and (v) RefSeq gene models for the locus. (**B**) Expanded view of a region of interest from the Rho LS-MPRA plot, showing peaks (i-v) that correspond to known regulatory regions (red bars), genomic conservation, and areas of open chromatin.

The online version of this article includes the following figure supplement(s) for figure 2:

**Figure supplement 1.** Grm6, Vsx2, and Cabp5 LS-MPRAs to identify *cis*-regulatory modules (CRMs) in the neonatal mouse retina.

several animals. Three retinas were pooled following in vivo electroporation, and 3–4 retinas were pooled following ex vivo electroporation and culture.

Analysis of the sequences of the fragment-association libraries showed that all regions of the BACs were fairly equally represented when aligned to the mouse mm10 reference genome (Library Coverage plots; *Figure 2*, *Figure 2—figure supplement 1*). When the frequencies of cDNA barcodes representing different genomic regions of each BAC were plotted across the genomic regions, clear signals were observed for all 4 BACs (LS-MPRA plots; *Figure 2*, *Figure 2—figure supplement 1*). The barcode frequency plot for the Rho LS-MPRA library revealed distinct candidate CRM regions (peaks; *Figure 2A*) in one, or both, of the in vivo and ex vivo conditions. Several of these regions are conserved across vertebrate and placental mammal species, correspond to areas of open chromatin, as indicated by ATAC-seq data (*Lyu et al., 2021*), and are annotated as candidate *cis*-regulatory elements in the mouse ENCODE database (not shown; *ENCODE Project Consortium et al., 2020*). Consistent with previous reports identifying the Rho proximal promoter region (PPR) at −176 bp to +70 bp relative to the *Rho* transcription start site (TSS) (*Zack et al., 1991*), the in vivo Rho LS-MPRA revealed a clear CRM peak at −251 bp to +65 bp (peak **iii**; *Figure 2A*). The expanded region of interest (ROI; *Figure 2B*) clearly demarcates the CRM peak (peak **iii**), which correlates with the ATAC-seq open chromatin profiles. Two additional Rho CRMs known to induce reporter expression in vitro—the rhodopsin enhancer region (RER; −1.575 kb to −1.476 kb) and the Crx bound region (CBR; −3.165 kb to −2.693 kb) (*Corbo et al., 2010*; *Nie et al., 1996*)—exhibited minor peaks only in the ex vivo LS-MPRA (peaks **iv, v**).

In addition to the CRM peaks directly associated with *Rho*, two peaks were identified by the Rho LS-MPRA at approximately −123 kb (peak **i**) and −78 kb (peak **ii**) upstream of the *Rho* TSS. These peaks localize within the putative promoter regions of *Rpl32* (a ribosomal RNA gene) and *Ift122* (encoding an intraflagellar transport protein), respectively, suggesting that the LS-MPRA captured regulatory activity in addition to that of *Rho*. To investigate this possibility, we analyzed publicly available single-cell RNA sequencing (scRNA-seq) data from P8 mouse retina (*Clark et al., 2019*). In this dataset of 10,200 retinal cells, *Rpl32* was detected in 7916 cells and *Ift122* in 1,092 cells, while *Rho* was detected in 8809 cells. The fragment immediately upstream of *Rpl32* was cloned into a plasmid upstream of a TATAA minimal promoter and EGFP reporter, and electroporated in vivo into P3 retinas and incubated for 7 days before analysis. GFP expression driven by this fragment was expressed in most electroporated cells (data not shown). Follow-up experiments might reveal that this small CRM (237 bp) is active in most tissues, suggesting it would be useful for applications where such breadth of expression driven by a small CRM is desirable. These expression patterns support the notion that some of the CRM activity captured by the Rho LS-MPRA reflects regulatory elements associated with neighboring genes that are co-expressed in retinal cells during this stage of development.

LS-MPRA plots for bipolar cell subtype markers also revealed several relevant peaks. In the Grm6 LS-MPRA, a distinct peak at −8.152 kb to −7.857 kb relative to the TSS (peak **i**, *Figure 2—figure supplement 1A*) overlapped with a previously identified CRM (−8.126 kb to −7.926 kb; *Kim et al., 2008*), which is moderately conserved across vertebrate and placental mammal species and localizes to an area of open chromatin. In the Vsx2 LS-MPRA, two distinct peaks at −37.617 kb to −37.364 kb and at −17.833 kb to −17.559 kb relative to the TSS (peaks **ii, iii**; *Figure 2—figure supplement 1B*) overlapped with a Vsx2 enhancer active in RPCs (−37.726 kb to −37.117 kb *Buenaventura et al., 2018* or −37.778 kb to −36.813 kb *Honnell et al., 2022*) and a Vsx2 enhancer active in bipolar cells (−17.724 kb to −17.560 kb *Kim et al., 2008* or −19.654 kb to −15.836 kb *Honnell et al., 2022*), respectively. Both peaks are conserved across vertebrate and placental mammal species and localize to areas of open chromatin. However, a broad region previously described as a Vsx2 enhancer (−2.145 kb to +87 bp; *Rowan and Cepko, 2005*) did not align with any peaks in the LS-MPRA. Similarly, no peaks in the Cabp5 LS-MPRA (*Figure 2—figure supplement 1C*) corresponded to the known Cabp5 CRM at −289 bp to +156 bp relative to the TSS (*Kim et al., 2008*). These results suggest that while LS-MPRA successfully identified several previously characterized CRMs, some known elements did not exhibit significant activity under these experimental conditions.

## LS-MPRA for genes expressed dynamically during retinal development

We are interested in how different RPCs produce different types of retinal cells. Previous work has suggested that there are intrinsic differences among different RPCs over time, and even at one time

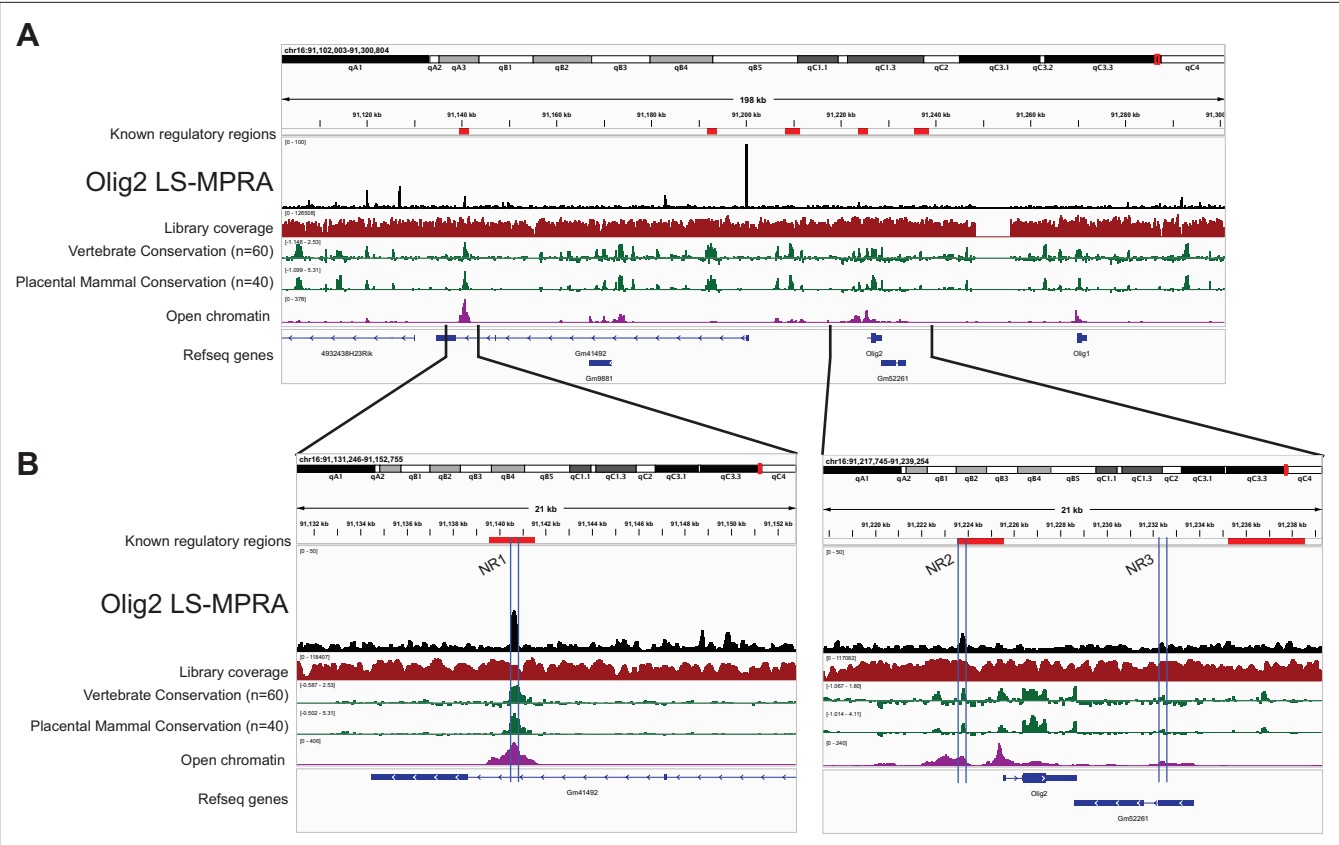

**Figure 3.** Olig2 locus-specific massively parallel reporter assay (LS-MPRA) to identify candidate *cis*-regulatory modules (CRMs) in the developing mouse retina. (**A**) Barcode enrichment plot from the Olig2 LS-MPRA, aligned with a genome browser track and annotated with (i) previously described *Olig2* regulatory regions identified in mouse embryonic stem cells, fertilized murine oocytes, a mouse lymphoma cell line, or in mouse ventral neural tube (*Chen et al., 2008*; *Fan et al., 2023*; *Friedli et al., 2010*; *Sun et al., 2023*), (ii) coverage of the barcode-fragment association library across the locus, (iii) log$_2$ base conservation across 60 vertebrate or 40 placental mammal species, (iv) regions of open chromatin in the E14 mouse retina, and (v) RefSeq gene models. (**B**) Two expanded regions of interest from the Olig2 LS-MPRA, showing peaks that align with known regulatory elements, genomic conservation, and open chromatin. Candidate CRMs identified within blue bars.

in development (*Cepko, 2014*). We previously found that Olig2, a transiently expressed proneural basic helix-loop-helix (bHLH) transcription factor, marks a subset of RPCs at each time in development (*Hafler et al., 2012*; *Shibasaki et al., 2007*; *Trimarchi et al., 2008*). It is expressed in neurogenic RPCs that divide one or two times and produce a limited repertoire of cell types among those born at that time. In the embryonic day 13 or 14 (E13–14) retina, cone photoreceptors and horizontal cells, a type of interneuron, are produced. In the postnatal period, Olig2+ RPCs produce rod photoreceptors and amacrine interneurons. We were interested in the regulators of Olig2 at these two different times, as different CRMs might lead us to the intrinsic differences between embryonic and postnatal RPCs. The Olig2 LS-MPRA library was thus generated (library characteristics in *Source data 1*) and delivered to E14 mouse retinas, which were incubated ex vivo for 28 hr. For each of the three experimental replicates, eight retinas were pooled. Barcode enrichment plots from this library revealed distinct candidate CRM regions (peaks; *Figure 3A*), several of which are conserved across vertebrate and placental mammalian species and correspond to areas of open chromatin. Notably, two CRM candidate regions identified by LS-MPRA (*Figure 3B*), located at –1.881 kb to –1.700 kb and –85.027 kb to –84.781 kb relative to the Olig2 TSS, overlap with broader regions previously implicated as potential Olig2 regulatory elements in mouse embryonic stem cells (*Chen et al., 2008*). Other regions with possible regulatory activity (indicated in *Figure 3A*), previously identified using in vitro assays or in the mouse spinal cord (*Fan et al., 2023*; *Friedli et al., 2010*; *Sun et al., 2006*), did not exhibit CRM activity in the retina by LS-MPRA.

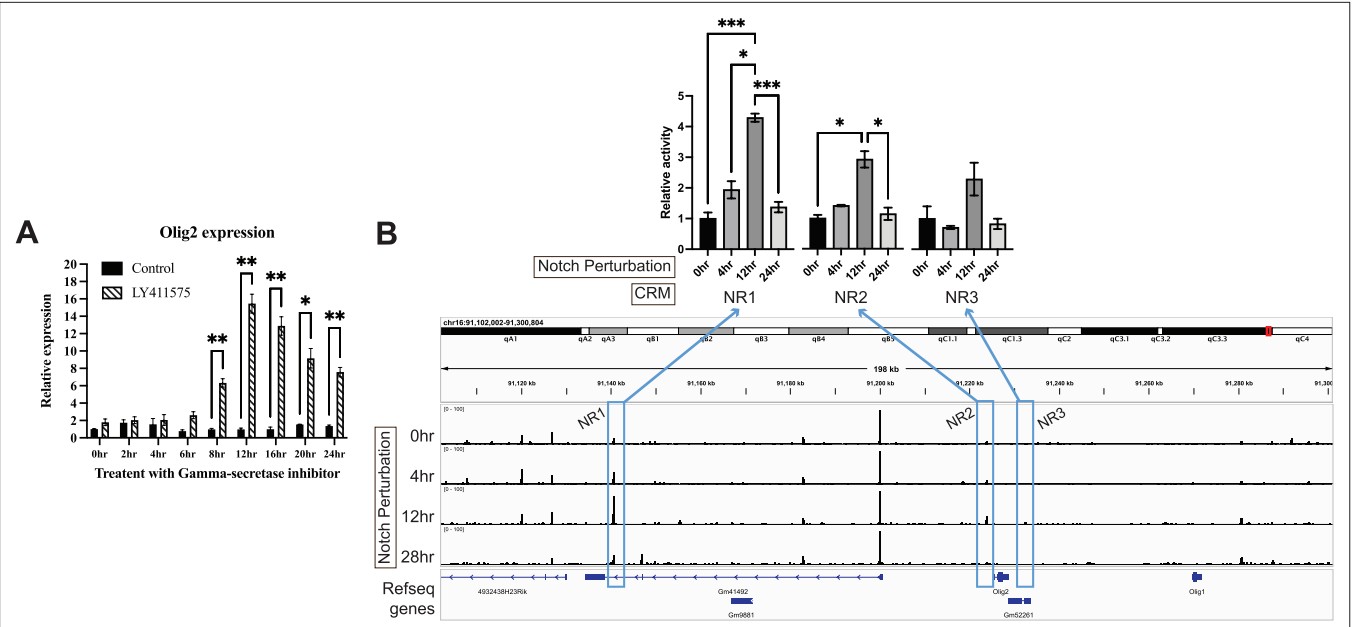

**Figure 4.** Dynamic cis-regulatory module (CRM) activity and *Olig2* expression following inhibition of Notch signaling. (**A**) Bar plot showing relative endogenous *Olig2* RNA expression in the retina over 0–24 hr of treatment with the γ-secretase inhibitor LY411575 or DMSO, normalized to the 0 hr control. Unpaired *t*-tests with Holm's multiple comparisons correction were performed for each Control vs. Treated (n=3) timepoint. (**B**) Barcode enrichment plot from the Olig2 locus-specific massively parallel reporter assay (LS-MPRA), aligned with a genome browser track after 0–28 hr of LY411575 treatment. Notch inhibitor-responsive CRM candidate regions (NR1–3, blue boxes) displayed differential barcode enrichment across the treatment period. Statistical analysis was performed using one-way Welch's ANOVA with Dunnett's T3 multiple comparisons test on AUC values, normalized to the 0 hr timepoint. Error bars in (**A**) and (**B**) represent SEM. Experimental replicates (n=3): each pooled 8 retinas. Asterisks indicate statistical significance (*$p \leq 0.05$, **$p \leq 0.01$, ***$p \leq 0.001$).

The online version of this article includes the following figure supplement(s) for figure 4:

**Figure supplement 1.** Olig2 locus-specific massively parallel reporter assay (LS-MPRA) analysis using unique barcodes.

The peaks at -1.881 kb to –1.700 kb were quite small, so an additional assay was run to determine if they were relevant for Olig2 regulation. Knock-out mice for Notch1 have been shown to overproduce cone photoreceptors during the embryonic period and rod photoreceptors in the neonatal period (*Jadhav et al., 2006*; *Yaron et al., 2006*). Moreover, Olig2 expression in mouse retinal explants is also transiently upregulated in response to Notch inhibition (*Kaufman et al., 2019*). To investigate the precise dynamics of Notch regulation of Olig2 in the E14 retina, as well as to determine if the small CRMs identified in the LS-MPRA were of relevance to Olig2 regulation, E14 retinal explants were treated with the γ-secretase inhibitor, LY411575, which inhibits Notch signaling. Notch inhibition was found to increase endogenous *Olig2* RNA levels in E14 retinas over 24 hr of treatment, with a peak at 12 hr (*Figure 4A*). Two small Olig2 CRM candidates, one just 5' (–1.881 kb to –1.700 kb) and one 3' (+6.721 kb to +6.933 kb) of the Olig2 TSS, also exhibited an increase in activity, also with a peak at 12 hr post Notch inhibition (*Figure 4B*). In addition, one CRM within the intron of the Gm41492 predicted gene, at –84.781 kb to –85.027 kb relative to the Olig2 TSS, also exhibited an increase in CRM activity with Notch inhibition. The three Olig2 potential CRMs, now named Notch signaling-responsive (NR) regions 1, 2, and 3, were conserved and exhibited open chromatin. Reanalysis of one replicate using only unique barcodes also identified these three peaks, including NR3, indicating that barcode collisions did not account for the minor NR3 peak (*Figure 4—figure supplement 1A*). Together, these findings suggest that NR1–3 function as CRMs that respond to disruptions in Notch signaling to enhance Olig2 expression in RPCs that produce cone photoreceptors. More broadly, these experiments demonstrate that the LS-MPRA can be used to rapidly identify CRMs that respond to perturbations that are relevant to the gene under study.

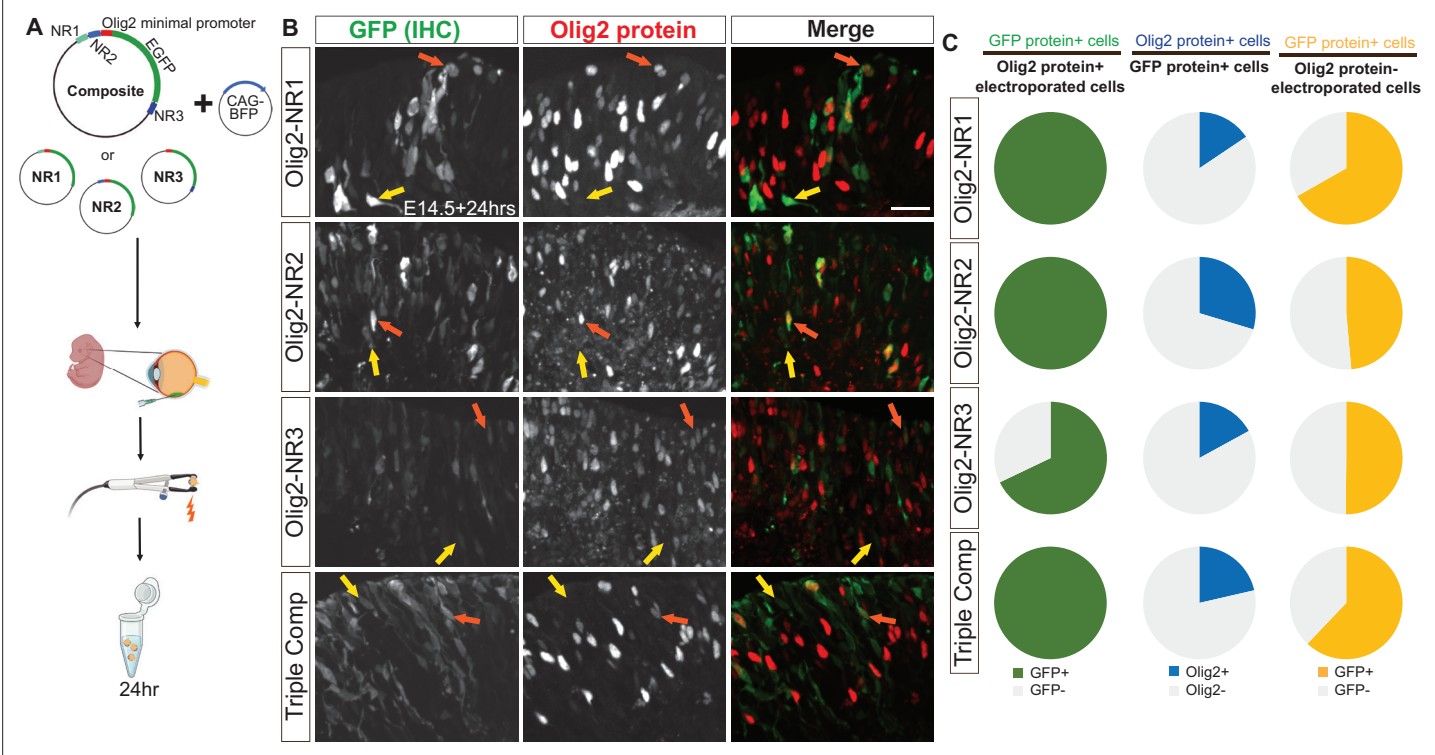

**Figure 5.** Co-localization of GFP driven by Olig2 *cis*-regulatory modules (CRMs) and Olig2 in retinal cells. (**A**) Schematic of plasmid constructs containing one or all Notch inhibitor-responsive regions driving GFP expression with the endogenous *Olig2* minimal promoter, alongside the experimental workflow. (**B**) Representative transverse sections of E14 retinas electroporated with these plasmids, incubated in vitro for 24 hr, and stained using immunohistochemistry (IHC) for GFP and Olig2. Merged and single-channel images show GFP colocalization with Olig2 (orange arrow) as well as GFP⁺ cells lacking detectable Olig2 expression (yellow arrow). (**C**) Quantification of GFP and Olig2 co-localization, represented as pie charts showing (i) the percentage of electroporated Olig2+ cells that express GFP protein, (ii) the percentage of GFP protein+ cells that express Olig2, and (iii) the percentage of electroporated Olig2- cells that express GFP protein in retinas electroporated at E14 with plasmids containing one or all Notch inhibitor-responsive regions driving GFP. Scale bar: 20 μm. Partially created with BioRender.com.

The online version of this article includes the following figure supplement(s) for figure 5:

**Figure supplement 1.** Activity of backbone plasmids containing the *Olig2* minimal promoter and *EGFP*.

## Analysis of cell type-specific CRM activity

To investigate the cellular specificity of the Olig2 CRMs, NR1, NR2, and NR3, each fragment was cloned individually or as a composite into a plasmid with the endogenous Olig2 promoter and TSS (–75 bp to +51 bp) upstream of an EGFP reporter. First, to determine whether the inclusion of the Olig2 promoter, and/or other sequences within the plasmid, had background transcriptional activity, we electroporated backbone plasmids containing GFP alone or GFP under the Olig2 minimal promoter into E14 retinas (*Figure 5—figure supplement 1A*). Minimal and sporadic *GFP* RNA signal was observed in these controls. This background level was judged to be low enough to allow for assay of specific CRM activity. The Olig2 CRMs were cloned into the backbone plasmid, oriented as they appear in the genome relative to the TSS. These constructs were electroporated into E14 retinas ex vivo and incubated for 24 hr before analysis (*Figure 5A*).

Electroporated cells were identified by expression of the CAG-BFP co-electroporated control plasmid. Immunolabeling for GFP and Olig2 in transverse retinal sections (*Figure 5B*) revealed that all Olig2+ electroporated cells, identified by immunohistochemistry (IHC), expressed GFP, also identified by IHC, when driven by NR1, NR2, or the combined NR1-3 composite; additionally, most Olig2+ electroporated cells expressed GFP when driven by NR3 alone, albeit at lower levels (*Figure 5C*, left column; *Supplementary file 1c*). These findings suggest that NR1, NR2, and NR3 contain functional CRMs for Olig2, with NR3 exhibiting very weak CRM activity. However, when the GFP+ electroporated cells were examined for Olig2 protein expression using IHC, few GFP+ cells demonstrated detectable Olig2 protein (*Figure 5C*, middle column; *Supplementary file 1c*), and many Olig2- electroporated

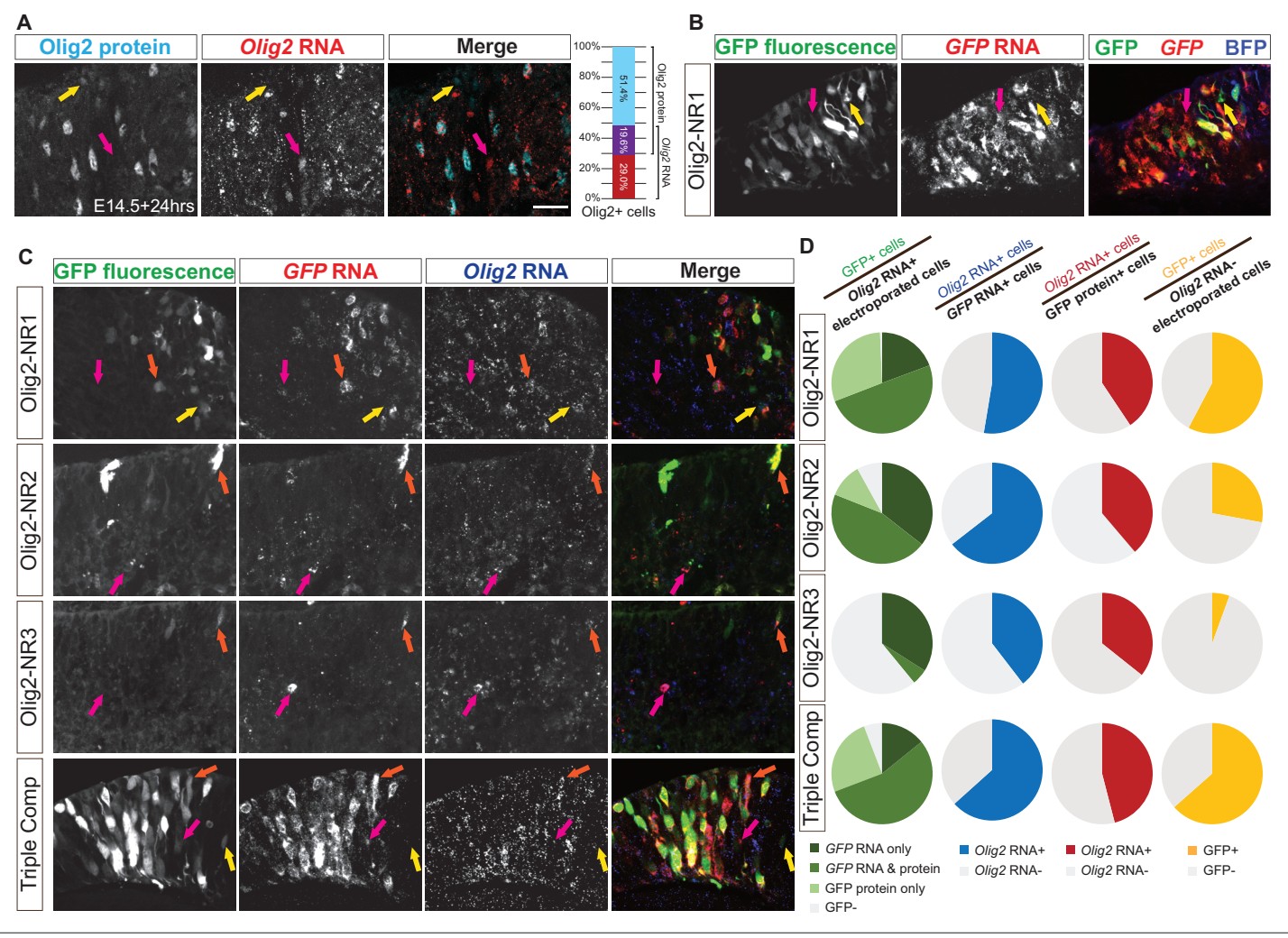

**Figure 6.** Co-localization of *cis*-regulatory module (CRM)-directed RNA and *Olig2* RNA or protein in retinal cells. (**A–C**) Representative transverse sections of E14 retinas incubated in vitro for 24 hr for analysis of localization patterns of Olig2 and/or GFP. (**A**) Sections stained for Olig2 protein (cyan, yellow arrows) using immunohistochemistry (IHC) and *Olig2* RNA (red, magenta arrows) using FISH. A 100% stacked column plot quantifies the overlap between Olig2 protein and *Olig2* RNA expression. (**B**) Intrinsic GFP fluorescence and *GFP* RNA driven by Olig2-NR1 (yellow and magenta arrows). (**C**) In retinas electroporated with plasmids containing one or all Notch inhibitor-responsive regions, *Olig2* RNA co-localized with GFP protein only (yellow arrow), *GFP* RNA only (magenta arrow), or both (orange arrow). (**D**) Pie charts quantifying (column 1) the percentage of electroporated *Olig2* RNA+ cells that expressed GFP, (column 2) the percentage of *GFP* RNA+ cells that expressed *Olig2* RNA, (column 3) the percentage of GFP protein+ cells that expressed *Olig2* RNA, and (column 4) the percentage of *Olig2* RNA- cells that expressed GFP in retinas electroporated. Scale bar: 20 µm.

cells expressed GFP (*Figure 5C*, right column; *Supplementary file 1c*). This lack of correlation raised two possibilities: either NR1-3 CRM activities were not confined to Olig2+ RPCs, and/or the transient nature of Olig2 expression, combined with the stability of EGFP, resulted in a temporal mismatch that limited the detection of co-localized Olig2 protein and GFP protein. This possibility was suggested by the known short half-life of murine Olig2 RNA (~1.66 hr) and protein (~3.5 hr) (*Rayon et al., 2020*).

We first determined if the short half-lives of Olig2 RNA and protein created a situation where detection of the endogenous Olig2 transcript and protein in the same cell was unlikely. Fluorescent in situ hybridization (FISH) for *Olig2* transcript, combined with immunolabeling for Olig2 protein, showed that most cells expressed either the transcript or the protein, but not both simultaneously (*Figure 6A*). A similar pattern was observed when comparing GFP protein expression and *GFP* transcript levels in retinas electroporated with the NR1 CRM (*Figure 6B*). This observation might indicate that the NR1 CRM activity is also dynamically regulated. To at least partially accommodate this temporal mismatch when assessing the specificity of the Olig2 CRMs, we examined RNA, rather than

protein, for Olig2 and GFP, reasoning that RNA might be a more accurate read-out of both the CRM activity and endogenous Olig2 transcription.

FISH for *Olig2* and *GFP* RNA, along with intrinsic GFP fluorescence (*Figure 6*), revealed a pattern similar to the quantification of immunostaining (*Figure 5C*). The vast majority of *Olig2* RNA+ electroporated cells expressed *GFP* RNA and/or protein when GFP expression was driven by NR1, NR2, or the composite construct (*Figure 6D*, left column; *Supplementary file 1d*). Approximately 40% of *Olig2* RNA+ electroporated cells co-expressed GFP when driven by NR3 alone, further supporting its lower but detectable CRM activity. Importantly, a substantial proportion of *GFP* RNA+ cells co-expressed *Olig2* RNA when GFP was driven by NR1, NR2, or the composite construct (*Figure 6D*, middle-left column; *Supplementary file 1d*), and this proportion exceeded that observed when the analysis was limited to GFP protein+ cells (*Figure 6D*, middle-right column; *Supplementary file 1d*). When the expression of GFP RNA or protein was compared in cells that expressed or did not express *Olig2* RNA, a greater fraction of Olig2+ cells expressed GFP when it was driven by any of the four constructs (Mann-Whitney test with Holm-Šídák method for correcting multiple comparisons, $p < 0.05$ each). These findings indicate a significant enrichment of CRM activities in the Olig2+ cells. In

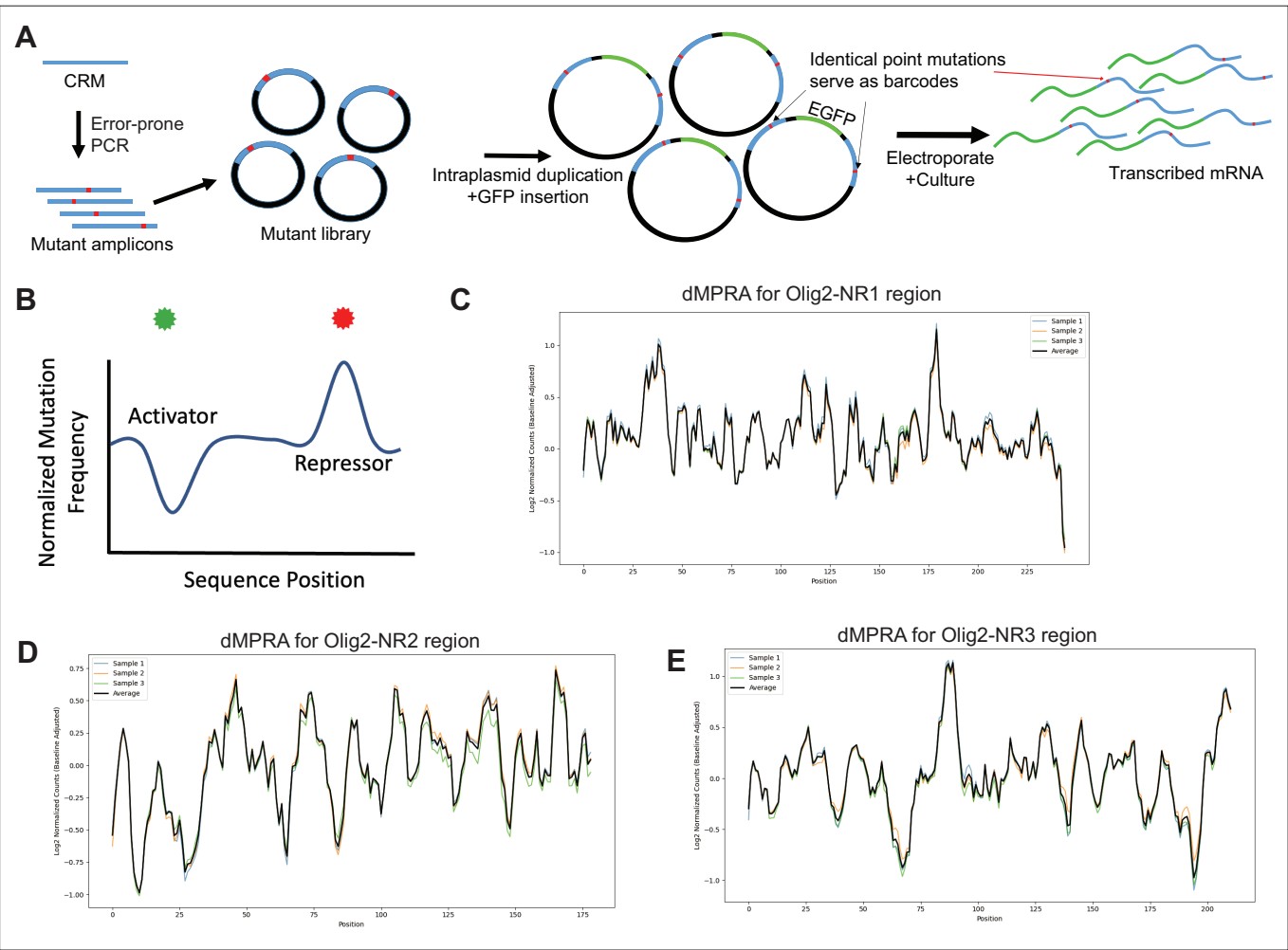

**Figure 7.** Degenerate massively parallel reporter assays (MPRAs) to identify functional residues within candidate *cis*-regulatory modules (CRMs). (**A**) Schematic of d-MPRA library assembly. Point mutations were introduced into *Olig2* CRM fragments via error-prone PCR, followed by intraplasmid duplication (IPD) to generate constructs with duplicated mutant CRMs flanking a minimal promoter and GFP ORF. The 3' CRM copy, located in the 3' untranslated region (UTR), served as a barcode, with a WPRE sequence included to potentially stabilize transcripts. (**B**) Conceptual diagram illustrating expected d-MPRA results, showing predicted changes in CRM activity upon disruption of enhancer or repressor binding sites. (**C–E**) d-MPRA plots displaying log₂ fold changes in mutational frequencies using a 5-base pair sliding window average, normalized across the *Olig2*-NR1 (**C**), *Olig2*-NR2 (**D**), and *Olig2*-NR3 (**E**) regions.

addition, these findings suggest that measuring co-localization at the RNA level may provide a more accurate assessment of CRM activity for transiently expressed genes.

## D-MPRA for identification of sequences important for CRM activity

To identify nucleotides within potential CRMs that are necessary for their activity, we developed the d-MPRA. Most nucleotides identified are suspected to be within TF binding sites, though other roles, e.g., sites that affect chromatin structure, might also play a role. Libraries with point mutations were created via error-prone PCR amplification of each NR1-3 CRM. Each library was then cloned into a vector backbone. An intraplasmid duplication (IPD) was then made to serve as a barcode before inserting a GFP reporter into the completed library (*Figure 7A*). In the IPD process, the mutant library was nicked on the plus strand upstream and the minus strand downstream of the mutated CRM fragments. Strand-displacing polymerase, lacking exonuclease activity, extended these nicks, linearizing the libraries such that each end carried an exact copy of a particular mutated fragment. The final step involved inserting a minimal promoter and GFP ORF cassette between the two duplicated CRM copies. The 3' copy of the mutated CRM was thus in the 3' UTR and served as a barcode in the cDNA (*Figure 7A*). This configuration allowed the possibility that the 5' and/or 3' mutated CRM could influence transcription of GFP. In addition, it was possible that the 3' fragment could influence the stability of the transcript. To somewhat override this effect, a WPRE sequence was also included in the 3' UTR of all library members. The IPD method was used so that the step of making and sequencing the association library, as was done for the LS-MPRA, was unnecessary.

Mutations reducing positive activity, presumed TF binding that enhances activity, should result in depleted barcode abundance, while those diminishing repressor TF binding should lead to enrichment (*Figure 7B*). Similar to LS-MPRAs, the abundance of barcodes of mutated fragments was normalized to their baseline abundance in the d-MPRA plasmid libraries.

The d-MPRA libraries for Olig2 NR1-3 CRMs were constructed with a mutational frequency of ~0.02–0.2% per nucleotide and a complexity of at least $2\times10^7$ transformants (see *Source data 2* for mutational rate, co-occurrence across CRMs, and analyses comparing bulk fragments vs singleton mutations). These libraries were electroporated into E14 ex vivo retinas, which were incubated for 24 hr before RNA extraction. The 3' UTR region containing transcribed mutated CRMs was reverse transcribed to generate cDNA, PCR amplified, gel purified, indexed, and sequenced. Normalized log2-transformed values were averaged across replicates, and a sliding window average (window size = 5) was applied to smooth the data and reduce noise. This approach ensured robust comparisons across experimental conditions. The resulting d-MPRA plots for Olig2-NR1 (*Figure 7C*), Olig2-NR2 (*Figure 7D*), and Olig2-NR3 (*Figure 7E*) revealed peaks and troughs, indicating that mutations at multiple sites influence CRM activity. In three independent electroporation experiments for each CRM, the d-MPRA readouts exhibited low variance between experimental replicates. The clear alignment of peaks and troughs with specific nucleotide positions supports the notion that these regions are important for CRM activity.

To identify over-represented sequences in sites corresponding to peaks or troughs in the d-MPRA plots, de novo motif discovery analysis, HOMER, was performed. This analysis focused on all three Olig2 CRMs, with an emphasis on NR2, which is proximal to the *Olig2* TSS. Across the Olig2-NR2 region, 16 de novo motifs were identified (*Figure 8A*), several of which overlapped with d-MPRA peaks or troughs. The presence and density of binding motifs in this region are consistent with the open chromatin profile observed in ATAC-seq data. Notably, motifs matching known binding sites for Foxn4, Pax6, Mybl1, and Otx2—recognizable regulators of retinal development (*Cvekl and Callaerts, 2017*; *Farhy et al., 2013*; *Kaufman et al., 2019*; *Li et al., 2004*; *Marquardt et al., 2001*; *Nishida et al., 2003*; *Oron-Karni et al., 2008*; *Remez et al., 2017*; *Wang et al., 2014*)—were detected (*Figure 8B–D*). Mybl1 and Otx2 motifs (Motifs 1 and 14, respectively) corresponded to troughs in the d-MPRA plot, whereas the shared motif (Motif 3) for Foxn4 (typically an enhancer) and Pax6 (frequently a repressor) did not align with any prominent peaks or troughs (*Figure 8A*). The shared motif for Foxn4 and Pax6 in Olig2-NR2 likely represents a regulatory balance between activation and repression, reflecting the complex interplay of these TFs in fine-tuning Olig2 expression.

To assess whether RPCs that express Olig2 could use these TFs to regulate *Olig2*, scRNAseq data (*Clark et al., 2019*) were examined for co-expression with *Olig2* at E14. The dataset, originally generated using the 10X Genomics platform, was further processed in Seurat, where filtering, normalization,

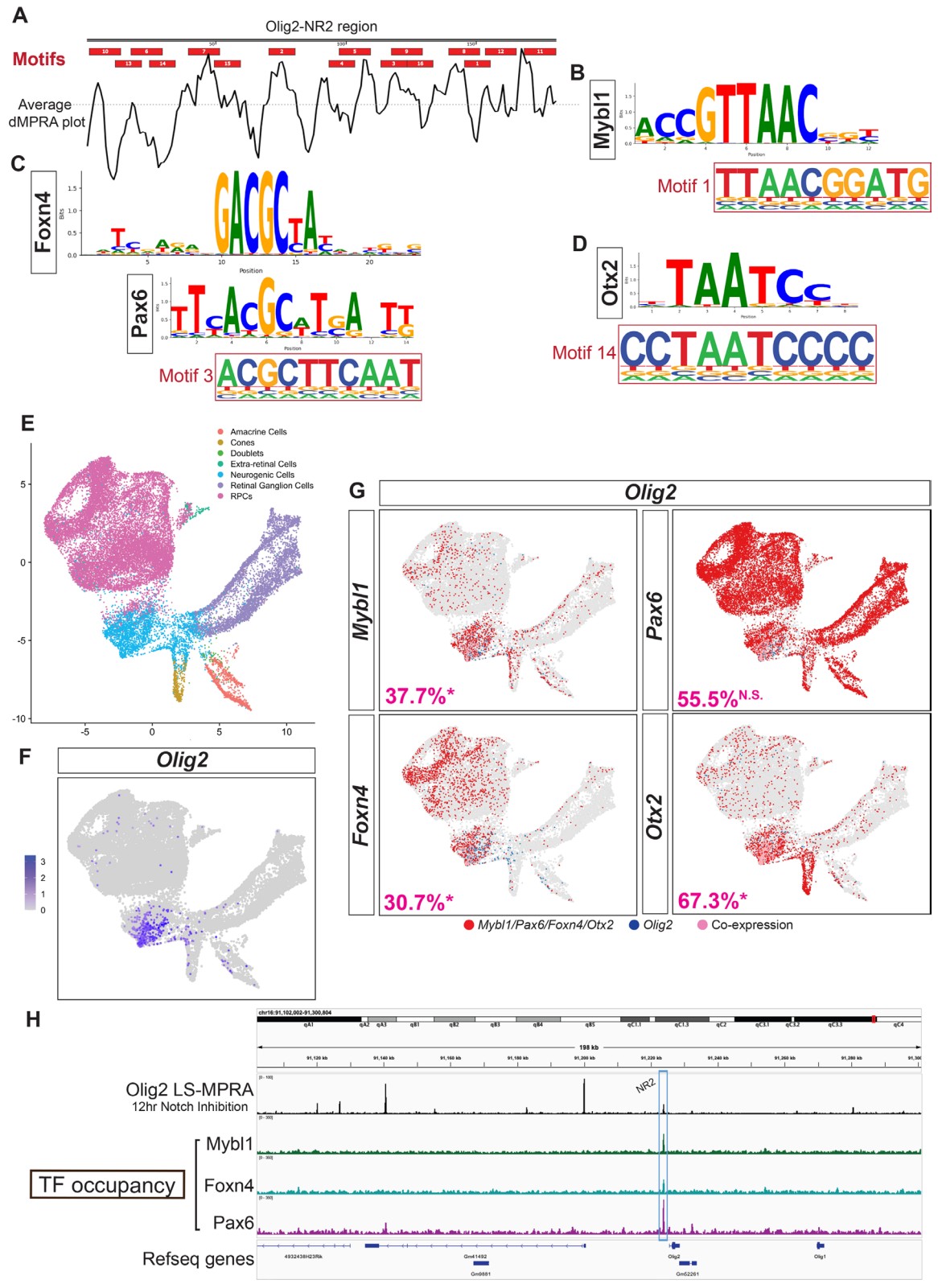

**Figure 8.** Transcription factor (TF) binding sites within the Olig2-NR2 *cis*-regulatory module (CRM). (**A**) TF binding motifs predicted using HOMER identified within the *Olig2*-NR2 CRM candidate, aligned with the average d-massively parallel reporter assay (MPRA) plot for this region (from **Figure 7**). (**B–D**) Position frequency matrices of TF binding sites aligned to motifs predicted by HOMER in *Olig2*-NR2: Mybl1 to Motif 1 (**B**), Foxn4 and Pax6 to Motif 3 (**C**), and Otx2 to Motif 14 (**D**). (**E**) UMAP visualization of scRNA profiles from E14 mouse retinas, with cell types previously identified by marker

*Figure 8 continued on next page*

*Figure 8 continued*

genes by *Clark et al., 2019*. (**F**) Expression of *Olig2* on UMAP. (**G**) Co-expression (pink) of *Olig2* (blue) with *Mybl1, Foxn4, Pax6,* or *Otx2* (red) on UMAP. Percentages of respective populations that co-express *Olig2* are indicated (* denotes significance using Fisher's exact test, $p \leq 0.05$). (**H**) TF occupancy track aligned to (i) *Olig2* locus-specific massively parallel reporter assay (LS-MPRA) barcode enrichment after 12 hr LY411575 treatment (replicate of *Figure 4B*) and (ii) gene models. Occupancy peaks for Mybl1, Foxn4, and Pax6 align with the Olig2-NR2 CRM candidate (blue box).

The online version of this article includes the following figure supplement(s) for figure 8:

**Figure supplement 1.** Transcription factor (TF) Binding sites in Olig2-NR1 and NR3 *cis*-regulatory modules (CRMs).

and dimensionality reduction was carried out to identify distinct retinal cell populations (*Figure 8E*). As expected, *Olig2* expression was enriched in a subset of neurogenic RPCs (*Figure 8F*). Notably, *Mybl1, Pax6, Foxn4,* and *Otx2* were co-expressed with *Olig2* in subsets of these cells (*Figure 8G*), with 37.7%, 55.5%, 30.7%, and 67.3% of neurogenic RPCs that expressed *Olig2* also expressed *Mybl1, Pax6, Foxn4,* and *Otx2*, respectively. The expression of *Mybl1, Foxn4,* and *Otx2* were significantly enriched in Olig2+ RPCs (Fisher's exact test Odds Ratios [OR]: 11.8 [$p<0.0001$], 7.1 [$p<0.0001$], and 8.5 [$p<0.0001$], respectively), whereas *Pax6* (OR: 0.96, $p=0.31$) was not, perhaps reflecting a role as a repressor. These results support the hypothesis that these factors are functionally intertwined in the regulation of *Olig2* expression.

To examine whether the suggested Olig2 regulatory TF's bind to the Olig2-NR2 CRM, a CUT&RUN analysis was carried out. A significant enrichment of Mybl1, Pax6, and Foxn4 binding, with peaks that aligned with the NR2 region (*Figure 8H*), provided evidence for their direct regulatory roles via the Olig2-NR2 CRM. Together, these findings suggest that the Olig2-NR2 CRM is a target for multiple retinal developmental regulators, potentially mediating Olig2 expression in neurogenic RPCs.

Using de novo motif discovery analysis on Olig2-NR1, 20 motifs were identified (*Figure 8—figure supplement 1A*), some of which correspond to binding sites for several retinal TFs, including Sox4, Sox11, Lhx2, Dlx2, Isl1, Foxp1, Mybl1, and Otx2. These factors play critical roles in retinal development: Sox4 and Sox11 are active in neurogenic RPCs and regulate the generation and survival of RGCs (*Jiang et al., 2013*); Lhx2 is suggested to influence the progression of RPC competence states (*Gordon et al., 2013*); Dlx2 is required for terminal RGC differentiation (*de Melo et al., 2005*; *Zhang et al., 2017*); Isl1 is essential in the gene regulatory network governing RGC development (*Mu et al., 2008*); Foxp1 has been implicated in regulating the competence of RPCs to generate early-born retinal cell types (*Zhang et al., 2023*); and Mybl1 and Otx2 are known to influence neural progenitor cell cycle regulation (*Kaufman et al., 2019*), and photoreceptor and horizontal cell development (*Nishida et al., 2003*; *Wang et al., 2014*), respectively.

In the d-MPRA analysis, specific motifs in Olig2-NR1 aligned with these predicted TF binding sites. For instance, Motifs 13 and 16 matched shared binding sites for Sox4 and Sox11 (*Figure 8—figure supplement 1B*); scRNA-seq co-expression data of RPCs (*Figure 8—figure supplement 1G*) revealed that 74.1% of *Olig2*-expressing neurogenic RPCs co-expressed *Sox4* (enrichment OR: 5.3 [$p<0.0001$]) and 89.3% co-expressed *Sox11* (enrichment OR: 3.1 [$p<0.0001$]). In contrast, Motif 11 contained a shared binding site for both Lhx2 and Dlx2 (*Figure 8—figure supplement 1C*); among *Olig2*+ neurogenic RPCs, 29.6% co-expressed *Lhx2* (enrichment OR: 1.0, $p=0.94$) and 21.1% co-expressed *Dlx2* (enrichment OR: 3.7 [$p<0.0001$]). Motif 8 matched an Isl1 binding site (*Figure 8—figure supplement 1D*) and was associated with only 5.0% co-expression in *Olig2*+ RPCs (*Figure 8—figure supplement 1G*) (enrichment OR: 0.33 [$p<0.0001$]), implying it might suppress *Olig2* expression. Motif 18 contained a Foxp1 binding site (*Figure 8—figure supplement 1E*), with 40.9% of *Olig2*+ neurogenic RPCs co-expressing *Foxp1* (enrichment OR: 0.81 [$p<0.0001$]), as shown in *Figure 8—figure supplement 1G*. Additionally, binding sites for Mybl1 and Otx2 were identified in Motifs 2 and 12, respectively (*Figure 8—figure supplement 1F*); scRNA-seq data (*Figure 8G*) indicate that these TFs may also contribute to the regulation of Olig2 expression via Olig2-NR1. Taken together, and considering the alignment of motifs with d-MPRA peaks and troughs as well as the quantified co-expression data from *Figure 8G*, *Figure 8—figure supplement 1G*, these results suggest that Sox4, Sox11, Dlx2, and Isl1 may predominantly function to repress Olig2 expression. Lhx2, Foxp1, Mybl1, and Otx2 are likely to act as activators through the Olig2-NR1 CRM.

The de novo motif discovery analysis of Olig2-NR3 identified 19 motifs (*Figure 8—figure supplement 1H*), several of which aligned with peaks or troughs in the d-MPRA plot. Among these, four motifs containing known TF binding sites active in the developing retina were associated with

troughs, suggesting potential enhancer activity. Motif 13 matched a binding site for Bhlhb5 (encoded by *Bhlhe22*) (*Figure 8—figure supplement 1I*), a TF involved in retinal interneuron subtype specification (*Feng et al., 2006*; *Huang et al., 2014*); scRNA-seq co-expression data (*Figure 8—figure supplement 1L*) revealed that among *Olig2*+ RPCs, 21.6% co-expressed *Bhlhb5* (enrichment OR: 3.5 [*p*<0.0001]). Motifs 3 and 5 contained binding sites for Ngn2 (*Figure 8—figure supplement 1J*), a TF expressed in neurogenic RPCs during both early and late retinal development (*Hufnagel et al., 2010*; *Kowalchuk et al., 2018*; *Ma and Wang, 2006*); 81.4% of *Olig2*-expressing RPCs co-expressed *Ngn2* (*Figure 8—figure supplement 1L*) (enrichment OR: 15.4 [*p*<0.0001]). Motif 15 matched a binding site for Lhx9 (*Figure 8—figure supplement 1K*), a TF expressed in early retinal development that is necessary for interneuron subtype specification (*Balasubramanian et al., 2014*; *Balasubramanian et al., 2018*). scRNA-seq analysis (*Figure 8—figure supplement 1L*) established that 7.7% of RPCs expressing *Olig2* co-expressed *Lhx9* (enrichment OR: 3.3 [*p*<0.0001]). The combined scRNA-seq analysis, d-MPRA, and motif discovery analysis confirmed that there are likely Bhlhb5, Ngn2, and Lhx9 binding sites in relevant regions of Olig2-NR3, and that they exhibit some degree of co-expression with *Olig2*, supporting the hypothesis that these TFs may regulate Olig2 expression via the Olig2-NR3 CRM.

The d-MPRA and motif analyses reveal a complex transcriptional network potentially regulating Olig2 expression, with some of the same TFs as candidate regulators across Olig2-NR1, NR2, and NR3 CRMs. Olig2-NR2 emerged as potentially the most important regulatory hub, where motif discovery, scRNA-seq enrichment, and CUT&RUN validation confirmed direct binding of Mybl1, Pax6, and Foxn4. The d-MPRA approach provided a robust, high-resolution method for nominating functionally relevant TF binding sites, with motif-associated peaks and troughs aligning with known retinal regulators. The identification of additional candidates in Olig2-NR1 and NR3 provides candidates to expand our understanding of the gene regulatory network governing Olig2 expression, setting the stage for future in vivo studies to dissect their functional roles in retinal development.

## LS-MPRAs for Neurogenin-2 in the developing mouse retina

To further test the LS-MPRA in identifying CRMs regulating transiently expressed genes of regulatory importance in the retina, *Ngn2* was analyzed. Ngn2, a proneural bHLH TF, is expressed in a subset of RPCs, those that will produce at least one postmitotic cell (*Hufnagel et al., 2010*). Though known regulatory regions for Ngn2 have been elucidated in the ventral neural tube, where it is also expressed, their activities in the developing retina were unknown (*Henke et al., 2009*; *Nakazaki et al., 2008*; *Scardigli et al., 2003*; *Scardigli et al., 2001*; *Simmons et al., 2001*). The Ngn2 LS-MPRA library (library characteristics in *Source data 1*) was delivered to the E14 mouse retina and processed through the LS-MPRA pipeline, as described for Olig2. The aligned barcode enrichment plot revealed distinct candidates for CRMs (*Figure 9A*). Among these, Ngn2-CRM3 was highly conserved across vertebrate and placental mammal species and corresponded to an open chromatin region located –136 bp to +150 bp relative to the *Ngn2* TSS. It is also located in a 4.4 kb fragment with regulatory activity in the embryonic mouse spinal cord (*Simmons et al., 2001*). To investigate the functionality of Ngn2-CRM3 and three additional CRM candidates identified in the LS-MPRA plot (Ngn2-CRM1, CRM2, and CRM4), each fragment was cloned into a plasmid containing a TATAA minimal promoter and EGFP reporter. These constructs were electroporated into E14 retinas and incubated ex vivo for 24 hr prior to analysis.

FISH for *Ngn2* and *GFP* RNA, alongside detection of intrinsic GFP fluorescence, in transverse retinal sections (*Figure 9B*), demonstrated that Ngn2-CRM2 and Ngn2-CRM3 fragments consistently drove GFP expression in Ngn2+ cells. Co-localization analysis revealed that a substantial majority of *Ngn2* RNA+ electroporated cells expressed *GFP* RNA and/or protein when GFP expression was driven by these two CRMs (*Figure 9C*, left column; *Supplementary file 1e*). Approximately 57% of *Ngn2* RNA+ cells co-expressed *GFP* RNA when driven by Ngn2-CRM1, suggesting that this fragment may also function as a CRM, albeit less specifically for Ngn2. In contrast, GFP expression driven by Ngn2-CRM4 was sparse, and even fewer cells co-expressed *Ngn2* RNA, indicating that the LS-MPRA peak associated with this region does not correspond to a functional Ngn2 CRM in these conditions.

The proportion of *GFP* RNA+ cells co-expressing *Ngn2* RNA when GFP expression was driven by Ngn2-CRM2 or CRM3 (approximately 21% and 38%, respectively; *Figure 9C*, middle-left column; *Supplementary file 1e*) was lower compared to the corresponding co-expression analysis for Olig2

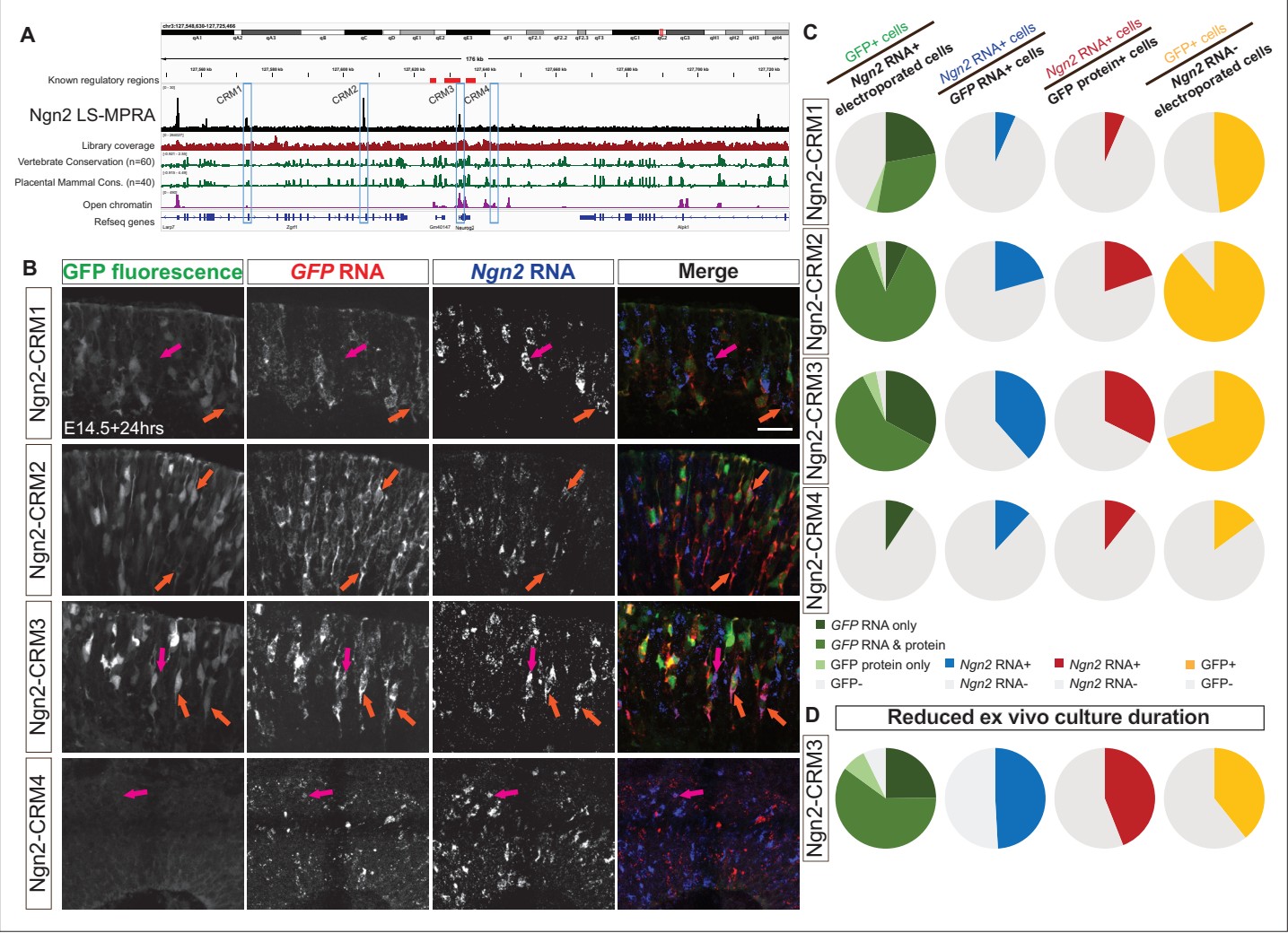

**Figure 9.** Ngn2 LS-MPRA to identify *cis*-regulatory module (CRM) candidates active in mouse retinal cells expressing *Ngn2*. (**A**) Ngn2 locus-specific massively parallel reporter assay (LS-MPRA) barcode enrichment plot aligned with a genome browser track, annotated with (i) known *Ngn2* regulatory regions, (ii) coverage of the barcode-fragment association library, (iii) log$_2$ base conservation across 60 vertebrate or 40 placental mammal species, (iv) regions of open chromatin in E14 mouse retina, (v) gene models, and (vi) CRM candidate regions 1–4 (blue boxes). (**B**) Representative transverse sections of E14 retinas incubated for 24 hr in vitro, showing localization of *Ngn2* RNA with *GFP* RNA only (magenta arrow) or both intrinsic GFP and *GFP* RNA (orange arrows) in retinas electroporated with plasmids containing one of the CRM1-4 regions. (**C**) Pie charts showing the percentage of *Ngn2* RNA+ cells expressing GFP (column 1), *GFP* RNA+ cells expressing *Ngn2* RNA (column 2), GFP protein+ cells expressing *Ngn2* RNA (column 3), and *Ngn2* RNA- cells expressing GFP (column 4) in retinas electroporated at E14 with plasmids containing CRM1-4 regions driving GFP. (**D**) Co-localization analysis of Ngn2-CRM3 plasmid and GFP following 16 hr incubation in vitro. Scale bar: 20 μm.

CRMs (*Figure 6*). Similar to Olig2 CRMs, however, the proportion of *GFP* RNA+ cells driven by any Ngn2-CRM candidate region was consistently greater than that observed when the analysis was restricted to GFP protein+ cells (*Figure 9C*, middle-right column; *Supplementary file 1e*). These findings suggest that the rapid degradation of transient Ngn2 protein (~30 min half-life, *McDowell et al., 2014*; *Vosper et al., 2007*) and, likely, *Ngn2* transcript, limits the overlap between Ngn2 and GFP expression.

To address this limitation, the ex vivo incubation period was reduced from 24 to 16 hr and focused on Ngn2-CRM3, the most robust CRM candidate. Shortening the incubation period resulted in an increase in the proportion of *GFP* RNA+ cells co-expressing *Ngn2* RNA (38% vs 49%; *Figure 9C and D*, right column; *Supplementary file 1e*). Further evidence for this is provided by the concurrent decrease in the proportion of *GFP* RNA+ cells within the population of *Ngn2* RNA-negative cells when the incubation period was shortened (69% vs 39%; *Figure 9C and D*, right column; *Supplementary*

*file 1e*). This improvement supports the hypothesis that Ngn2's transient expression window and rapid turnover obscured co-expression of *GFP* and *Ngn2* RNA.

## LS-MPRAs in chick retina and spinal cord

To determine if LS-MPRAs can be used to identify CRMs across species and tissues, regulatory elements controlling chick OLIG2 (cOLIG2) expression were carried out. Expression in the chick embryonic day 5 (E5) retina, a developmental stage equivalent to the E14 mouse retina, as well as in the chick spinal cord at two time points: E2 and E4, were examined with LS-MPRA. Olig2 is expressed in spinal cord progenitor cells that give rise to motor neurons and oligodendrocytes (*Lu et al., 2000*; *Zhou et al., 2000*). Electroporating the cOLIG2 LS-MPRA at E2 should capture the transcriptional activity of motor neuron progenitor cells (*Hollyday and Hamburger, 1977*), while delivery at E4 should capture the activity within oligodendrocyte precursor cells (*Soula et al., 2001*). The cOLIG2 LS-MPRA library (library characteristics in *Source data 1*) was electroporated into E5 chick retina explants and incubated ex vivo for 24 hr or electroporated into the E2 or E4 spinal cord and incubated in ovo for 24 hr or 48 hr, followed by LS-MPRA analysis.

Barcode enrichment plots from these experiments, aligned to the galGal6 chick reference genome, revealed distinct candidate CRM regions in both spinal cord and retina (*Figure 10A*). One region, spanning −36.729 kb to −36.428 kb relative to the chick *OLIG2* TSS, is highly conserved across vertebrates and orthologous to the Olig2-NR1 CRM identified in the mouse retina (97.3% sequence conservation shared across a 225 bp region). Interestingly, this CRM candidate (cOLIG2-CRM1) was active in RPCs and in the E4 spinal cord but showed no detectable activity in electroporated E2 spinal cord. In addition to CRM1, two proximal CRM candidates—CRM2 (−264 bp to −53 bp) and CRM3 (+8.948 kb to +9.036 kb)—were identified in chick and assessed for their functional relevance to cOLIG2 expression.

To determine whether cOLIG2-CRM1, CRM2, and CRM3 function as CRMs, each fragment was cloned into a plasmid containing the chick *OLIG2* minimal promoter and an EGFP reporter. These constructs were electroporated into E5 chick retinas and incubated ex vivo for 24 hr prior to analysis. FISH for chick *OLIG2* and *GFP* RNA, alongside intrinsic GFP protein expression in transverse retinal sections (*Figure 10B*), demonstrated that cOLIG2-CRM1 and cOLIG2-CRM2 consistently drove GFP expression in cOLIG2+ cells. Co-localization analysis revealed that the majority of electroporated chick *OLIG2* RNA+ cells expressed *GFP* RNA and/or protein when GFP expression was driven by these two CRMs (*Figure 10C*, left column; *Supplementary file 1f*). In contrast, GFP expression driven by cOLIG2-CRM3 was nearly absent, indicating that the LS-MPRA peak associated with this region does not correspond to a functional cOLIG2 CRM in these conditions. Notably, the CRM3 peak overlaps with an AscI recognition site, a sequence rarely found in vertebrate genomes, which was used in the LS-MPRA design to facilitate barcode–fragment association (*Figure 1C*). This likely led to an under-representation of this region in the association library, thereby generating a false positive due to normalization using the association library flanking the AscI site in the BAC.

The proportion of *GFP* RNA+ cells co-expressing chick *OLIG2* RNA when GFP was driven by cOLIG2-CRM1 or CRM2 (*Figure 10C*, middle-left column; *Supplementary file 1f*) was nearly identical to the proportion of GFP protein+ cells (*Figure 10C*, middle-right column; *Supplementary file 1f*). This suggests a more synchronized developmental window of chick *OLIG2* RNA and protein expression in chick compared to mouse (*Figure 6A*). However, as observed in the mouse retina, a high proportion of chick *OLIG2*- electroporated cells expressed *GFP* RNA (*Figure 10C*, right column; *Supplementary file 1f*), underscoring the challenges of detecting CRMs for transiently expressed transcription factors.

To further assess CRM activity in spinal cord progenitor cells, the cOLIG2-CRM constructs were electroporated into the ventral horn of E2 chick spinal cords (*Figure 10—figure supplement 1A*) and incubated in ovo for 24 hr. FISH for chick *OLIG2* and *GFP* RNA in transverse spinal cord sections (*Figure 10—figure supplement 1B*) revealed that, similar to the chick retina, cOLIG2-CRM1 and cOLIG2-CRM2 drove *GFP* RNA expression in chick *OLIG2*+ cells. However, both regions also exhibited GFP expression in chick *OLIG2*- cells, suggesting either that their CRM activity is not restricted to cOLIG2-expressing progenitors in the spinal cord, or that low levels of chick *OLIG2* RNA are undetectable in some GFP+ cells. No GFP expression was observed from cOLIG2-CRM3, further supporting the conclusion that this fragment does not function as a cOLIG2 CRM.

These findings highlight the ability of LS-MPRAs to identify orthologous CRMs that may regulate gene expression across species and developmental contexts. The conservation of cOLIG2-CRM1 with

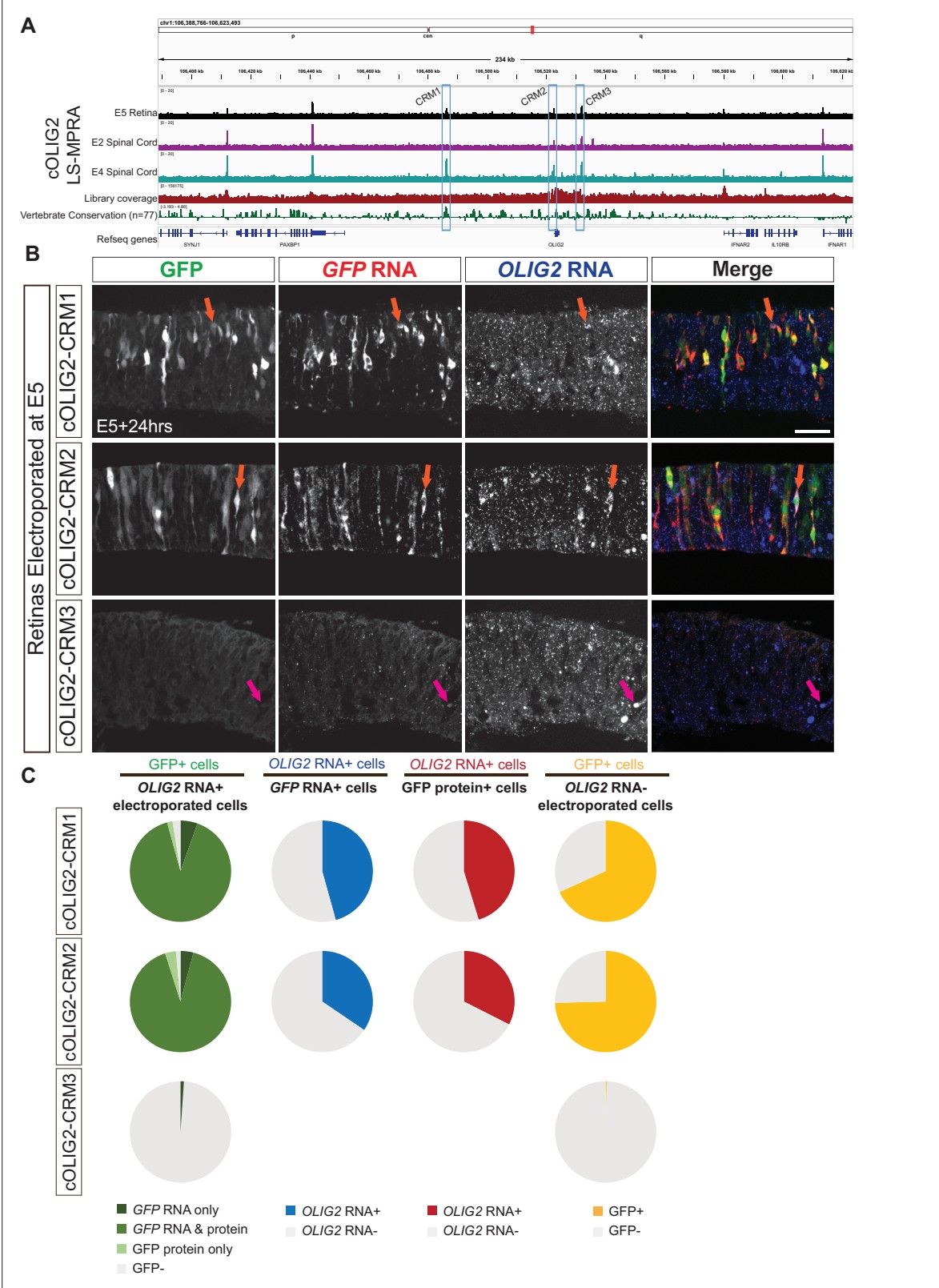

**Figure 10.** OLIG2 locus-specific massively parallel reporter assay (LS-MPRA) to identify cis-regulatory module (CRM) candidates in chick embryos. (**A**) OLIG2 LS-MPRA barcode enrichment plot following electroporation into E5 chick retinal explants, and E2 spinal cords and E4 spinal cords in ovo, aligned with a genome browser track and annotated with (i) coverage of the barcode-fragment association library, (ii) log₂ base conservation across 77 vertebrate species, (iii) gene models, and (iv) CRM candidate regions 1–3 (blue boxes). (**B**) Representative transverse sections of E5 chick retinas

*Figure 10 continued on next page*

*Figure 10 continued*
incubated for 24 hr in vitro, showing localization of chick *OLIG2* RNA with *GFP* RNA only (magenta arrow) or both intrinsic GFP fluorescence and GFP RNA (orange arrows) driven by plasmids containing one of the CRM1-3 regions. (**C**) Pie charts showing the percentage of *OLIG2* RNA+ cells expressing GFP (column 1), *GFP* RNA+ cells expressing *OLIG2* RNA (column 2), GFP protein+ cells expressing *OLIG2* RNA (column 3), and *OLIG2* RNA- cells expressing GFP (column 4) in retinas electroporated at E5 with plasmids containing CRM1-3 regions driving GFP and visualized by FISH. Scale bar: 20 µm.

The online version of this article includes the following figure supplement(s) for figure 10:

**Figure supplement 1.** Activity of OLIG2 *cis*-regulatory modules (CRMs) in OLIG2+ cells within embryonic chick spinal cords.

the mouse Olig2-NR1 CRM suggests a shared regulatory mechanism for Olig2 in vertebrate retinal and oligodendrocyte precursor cells, while the species- and tissue-specific activities of other CRMs underscore the complexity of gene regulation across development and across species.

## Olig2 CRMs active in embryonic RPCs are active in postnatal RPCs that produce rod photoreceptors

Thus far, we have primarily focused on the regulatory elements controlling Olig2 expression in RPCs destined to produce cones and horizontal cells. Later in development, after all cone and horizontal cells have been generated, postnatal neurogenic RPCs express Olig2 and produce primarily rod photoreceptors (*Hafler et al., 2012*). An interesting question concerns whether these two distinct Olig2+RPC populations share the same regulatory mechanisms governing Olig2 expression. To address this, we electroporated plasmids containing Olig2-NR1, NR2, NR3, or all three CRMs into P0 retinas and incubated them in vivo for 24 hr before analysis.

Immunolabeling for GFP and Olig2 in transverse P1 retinal sections (*Figure 11A*) and co-localization analysis revealed that the vast majority of Olig2+ electroporated cells expressed GFP when driven by NR1, NR2, NR3, or the combined NR1-3 composite (*Figure 11B*, left column; *Supplementary file 1g*). These findings indicate that NR1, NR2, and NR3 function as CRMs in RPCs that primarily produce rods in the postnatal retina. Notably, at this postnatal stage, most GFP+ electroporated cells also expressed Olig2 protein (*Figure 11B*, middle column; *Supplementary file 1g*), suggesting that the temporal activation of these CRMs is more closely aligned with Olig2 expression compared to their earlier activity in embryonic RPCs.

Since the majority of RPCs after P0 express Olig2, only a small number of Olig2- electroporated cells were present in the analysis. Despite their limited representation, most of these cells still exhibited GFP expression (*Figure 11B*, right column; *Supplementary file 1g*). Similar to the embryonic retina, this could reflect a temporal mismatch in the co-expression of Olig2 and GFP or indicate a subset of Olig2- cells with CRM activity.

Overall, these findings suggest that the same regulatory elements govern Olig2 expression in embryonic RPCs and the neonatal RPCs, as NR1, NR2, and NR3 serve as CRMs in both populations. The more synchronized co-expression of Olig2 and GFP in postnatal RPCs further implies that Olig2 regulation in late RPCs is more tightly coupled to its downstream gene expression programs compared to its earlier, less synchronous expression in early RPCs. This may be a reflection of slower developmental processes, as at least the cell cycle is longer in the later stages of retinal development (*Alexiades and Cepko, 1996*; *Young, 1985*).

## Discussion

In this study, we introduce novel LS-MPRA and d-MPRA approaches to search the *cis*-regulatory landscape of key developmental genes. By leveraging BAC-based libraries to survey large genomic regions and employing a degenerate MPRA strategy for nucleotide-resolution mapping, these methods overcome some of the limitations inherent to synthetic MPRAs. This approach enables the unbiased identification of candidate CRMs while precisely delineating potential TF binding sites, thereby nominating both activators and repressors that regulate expression of a gene. These methods also offer a rapid and simple route to identify CRMs that can be deployed in applications where driving gene expression in a specific context is required. The scale, expense, and relatively simple molecular methods enable CRM discovery by any laboratory with basic molecular biology skills.

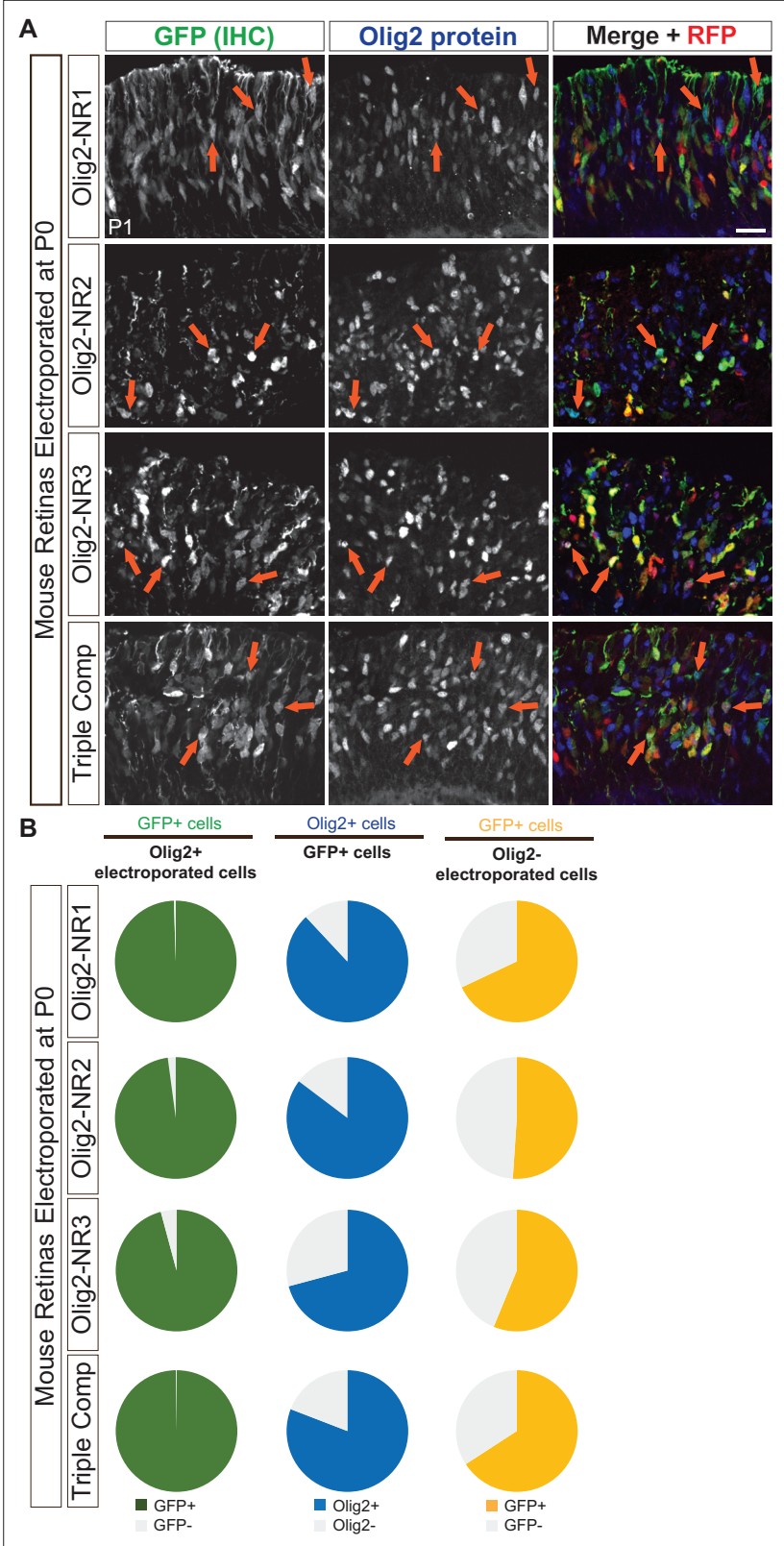

**Figure 11.** Activity of Olig2 *cis*-regulatory module (CRM) candidates in postnatal retinal cells expressing Olig2. (**A**) Representative transverse sections of postnatal retinas electroporated at P0 and incubated in vivo for 24 hr, stained with antibodies against Olig2 and GFP to identify co-localized expression (orange arrows). GFP expression was driven by plasmids containing one or more Notch-inhibitor responsive regions (NR1-3). (**B**) Analysis of GFP and

*Figure 11 continued on next page*

*Figure 11 continued*

Olig2 co-localization, shown with pie charts depicting the percent of Olig2+ electroporated cells expressing GFP (column 1), the percent of GFP+ cells expressing Olig2 (column 2), and the percent of Olig2- electroporated cells expressing GFP in retinas electroporated at P0 with plasmids containing one or more Notch-inhibitor responsive regions driving GFP. Scale bar: 20 μm.

## LS-MPRA for stably expressed genes

The LS-MPRA was benchmarked using genes with known CRMs for stably expressed, cell-type-specific genes. The experiments were carried out in vivo or in intact retinal explants, for discovery of CRMs relevant to authentic gene regulation in specific cell types. In postnatal retinas, CRMs were identified for the Rho PPR as well as some of the known CRMs for Grm6 and Vsx2. Interestingly, while the Rho LS-MPRA consistently detected a clear CRM peak at the PPR in both in vivo and ex vivo conditions, the previously identified RER and CBR Rho CRMs were seen only in ex vivo retinas. Of note, deletions of RER and CBR from the mouse genome showed that these two CRMs are dispensable for Rho expression, while the PPR is required (*Sun et al., 2023*). The LS-MPRA in ex vivo retinas displayed a smaller signal-to-noise ratio around the *Rho* locus, particularly for the RER and CBR regions, but not around other loci on this BAC, perhaps reflecting differences in the requirements for a specific tissue environment or time course for proper *Rho* expression. Another requirement for RER and CBR activity might be the use of the Rho PPR. Both the RER and CBR were shown to enhance expression in constructs that used the Rho promoter (*Corbo et al., 2010*), but neither had been tested using a heterologous promoter, as was used here. In the Vsx2 LS-MPRA, 2 of the 3 known CRMs were readily identified, with those 2 CRMs located quite some distance from the TSS, at –37 and –17 kb. It is unclear why the previously identified promoter-proximal CRM was not captured. One possible explanation is that the relatively short fragment lengths used to construct the libraries may not encompass the broader genomic context required for activation of this proximal CRM. Another potential explanation is again the choice of the promoter, as the Vsx2 promoter was used to identify the promoter-proximal CRM in the previous study. The LS-MPRA for two other bipolar cell genes, Grm6 and Cabp5, also showed some positive and some negative results. Grm6 is expressed in the majority of retinal bipolar cells, and our previously described CRM for this gene, at –8 kb, was identified by the LS-MPRA in both ex vivo and in vivo conditions. In contrast, the LS-MPRA failed to capture the known Cabp5 enhancer. It is not clear why this CRM was not identified here. Deeper sequencing of the cDNA libraries and/or using only a single BAC library for Cabp5, rather than the multiplexed format with 4 BACs used here, likely would increase the sensitivity. Sensitivity should also be increased by isolating electroporated cells of interest and/or pharmacologically or genetically increasing gene expression, as we did for Olig2. Finally, the choice of promoter may again play a role as the SV40 promoter was used in the earlier study for identification of the Cabp5 CRM.

## LS-MPRA for transiently expressed genes

A focus of our laboratory is the mechanisms of cell fate determination in the retina. Our early study of RNA expression in RPCs showed that some TFs were expressed in subsets of RPCs, at one time and across the period of neurogenesis (*Trimarchi et al., 2008*). We are interested in whether heterogeneity in RPC gene expression accounts for differences in the types of cells produced. Olig2 was identified as expressed in subsets of RPCs. Using a retroviral lineage tracing method to mark clones derived from Olig2 RPCs, we found that they divide only once or twice and produce a limited repertoire of cell types: cones and horizontal cells in the embryonic period and rods and amacrine cells in the postnatal period. Interestingly, intermingled with the embryonic Olig2+ RPCs are Olig- RPCs, which produce RGCs, the output neurons of the retina. To address how these Olig2+ RPCs might differ from Olig2- RPCs, our goal is to leverage Olig2 regulation. We thus developed the MPRA methods to identify candidates for its regulation and to see if these regulators might vary over time, in correlation with the different daughter cell types produced at different times.

The LS-MPRA for Olig2 did indeed identify CRMs that lead to potential regulatory sites as well as their cognate TFs. However, it was difficult to validate the specificity of these CRMs due to the short half-life of endogenous Olig2 RNA and protein. Nonetheless, when tested as single or combined CRMs in individual plasmid constructs, there was activity in a high proportion of Olig2+ cells. CRM activity in Olig2- cells may reflect a mix of cell states, such as those that have recently downregulated

Olig2 but retain GFP, RPCs initiating CRM-driven GFP expression prior to detectable *Olig2* levels, or cells in which the CRM functions independently of Olig2 expression. In addition, several technical and biological considerations may contribute to the apparent disconnect between reporter activity and endogenous Olig2 or Ngn2 expression. Introducing putative CRMs on plasmids removes them from their native genomic context, which may bypass normal chromatin accessibility constraints or local repressive influences and allow activity in cells where the endogenous gene is not expressed. These CRMs may also regulate genes other than Olig2 or Ngn2, reflecting broader or context-dependent roles. Finally, both Olig2 and Ngn2 are likely regulated by many more regulatory elements than the few tested here, so their endogenous expression may not depend solely on these specific CRMs. Despite GFP expression in Olig2- cells, several other lines of evidence increase confidence that Olig2-NR1, NR2, and NR3 are excellent candidates for specific Olig2 regulation. First, in the LS-MPRA, the same CRMs were found across three replicates, even though the signal to noise near the Olig2 locus was modest. The modest ratio is likely due to the fact that a modest or small percentage of RPCs express Olig2, reported to be ~14% in a previous study (*Brzezinski et al., 2011*), or 2.3% in a more recent study (*Clark et al., 2019*). Additionally, and perhaps most importantly, it is expressed only briefly. To boost the signal, to potentially gain confidence in the candidate CRMs identified in the LS-MPRA, we were able to implement a drug manipulation. Notch inhibition greatly increases the production of cones (*Jadhav et al., 2006*; *Yaron et al., 2006*), and here we show that this same inhibition leads to greater Olig2 expression, with a peak at 12 hr post inhibition. Importantly, the activity of Olig2-NR1, NR2, and NR3 was upregulated by Notch inhibition with the same magnitude and kinetics as seen for the endogenous Olig2 locus. This experiment shows that the LS-MPRA can be used not only to nominate CRMs for a gene, but to nominate CRMs that respond to perturbation, which might be particularly useful when the experiment is run in the full library format (i.e. all BAC fragments are assayed for a response). Additional evidence that Olig2-NR1, NR2, and NR3 are valid came from the d-MPRA and the follow-on experiments. The d-MPRA showed some of the same potential TF binding sites in more than one CRM, and several of these TFs have been shown to be important in retinal development, including TFs that play a role in cone and horizontal cell development, e.g., Otx2. In keeping with this, scRNAseq data show significant co-expression of Olig2 with the predicted TFs. Finally, CUT&RUN, using retinal tissue, showed binding of the suggested TFs to the CRMs. These data collectively provide support for the legitimacy of Olig2-NR1, NR2, and NR3 as key regulatory regions.

The binding of Mybl1 to Olig2-NR2 and potentially to Olig2-NR1 suggests that it plays a role in cone and/or horizontal cell development, likely as an activator of Olig2 expression. Its role in cone development was suggested previously by a study which tracked RNA expression in cultures induced to produce cones (*Kaufman et al., 2019*). In contrast, Pax6—a master regulator essential for maintaining RPC multipotency and guiding cell fate decisions—has been shown to function as both an activator and a repressor (*Cvekl and Callaerts, 2017*; *Farhy et al., 2013*; *Marquardt et al., 2001*; *Oron-Karni et al., 2008*; *Remez et al., 2017*); its binding to Olig2-NR2 may keep Olig2 off in multipotent RPCs, and/or keep Olig2 off in RPCs that produce RGCs. Meanwhile, Foxn4, which is critical for the specification of amacrine and horizontal cells (*Li et al., 2004*), may modulate Olig2 expression for control of the ratio of cone to horizontal cell fates. The overlapping motifs for Pax6 and Foxn4 in Olig2-NR2 may represent a region that balances activation and repression for modulation of early cell fate decisions.

Our results also reveal that different CRMs may be differentially engaged during retinal development. For instance, while Olig2-NR1 appears to integrate inputs from TFs, such as Sox4, Sox11, Lhx2, Foxp1, Mybl1, and Otx2, Olig2-NR3 harbors binding sites for TFs, including Bhlhb5, Ngn2, and Lhx9. As Sox4 and Sox11 binding sites might represent repressor sites, it is curious that they are co-expressed in 74% and 89% of Olig2+ cells. As they, as well as Lhx2 and Dlx2, play a role in RGC development, these expression patterns and roles in Olig2 regulation might represent a dynamic and complex gene regulatory network governing the production of the early born cell types. Consistent with this idea, and with the role of Bhlhb5 in later-born cell types, its binding site within the Olig2-NR3 may be engaged later in retinal development, considering a subset of bipolar cells exhibit a history of Olig2 expression (*Hafler et al., 2012*), and some mature bipolar cells maintain Olig2 expression. This nuanced and complex regulation underscores the broader utility of LS-MPRA and d-MPRA for discerning CRMs and TFs that are either cell type-specific or shared across distinct developmental competence states. Future studies employing targeted perturbations, such as CRISPR/Cas9-mediated

disruption of specific TF binding sites, will be essential to validate these CRMs and establish the roles that they play in gene regulation.

## Considerations for use of LS-MPRA and d-MPRA

There are several issues to consider when choosing a method for CRM identification. Data describing open/active chromatin, conserved genomic sequences, and RNA expression, are increasingly comprehensive but do not cover all cell types and non-model organisms. However, with the advent of single-cell multi-omics, one can anticipate that these gaps will eventually be filled over the period of the next few years. When such data are available and predict very few CRMs, one can simply PCR a potential CRM from genomic DNA and test individually. In other cases, these types of data nominate a high number of potential CRMs. Producing many individual PCR products, cloning them into expression vectors, sequencing, introducing into cells, and assaying the results from single constructs, are much more time-consuming and expensive than LS-MPRAs. For example, we have identified potential CRMs for Otx2. Using predictions from open chromatin, eRNAs, and DNA conservation, >20 CRMs were nominated (*Chan et al., 2020*; *Emerson and Cepko, 2011*; *Lonfat et al., 2021*). Assessing each potential CRM and its relevant TF binding sites spanned more than several years. In addition to this level of effort, testing each potential CRM in isolation precludes a quick recognition of the relative strength of each CRM. Speed and expense can also be addressed using LS-MPRA by multiplexing libraries from several genes. Finally, when one needs a relatively small CRM, e.g., for expression in a viral vector, one can choose the fragment size during the BAC library preparation step. CRMs predicted by open chromatin typically do not provide this option.

One critical requirement for identification of CRMs using any method is introduction of constructs into cells. Tissue culture cells offer easy access. However, they often do not recapitulate the endogenous in vivo context and are often a single, irrelevant cell type. Even when multiple cell types are present, as in organoids, the correct environment may not be present. This may be exemplified by the difference between CRM identification for the *Rho* gene in the in vivo versus ex vivo retinal explants. The LS-MPRA from ex vivo retinas showed noisier CRM signals and failed to show as good a signal-to-noise ratio for the required Rho CRM, the PPR. This did not seem to hold true for the other in vivo versus ex vivo comparisons for Vsx2 and Grm6. If one can electroporate in vivo, as we do here, the LS-MPRA likely will provide the most reliable results. If electroporation into the cell type of interest is not possible, one can introduce libraries using viral vectors, as was done with an AAV library for identification of CRMs for specific cell types of the central nervous system (*Hrvatin et al., 2019*). Selected genomic regions were chosen bioinformatically, which may introduce a bias, and each predicted element was individually generated by PCR, which is time-consuming and limits library size. Bias may also be introduced by the choice of the promoter or other elements in the expression construct. We chose a simple TATAA promoter to reduce background activity, but as genomic sequences can influence each other, we may have missed some CRMs due to this choice. In addition, while the BAC-based approach enables the survey of large genomic regions, it cannot capture all potential regulatory inputs, and it remains possible that some distal regulatory sequences lie outside of the BACs used here.

When using a CRM as a reagent to gain genetic access to specific cell types, further analysis of its architecture is often unnecessary. However, for those aiming to understand the regulation of a gene of interest, the d-MPRA offers a powerful approach to rapidly pinpoint individual nucleotides that positively or negatively influence CRM activity. Traditional methods, such as introducing single-base mutations or deletions and testing each construct individually, are time-consuming and often yield qualitative, rather than quantitative, results. Repressor sites, in particular, are rarely identified this way. With the growing availability of TF binding site databases, d-MPRA can efficiently highlight key nucleotides and nominate candidate TFs involved in CRM regulation. When multiple CRMs are available, comparing predicted TF binding sites across them can further strengthen TF candidacy. Examining RNA expression data provides additional support. To validate these predictions, TF binding assays performed in vivo or in vitro can confirm binding to the endogenous locus. It would also be valuable to integrate LS-MPRA and d-MPRA with analyses of DNA methylation, particularly in regions near genes of interest across stages of retinal development, building on prior studies of methylation in adult tissues associated with disease (*Advani et al., 2024*). Importantly, LS-MPRA and d-MPRA methods are adaptable across species, as demonstrated here in both chick and mouse, and can be applied to different regions of the developing embryo.

# Methods

**Key resources table**

| Reagent type (species) or resource | Designation | Source or reference | Identifiers | Additional information |
|---|---|---|---|---|
| Strain, strain background (*Mus musculus*) | CD-1 | Charles River Laboratories | RRID:IMSR_CRL:022 | |
| Strain, strain background (*Gallus gallus*) | SPF Eggs | AVS Bio | Material Number: 10100330 | |
| Strain, strain background (*Escherichia coli*) | DH10β Cells | Thermo Fisher | cat. #18290015 | Electrocompetent cells |
| Gene (*Mus musculus*) | *Rho* | NCBI GenBank | NM_145383.2 | |
| Recombinant DNA reagent | *Rho* BAC clone | BACPAC Resources | RP23-219M6 | |
| Gene (*Mus musculus*) | *Grm6* | NCBI GenBank | NM_173372.2 | |
| Recombinant DNA reagent | *Grm6* BAC clone | BACPAC Resources | RP23-417M10 | |
| Gene (*Mus musculus*) | *Vsx2* | NCBI GenBank | NM_001301427.1 | |
| Recombinant DNA reagent | *Vsx2* BAC clone | BACPAC Resources | RP23-127O21 | |
| Gene (*Mus musculus*) | *Cabp5* | NCBI GenBank | NM_013877.4 | |
| Recombinant DNA reagent | *Cabp5* BAC clone | BACPAC Resources | RP24-125H22 | |
| Gene (*Mus musculus*) | *Olig2* | NCBI GenBank | NM_016967.2 | |
| Recombinant DNA reagent | *Olig2* BAC clone | BACPAC Resources | CH29-613 | |
| Gene (*Mus musculus*) | *Ngn2* | NCBI GenBank | NM_009718.4 | Formal gene name: *Neurog2* |
| Recombinant DNA reagent | *Ngn2* BAC clone | BACPAC Resources | RP23-182M12 | |
| Gene (*Gal gallus*) | *OLIG2* | NCBI GenBank | NM_001031526.1 | |
| Recombinant DNA reagent | *OLIG2* BAC clone | BACPAC Resources | CH261-60J3 | |
| Recombinant DNA reagent | Stagintbc7 | This paper | | Deposited to Addgene; used to create LS-MPRA libraries |
| Recombinant DNA reagent | Statadual-WPRE | This paper | | Deposited to Addgene; used to create d-MPRA libraries |
| Chemical compound, drug | Fast Green FCF | Millipore Sigma | cat. #F7252 | |
| Chemical compound, drug | L-glutamine | Sigma Aldrich | cat. #G3126 | |
| Chemical compound, drug | Penicillin/ streptomycin | Invitrogen/Gibco | cat. #15140–122 | |
| Chemical compound, drug | Minimum Essential Medium | Millipore Sigma | cat. #51,412 C | |
| Chemical compound, drug | HBSS | Thermo Fisher Scientific | cat. #14-025-092 | |
| Chemical compound, drug | horse serum | Thermo Fisher Scientific | cat. #26050–088 | |
| Chemical compound, drug | HEPES | Invitrogen/Gibco | cat. #15630–080 | |
| Chemical compound, drug | buprenorphine | PAR Pharma | Custom formulation | |
| Chemical compound, drug | proparacaine hydrochloride | Bausch & Lomb | cat. #24208-730-06 | |
| Chemical compound, drug | LY411575 | Millipore Sigma | SML0506 | γ-secretase inhibitor |
| Chemical compound, drug | TRI Reagent | Millipore Sigma | cat. #T9424 | |
| Commercial assay or kit | Zymo Directzol Microprep Kit | Zymo Research | cat. #R2061 | |
| Commercial assay or kit | Dynabeads mRNA Purification Kit | Thermo Fisher Scientific | cat. #61006 | |
| Chemical compound, drug | papain solution | Worthington Biochemical | cat. #LS003126 | |

*Continued on next page*

*Continued*

| Reagent type (species) or resource | Designation | Source or reference | Identifiers | Additional information |
|---|---|---|---|---|
| Commercial assay or kit | HCR Buffers [v3.0] | Molecular Instruments | v3.0 reagents (discontinued) | |
| Chemical compound, drug | donkey serum | Jackson ImmunoResearch | cat. #017-000-121 | |
| Antibody | Pax6 (mouse monoclonal) | Developmental Studies Hybridoma Bank | RRID:AB_528427 | CUT&RUN primary antibody; 500 ng per sample |
| Antibody | Foxn4 (mouse monoclonal) | Santa Cruz Biotechnologies | cat# sc-377166 | CUT&RUN primary antibody; 500 ng per sample |
| Antibody | Mybl1 (rabbit polyclonal) | Millipore-Sigma | RRID:AB_1078540; cat# HPA008791 | CUT&RUN primary antibody; 500 ng per sample |
| Antibody | Olig2 (mouse monoclonal) | Millipore | RRID:AB_10807410; Cat# MABN50 | Primary antibody for IHC; dilution 1:500. |
| Antibody | GFP (chicken polyclonal) | Abcam | RRID:AB_300798; cat# ab13970 | Primary antibody for IHC; dilution 1:1000. |
| Other | DAPI | Invitrogen | RRID:AB_2629482; cat# D1306 | Nuclear counterstain; used at 1:1000 in IHC. |
| Commercial assay or kit | Zymo DNA Clean & Concentrator Kit | Zymo Research | cat. #D4013 | |
| Commercial assay or kit | Large construct DNA isolation kit | Qiagen | cat. #12462 | |
| Commercial assay or kit | Qubit dsDNA HS Assay Kit | Thermo Fisher Scientific | cats. #Q32851 and # Q32856 | |
| Commercial assay or kit | NEBNext Ultra II FS DNA Module kit | NEB | cat. #E7810S | |
| Chemical compound, drug | T4 DNA ligase buffer | NEB | cat. #B0202S | |
| Commercial assay or kit | Blunt/TA Ligase Master Mix | NEB | cat. #M0367 | |
| Chemical compound, drug | NEBNext Ultra II Q5 Master Mix | NEB | cat. #M0544 | |
| Chemical compound, drug | Agarose Dissolving Buffer | Zymo Research | cat. #D4001-1-100 | |
| Commercial assay or kit | Plasmid Maxi kit | Qiagen | cat. #12163 | |
| Commercial assay or kit | GeneMorph II Random Mutagenesis Kit | Agilent | cat. #200550 | |
| Chemical compound, drug | Bst 2.0 WarmStart Polymerase | NEB | cat. #M0538 | |
| Chemical compound, drug | Taq DNA polymerase | NEB | cat. #M0273 | |
| Commercial assay or kit | CUTANA ChIC/CUT&RUN Kit | EpiCypher | cat# 14–1048 | |
| Commercial assay or kit | NEBNext UltraII DNA Library Prep Kit | NEB | cat. #E7103 | |
| Commercial assay or kit | NEBNext Multiplex Oligos for Illumina Dual Index Sets | NEB | cats. #E7600S or #E7780S | |
| Commercial assay or kit | Quick Ligation reaction | NEB | cat. #M2200 | |
| Chemical compound, drug | Exonuclease | Lucigen | cat. #E3101K | |
| Commercial assay or kit | Protoscript II Kit | NEB | cat. #E6560 | |
| Commercial Assay or kit | Zymo Oligo Clean & Concentrator Kit | Zymo Research | cat. #D4060 | |
| Commercial assay or kit | LunaScript RT SuperMix Kit | NEB | cat. #E3010 | |
| Software/algorithm | Integrative Genomics Viewer | PMID:21221095 | RRID:SCR_011793 | |

*Continued*

| Reagent type (species) or resource | Designation | Source or reference | Identifiers | Additional information |
|---|---|---|---|---|
| Software/algorithm | Prism | GraphPad | RRID:SCR_002798 | |
| Software/algorithm | FIJI | ImageJ | RRID:SCR_002285 | |
| Software/algorithm | R | The R Foundation for Statistical Computing | RRID:SCR_001905 | |

## Resource availability

Raw image data, all original code, and unprocessed sequencing files are available on the Harvard Dataverse (DOI: doi:10.7910/DVN/TW0ZQL) (*Tulloch, 2025*). All processed LS-MPRA genome tracks are available to view on a Track Hub in the UCSC Genome Browser (https://github.com/cattapre/ALAS00, copy archived at *Catta-Preta, 2025*). Primer sequences are available in *Supplementary file 1a*. Any additional information is available from the lead contact upon request. Plasmids that are required for these methods will be deposited at Addgene.

## Experimental model and subject details

All experimental procedures were preapproved by the Harvard Medical Area Standing Committee on Animals (HMA IACUC). Timed and untimed pregnant CD-1 mice used for sequencing and histology were purchased from Charles River Laboratories and were housed in a climate-controlled pathogen-free facility. Mouse tissues were collected for sequencing experiments at E14, P3, or P10 and for histological experiments at E14 or P1 as noted within the text. Chick tissues were collected for sequencing experiments at E3, E5, and E6 and for histological experiments at E3 and E5, as noted within the text.

## Method details

### Electroporation and culture

For plasmid DNA injections, glass needles were created by pulling Wiretrol II capillaries (Drummond Scientific Company, cat. #5-000-2005) using a needle puller (Sutter Instrument, Model P-97). The glass needles were beveled on two edges with a microgrinder (Narishige, cat. #EG-401). Plasmid DNA with individual or composite enhancers, as well as d-MPRA libraries and most LS-MPRA libraries, were concentrated to 2.0 µg/µl along with control plasmids containing either bpCAG-mCherry or CAG-BFP concentrated to 0.5 µg/µl. When pooling libraries for multiplexing, LS-MPRAs for Grm6, Vsx2, and Cabp5 were concentrated to 1 µg/µl and the Rho LS-MPRA library was concentrated to 50 ng/µl, with the bpCAG-mCherry control plasmid concentrated to 0.25 µg/µl.

### Ex vivo electroporation

perinatal mouse and chick eyes were enucleated from animals and placed in fresh 1x PBS. Using a beveled glass needle, plasmid DNA prepared in 1x PBS supplemented with 0.01% Fast Green (Millipore Sigma cat. #F7252) was injected into the subretinal space. Injected eyes were either transferred to a modified electroporation chamber (*Montana et al., 2011*) or placed between platinum disk electrode tweezers (Bulldog Bio, cat. #CUY650P2). All retinas were electroporated with five 50 ms pulses at 40 V delivered at 1 s intervals using a NEPA21 type II Nepagene electroporator; electric currents were applied in both directions between electrodes to maximize electroporation efficiency.

Following electroporation, E14 mouse and E5 chick retinas were removed from surrounding tissues and placed in a culture medium consisting of a 1:1 mixture of DMEM:F12, supplemented with 10% FBS, 200 mM L-glutamine (Sigma Aldrich, cat. #G3126), and 100 U/mL penicillin with 100 mg/mL streptomycin (Invitrogen/Gibco, cat. #15140–122) in microcentrifuge tubes, and cultured on a nutator at 37°C for 16–28 hr, depending on the experiment. P3 mouse retinas were removed from surrounding tissues and placed directly onto a nucleopore track-etched polycarbonate membrane (Watman, cat. #110410) in a 12-well tissue culture dish, floating on 2 ml explant media consisting of 50% Minimum Essential Medium (Millipore Sigma, cat. #51,412 C)(v/v), 25% HBSS (Thermo Fisher Scientific, cat. #14-025-092) (v/v), 25% heat-inactivated horse serum (Thermo Fisher Scientific, cat. #26050–088), 200 mM L-glutamine (Sigma Aldrich, cat. #G3126), 5.75 mg/mL glucose, 25 mM HEPES (Invitrogen/Gibco, cat. #15630–080), and 100 U/mL penicillin with 100 mg/mL streptomycin (Invitrogen/Gibco,

cat. #15140–122). The P3 retinas were covered with 20 µl of explant media and incubated at 37°C, 5% $CO_2$ for 7 days; each day, 1 ml of media was replaced with new media and 20 µl of explant media was reapplied on the retina to keep it moist.

## In vivo electroporation

P0 and P3 mouse pups were anesthetized by cryoanesthesia on ice and injected subcutaneously with 0.012 mg/ml buprenorphine (approximately 20 µl; PAR Pharma). A small skin incision was made by a 30-gauge needle across the eyelid and the eye was exposed. A small drop of 0.5% proparacaine hydrochloride (Bausch & Lomb, cat. #24208-730-06) was placed onto the surface of the eye. A beveled glass needle containing plasmid DNA (prepared in 1×PBS supplemented with 0.01% Fast Green, Millipore Sigma cat. #F7252) was inserted through the sclera into the subretinal space. Plasmid DNA was delivered using a Femtojet Express pressure injector (Eppendorf, cat. #920010521) set at 330 Pa for 3 s. The eyelids were then closed with a cotton applicator. Tweezer-type electrodes (Harvard Apparatus, BTX, model 520, 7 mm diameter, cat. #450165) were placed with the positive end slightly above the injected eyelid and the negative electrode on the contralateral side of the head. An electric field was applied using an electroporator (BEX, cat. #CUY21EDIT), delivering five 50 ms pulses at 80 V with 1 s intervals. Eyelids were dried with a cotton applicator to ensure eyelid closure.

## In ovo electroporation

Fertilized chick eggs were incubated at 37°C in a humidified chamber until they reached Hamburger and Hamilton (HH) stages 11–13 (E2) or stages 22–24 (E4). A small window was created in the eggshell, and a solution containing plasmid DNA was injected into the neural tube using a beveled glass needle. Platinum chopstick electrodes with 4 mm tips (Bulldog Bio, cat. #CUY611P7-4) were positioned flanking the neural tube: at E2, electrodes were placed against the outer membrane to ensure targeted DNA delivery into the ventral neural tube. At E4, due to the natural 90° rotation of the neural tube, a small hole was made in the membrane to enable precise electrode placement on the lateral sides. Electroporation was performed using a NEPA21 Type II Nepagene electroporator with five 50 ms pulses at 25 V delivered at 1 s intervals. After electroporation, the eggs were resealed and incubated for 24–48 hr before tissue collection. Electroporation success was assessed by detecting fluorescence from control plasmids (bpCAG-mCherry or CAG-BFP).

To ensure sequencing replicates contained a sufficient representation of enriched barcodes at each age, pools of at least eight mouse retinas for each E14 LS-MPRA and d-MPRA experiment, three mouse retinas for each P3 LS-MPRA experiment, three chick retinas or 1–2 chick spinal cords for each LS-MPRA experiment were used. When indicated and either coupled with ex vivo electroporation or ex vivo culture alone, the γ-secretase inhibitor LY411575 (MilliporeSigma, cat. #SML0506) was added to a final concentration of 1 µM. A 10,000x stock was prepared at 10 mM in DMSO and stored at –20°C. Just prior to use, the 10 mM stock was diluted 1:10 in 1x PBS to yield a 1000x working solution; 1 µl of the working solution was added per 1 ml culture.

## RNA isolation

To isolate RNA for barcode sequencing or to quantify gene expression, up to eight mouse retinas, three chick retinas, or two chick spinal cords were pooled and homogenized using 21-gauge needles in 250 µl TRI Reagent (MilliporeSigma, cat. #T9424). Total RNA was isolated using the Zymo Directzol Microprep Kit (Zymo Research, cat. #R2061), omitting on-column DNase treatment. For LS-MPRA and d-MPRA library preparations, polyA RNA was subsequently enriched using the Dynabeads mRNA Purification Kit (Thermo Fisher Scientific, cat. #61006). The specific cDNA synthesis protocols for each experiment are detailed in the following sections.

## Retinal cell dissociation

Mouse retinas electroporated at E14 or P3 and cultured ex vivo were dissociated using papain digestion. For each sample, a papain mix was prepared by combining 315 µl HBSS, 35 µl 1 M HEPES (pH 7), 20 µl 50 mM L-cysteine (freshly prepared), 0.4 µl 0.5 M EDTA (pH 8), 19.6 µl UltraPure Water, and 10 µL papain solution (Worthington Biochemical, cat. #LS003126), followed by incubation at 37°C for 15 min to activate the enzyme. Retinas were incubated in 400 µL of activated papain mix at 37°C for 7 min, then centrifuged at 600×g for 2.5 min. The supernatant was removed, and cells were resuspended

in 1 mL HBSS/FBS mix (10% FBS in HBSS), centrifuged again, and resuspended in 600 µL DMEM/ BSA mix (4 mg/mL BSA in DMEM). Cells were gently triturated using a fire-polished Pasteur pipette or a P1000 low-retention pipette, centrifuged again at 600×g for 2.5 min, and finally resuspended in 200–1000 µl of the appropriate solution for downstream applications.

## Fixed tissue preparation

Retinas electroporated in vivo were first dissected from surrounding tissues while retaining the lens in 1x PBS. Retinas in ex vivo culture were removed from the culture media. Spinal cords in ovo were dissected from surrounding tissues in 1x PBS. All tissues were fixed in 4% paraformaldehyde in 1x PBS for 20 min and rinsed three times in 1x PBS for 30 min. Tissues were transferred to 30% sucrose in 1x PBS and gently agitated until fully infiltrated (approximately 30–60 min). Retinas were equilibrated in a 1:1 mixture of 30% sucrose in 1x PBS and Tissue-Tek OCT (VWR, cat. #25608 930), and were then placed in cryomolds (Sakura Finetek, cat. #4565) in this medium and snap frozen on dry ice. Spinal cords were equilibrated in 100% Tissue-Tek OCT prior to snap freezing. Cryosections (20 µm thick) were air dried for 20 mins at 37°C and stored at –80°C.

## Fluorescent in situ hybridization

For multiplexed HCR RNA-FISH, probe sets were manufactured by Molecular Instruments based on the sequences provided: *Olig2* (NM_016967.2), *Neurog2* (XM_030252394), *OLIG2* (NM_001031526.1), and *GFP* (d2eGFP sequence; Lot no. RTA210). Frozen sections on slides were rinsed two times for 5 mins in PBS, pre-hybridized in probe hybridization buffer (Molecular Instruments HCR Buffers [v3.0]) for 10 min at 37°C, and hybridized with 0.4 pmol of each probe set (Molecular Instruments HCR Probes [v3.0]) in hybridization buffer O/N at 37°C. Excess probes were removed in serial 15 min incubations at 37°C in mixtures of 1:0, 3:1, 1:1, 1:3, and 0:1 of probe wash buffer (Molecular Instruments HCR Buffers [v3.0]) and 5x sodium chloride sodium citrate with 0.1% Tween-20 (SSCT) followed by a 5 min rinse at RT in 5x SSCT. Sections were then incubated in amplification buffer (Molecular Instruments HCR Buffers [v3.0]) for 30 min at RT before incubating in a dark humidifying chamber with 6 pmol of h1 and 6 pmol of h2 snap-cooled (heated at 95°C for 90 s and incubated at RT for 30 min in the dark) hairpins for each probe set in amplification buffer for 6–18 hr at RT. Alex546- and Alexa647-labeled hairpins were used. Excess hairpins were removed in sequential rinses in 5x SSCT at RT for 5 min, 2×30 min, and 5 min and proceeded to immunostaining.

## Immunohistochemistry

New frozen sections on slides that were first washed two times for 5 min in PBSTw (1x PBS with 0.1% Tween-20), or immediately following RNA-FISH, were blocked with 2.5% donkey serum (Jackson ImmunoResearch, cat. #017-000-121) in PBSTw for 1 hr at RT. Tissues were incubated in primary antibodies O/N at 4°C and with secondary antibodies for 1 hr at RT in blocking solution, rinsed three times with PBSTw between and after antibody incubations, and mounted using Fluoromount-G (SouthernBiotech, cat. # 0100–01). Mouse monoclonal antibodies raised against Olig2 (1:500, Millipore cat. #MABN50), chicken polyclonal antibodies raised against GFP (1:1000, Abcam, cat. #ab13970), and rabbit polyclonal antibodies raised against BFP (1:200, Abcam, cat. #ab286131) were used. Alexa405-, Alexa488-, and Alexa647-conjugated secondary antibodies (1:200; Jackson ImmunoResearch) raised in donkey against respective primary antibody species, as well as DAPI (1:1000, Invitrogen cat. #D1306), were also used.

## Microscopy

Samples were imaged with a Yokogawa CSU-W1 single disk (50 mm pinhole size) spinning disk confocal unit attached to a motorized Nikon Ti inverted microscope equipped with a Nikon linear-encoded motorized stage with a Mad City Labs 500 mm range Nano-Drive Z piezo insert, an Andor Zyla 4.2 Plus sCMOS camera using a Nikon S Plan Fluor 40 x/1.3 DIC H/N2 oil immersion objective lens. The final digital resolution of the image was 0.16 µm/pixel. Fluorescence from (405 nm, 488 nm, 550 nm, and 640 nm) was collected by illuminating the sample with directly modulated solid-state lasers 405 nm diode 100 mW (at the fiber tip) laser line, 488 nm diode 100 mW laser line, 561 nm DPSS 100 mW laser line and 640 nm diode 70 mW laser line in a Toptica iChrome MLE laser combiner, respectively.

Signal from each channel was acquired sequentially with hard-coated Chroma ET455/50 nm, Chroma ET525/36 nm, Chroma ET605/52 nm, and Chroma ET705/72 nm emission filters in a filter wheel placed within the scan unit, for blue, green, red, and far-red channels, respectively. All images were captured using the 16-bit dual-gain high dynamic range camera mode and no binning. Nikon Elements AR acquisition software was used to acquire the data. Z-stacks were acquired using a Piezo Z-device, with the shutter closed during axial movement. Images were acquired by collecting all colors in a Z-step, then each Z-stack before XY-stage movement, with a 15% tile overlap for correct stitching. Data were saved as ND2 files.

## Vector design

### Barcoded Stagintbc7

The parent vector, Stagint7, is a minimal pUC19-based construct containing a kanamycin resistance gene, an SV40 polyadenylation site, a multiple cloning site immediately upstream of a minimal TATAA promoter, an intron-containing EGFP open reading frame, and a rabbit β-globin polyadenylation sequence. This vector is based on the previously described Stagia3 vector used for retinal enhancer assays (*Billings et al., 2010*). The parent vector was linearized between the EGFP ORF and polyadenylation sequence via an AscI (NEB, cat. #R0558) recognition sequence, inserting Illumina adapter sequences flanking a recreated AscI recognition sequence, a unique 8 bp library barcode tag for multiplexing (used only for Rho, Grm6, Vsx2, and Cabp5 libraries), and a 24 bp barcode using an Enhanced Gibson Assembly reaction (*Rabe and Cepko, 2020*). Barcodes consisted of either twelve SW base repeats or four VHBD base repeats to prevent high GC content and homopolymer repeats as well as to provide sufficient barcode complexity. The barcoded Stagintbc7 vector used to create LS-MPRA libraries (*Rabe, 2020*) was generated by combining 100 ng of linearized parent vector with 25 nM of each primer pair (see *Supplementary file 1a*) and, along with the pre-mixed Enhanced Gibson Assembly reaction, incubated at 50°C for 1 hr. The assembled product was purified with a Zymo DNA Clean & Concentrator Kit (Zymo Research, cat. #D4013).

For the creation of the four Stagintbc7 multiplexed libraries, four unique DNA library tags were generated using the DNABarcodes R package (*Buschmann, 2017*) to ensure that all barcodes differed from one another by at least five bases while avoiding homopolymer repeats. The create.dnabarcodes function was used with a Hamming distance metric of 5, and additional filters were applied to exclude triplet repeats, ensure GC content balance, and prevent self-complementarity. The selected library barcode tags included: ATACAGGC, TGTGCAAG, CACGAAGA, and GTGAACCT.

### Statadual_WPRE

This vector, used for d-MPRA library construction, is derived from Stagia3 but replaces the minimal TATAA promoter and intron-containing EGFP ORF of Stagint7 with a WPRE sequence.

## Assembly of LSMPRA libraries

### BAC isolation and preparation

BACs were first isolated from stab cultures (BACPAC Resources). Cultures were streaked onto chloramphenicol-treated LB agar plates and incubated overnight at 37°C. A single colony was picked and grown in LB media supplemented with chloramphenicol (12.5 µg/mL) in 5 ml overnight, followed by expansion to 500 ml. The culture was pelleted, and BAC DNA was extracted using a large construct DNA isolation kit with ATP (Qiagen, cat. #12462). DNA concentrations were determined using the Qubit dsDNA HS Assay Kit (ThermoFisher, cats. #Q32851 and # Q32856). For sequencing validation, samples were sent to Plasmidsaurus for large plasmid sequencing. BAC clones used in all experiments included RP23-219M6 (*Rho*), RP23-417M10 (*Grm6*), RP23-127O21 (*Vsx2*), RP24-125H22 (*Cabp5*), CH29-613 (*Olig2*), RP23-182M12 (*Ngn2*), and CH261-60J3 (*OLIG2*).

### BAC fragmentation and adapter ligation

BAC DNA (500 ng) was enzymatically digested, end-repaired, and dA-tailed using the NEBNext Ultra II FS DNA Module kit (NEB, cat. #E7810S) with a 20-min enzyme incubation. Digested DNA was purified using SPRIselect beads with a 1.2x left-side size selection ratio. Concentrations were determined using the Qubit dsDNA HS Assay Kit. Double-stranded adapters (*Supplementary file*

*1a*) were annealed from complementary oligonucleotides and diluted to a molar concentration approximately 100-fold higher than fragments in 1x T4 DNA ligase buffer (NEB, cat. #B0202S). The 5' adapter sequence included an Illumina adapter sequence and a 25-bp homologous sequence for Gibson Assembly insertion into the Stagintbc7 vector, while the 3' adapter sequence included a 23-bp homologous sequence and an AscI recognition sequence. Adapters were ligated to fragments using the Blunt/TA Ligase Master Mix (NEB, cat. #M0367), adding a 1:1 mixture of ligase mix and 18 µl fragments with 2 µl adapters, incubating at RT for 1 hr. Adapter-ligated DNA was purified using the Zymo DNA Clean & Concentrator Kit and quantified using the Qubit dsDNA HS Assay Kit.

## Adapter-ligated fragment amplification and purification

Adapter-ligated fragments were amplified using NEBNext Ultra II Q5 Master Mix (NEB, cat. #M0544) in a 300 µl reaction with 0.5 µM primers (**Supplementary file 1a**) complementary to the 5' and 3' adapter ends and at an annealing temperature of 64°C for 15 cycles. Amplified DNA was purified using the Zymo DNA Clean & Concentrator Kit. To ensure proper fragment size selection, adapter-ligated fragments were resolved on a 1% agarose gel, and the desired size range (fragment plus ~90 bp of adapter sequence) was gel-purified using Agarose Dissolving Buffer (Zymo Research, cat. #D4001-1-100) and the Zymo DNA Clean & Concentrator Kit.

## Cloning into barcoded library

The Stagintbc7 barcoded library was linearized with PacI (NEB, cat. #R0547) and EcoRI-HF (NEB, cat. #R3101). The digested vector was purified via 0.75% agarose gel electrophoresis and gel extraction. An Enhanced Gibson Assembly was performed with 100 ng vector and a 10:1 molar insert:vector ratio at 50°C for 1 hr. The assembled product (Rabe, 2020) was purified with a Zymo DNA Clean & Concentrator Kit.

## Library transformation and amplification

All barcoded libraries—Stagintbc7, LS-MPRA, and d-MPRA—were electroporated into DH10β electrocompetent cells in a pre-cooled 1 mm cuvette (BTX, cat. #45 0134) with the following parameters: 2 kV, 25 µF, and 200 Ω on a BioRad GenePulser II. Transformed cells were outgrown in 2 ml SOC media at 37°C for 2 hr, and then grown in 3D culture by embedding bacterial transformations in 0.5% sodium alginate matrices with $CaCO_3$ and LB to reduce clonal bias inherent in 2D cultures. To prepare 3D cultures (Rabe, 2020), the transformed bacterial cells grown in SOC were mixed with 12.5 ml of 1% sodium alginate (mixed 1 g/100 ml LB O/N, then autoclaved) and LB to a volume of 20 ml. Then, 250 µl of a $CaCO_3$ suspension (10 g in 100 ml diH$_2$O; Sigma Aldrich, 310034) and 1000x kanamycin (25 µL) were combined in 2.5 ml LB, mixed immediately before use, and added to the bacterial-alginate solution. Lastly, 41.7 µl of 37% HCl in 2.5 ml LB was added for a final volume of 25 ml, and the mixture was gently rocked for 5 min before being poured into a sterile 150 mm petri dish. Cultures were incubated at 37°C for 16–20 hr to allow gel formation and outgrowth.

Following incubation, 6.25 ml of 20x SSC+ 40 mM EDTA solution (pH 8.0) was added to dissolve the 3D culture and was gently rocked at RT for 15 min. Bacterial cells were pelleted (6000×g, 15 min), washed with 25 ml LB, and centrifuged again. The pellet was resuspended in 300 ml LB containing 50 µg/ml kanamycin and 0.05 mg/ml alginate lyase (Sigma Aldrich, cat. #A1603) and incubated at 37°C with shaking (250 RPM) for 6–8 hr. Cultures were then pelleted for plasmid preparation or storage at –20°C. Plasmids were extracted using a Plasmid Maxi kit (Qiagen, cat. #12163).

To estimate the complexity of transformed libraries, a serial dilution was performed after the transformation and prior to 3D culture. One microliter of the 2 ml culture in SOC was removed and subjected to three sequential dilutions (1:500, 1:5000, and 1:50,000) to represent 1/10,000[th], 1/100,000[th], and 1/1,000,000[th] of the total transformation, respectively. For dilution preparation, 1 µl of the culture was added to 499 µl SOC, mixed, and 50 µl of this dilution was transferred to 450 µl SOC, followed by an additional 1:10 dilution under the same conditions. 100 µl of each dilution was plated onto kanamycin-treated LB agar and incubated O/N at 37°C. The number of resulting colonies was used to estimate transformation complexity, with a quantifiable optimal range of 20–200 colonies per plate.

## Assembly of d-MPRA libraries

CRMs of interest were PCR-amplified from their respective BAC clones using primers designed to incorporate 20–30 bp overlapping sequences for Gibson Assembly into the Statadual_WPRE vector (*Supplementary file 1a*). PCR reactions were performed using NEBNext Ultra II Q5 Master Mix under standard conditions, and amplicons were gel-purified using the Zymo DNA Clean & Concentrator kit.

Mutagenesis was performed using the GeneMorph II Random Mutagenesis Kit (Agilent, cat. #200550), following the manufacturer's protocol with 100 pg of input DNA and MutF/R primers (*Supplementary file 1a*), which amplify the entire CRM region and incorporated overlapping sequences. PCR products were gel-purified, and 100–200 ng of each mutagenized fragment was assembled into the EcoRI/PacI-linearized Statadual_WPRE vector via Enhanced Gibson Assembly at a 10:1 insert-to-vector molar ratio. The assembled libraries were electroporated into DH10β electrocompetent cells, expanded in 3D bacterial culture, and purified as previously described.

For intraplasmid duplication (IPD), 3 µg of purified intermediate plasmid libraries were digested with the nicking endonuclease Nt.BspQI (NEB, cat. #R0644) at 50°C for 1 hr, column-purified, and incubated with Bst 2.0 WarmStart Polymerase (8 units, NEB, cat. #M0538) in 1x Isothermal Amplification Buffer supplemented with 6 mM $MgSO_4$ and 200 µM dNTPs (GoldBio, cat. #D9001) in a 50 µl reaction at 65°C for 45 min, followed by enzyme inactivation at 80°C for 20 min. To reduce self-ligation, 2.5 units of Taq DNA polymerase (NEB, cat. #M0273) was added, and the reaction was incubated at 65°C for 20 min for dA-tailing. The prominent linear band was gel-excised and purified.

The final d-MPRA insert (a minimal TATAA promoter and intron-containing EGFP ORF) was PCR-amplified from Stagint7 with primers designed to incorporate 20–30 bp overlapping sequences for Gibson Assembly into the linearized IPD product (*Supplementary file 1a*), gel-purified, and assembled with 25 fmol of linearized, IPD-processed vector in an Enhanced Gibson Assembly reaction (Rabe, 2020). The completed d-MPRA libraries were electroporated into DH10β electrocompetent cells, expanded in 3D bacterial culture, and purified as previously described.

## CUT&RUN

Experiments were performed using freshly dissociated E14 mouse retinas with the CUTANA ChIC/CUT&RUN Kit (EpiCypher, cat. #14–1048) according to the manufacturer's protocol. Mouse monoclonal antibodies raised against Pax6 (Developmental Studies Hybridoma Bank, cat. #AB_528427) and against Foxn4 (Santa Cruz Biotechnologies, cat. #sc-377166) as well as rabbit polyclonal antibodies raised against Mybl1 (Millipore-Sigma, cat. #HPA008791) were used, adding 500 ng of the respective antibody to each sample during the antibody binding stage. Indexed libraries were prepared with the NEBNext Ultra II DNA Library Prep Kit (NEB, cat. #E7103) and the NEBNext Multiplex Oligos for Illumina Dual Index Sets (NEB, cats. #E7600S or #E7780S). All experiments were validated following Next-Generation sequencing using the SNAP-CUTANA K-MetStat Panel of nucleosome spike-ins in control reactions and the manufacturer's protocol (Appendix 1.6; EpiCypher, cat. #14–1048).

## Sequencing library preparation

### LSMPRA fragment-barcode association libraries

To excise the minimal promoter and GFP ORF, enabling BAC fragments to be positioned adjacent to their corresponding barcodes for sequencing, 2–3 µg of the LS-MPRA library was digested with AscI and resolved on a 1% agarose gel (Rabe, 2020). The fragment containing both the barcode and insert (~2.1 kb plus the fragment size) was excised and gel-purified. 500 ng of purified fragment was circularized in a 100 µl Quick Ligation reaction (NEB, cat. #M2200) at RT for 5 min. The ligation product was column-purified and treated with 50 µl exonuclease (Lucigen, cat. #E3101K) to remove residual linear fragments.

Following gel resolution, the smallest and brightest band was excised and gel-purified. Because the ligation product already contained Illumina read primer sequences, 50 ng of purified product was used as a template for index PCR with NEBNext Q5 Ultra II Master and NEBNext Multiplex Oligos for Illumina Dual Index Set for 15 cycles. PCR products were purified via two-sided SPRISelect bead purification (0.5x to 0.9x).

## LSMPRA barcode abundance libraries

To assess barcode abundance in RNA, polyA RNA was reverse-transcribed using the Protoscript II Kit (NEB, cat. #E6560) and a reverse transcription primer at 1µM specific to all barcoded transcripts (bc7_cdnaR; *Supplementary file 1a*). The resulting cDNA was purified using the Zymo Oligo Clean & Concentrator Kit (Zymo, cat. #D4060) and then amplified for 30 cycles using NEBNext Q5 Ultra II Master Mix with two primers: one that anneals to a region of the GFP ORF upstream of the intron and the other downstream of the barcode-flanking Illumina adapter sequences [bc7_cdnaF and bc7_cdnaR2 (*Supplementary file 1a*), respectively], ensuring the intron-containing plasmid DNA was not included. The cDNA-derived amplicons (~356–364 bp) were gel-purified from a 1% agarose gel. Indexed barcode amplicons were generated with 50 ng purified product, NEBNext Q5 Ultra II Master Mix, and NEBNext Multiplex Oligos for Illumina Dual Index Set for 15 cycles, followed by two-sided purification using SPRISelect beads (0.95x-1.2x).

## d-MPRA libraries

Amplicons for d-MPRA mutagenesis analysis were generated from both plasmid libraries and RNA (*Rabe, 2020*). cDNA was synthesized from polyA RNA using the Protoscript II Kit and the reverse transcription primer at 1 µM specific to all d-MPRA barcoded transcripts [bc7_cdnaR_WPRE, (*Supplementary file 1a*)], purified with the Zymo Oligo Clean & Concentrator Kit, and amplified for 30 cycles using NEBNext Q5 Ultra II Master Mix with two primers: one that anneals to a region of the GFP ORF upstream of the intron and the other downstream of the barcode and upstream of the WPRE sequence [bc_cdnaF and MutR, respectively (*Supplementary file 1a*)]. The resulting cDNA band (~290 bp plus the CRM size) was gel-purified. For plasmid d-MPRA libraries, the corresponding fragment was generated by digesting with BsrGI (NEB, cat. #R3575) and NdeI (NEB, cat. #R0111), which excises the CRM barcode and WPRE sequence, and gel purifying the product (~760 bp plus the CRM size).

For both plasmid library and RNA-derived amplicons, Illumina adapter sequences were added to each side of the mutated CRM sequence [dualmut_IllR1F or dualmut_IllR2F and dualmut_IllR2R primers (*Supplementary file 1a*)]. These fragments, amplified from 50 ng template with NEBNext Q5 Ultra II Master Mix, were gel-purified and subjected to index PCR with NEBNext Multiplex Oligos for Illumina Dual Index Set for 15 cycles. PCR products were purified via two-sided SPRISelect bead purification (0.6x to 0.9x).

## Illumina sequencing

All indexed LS-MPRA, d-MPRA, and CUT&RUN libraries were analyzed using an Agilent TapeStation High Sensitivity D1000 assay. Indexed samples were sequenced using 2×150 bp paired-end reads on an Illumina NovaSeq instrument at either MedGenome Inc or Admera Health, with the range of sequencing depth between 20 M-50M reads.

## Expression constructs

CRM expression constructs were cloned into the Stagia3 expression vector with an Enhanced Gibson Assembly (*Rabe and Cepko, 2020*). For all constructs, the vector backbone was amplified from the expression vector using primers compvec_F/R (*Supplementary file 1a*). For Olig2 and OLIG2 constructs, the intron-containing EGFP (GFPint) along with the rabbit β-globin polyadenylation sequence was amplified from Stagint7, whereas the insert for the Ngn2 constructs also included the TATAA minimal promoter upstream of EGFP. Olig2 CRMs, the Olig2 minimal promoter, Ngn2 CRMs, OLIG2 CRMs, and the OLIG2 minimal promoter were amplified from their respective BAC clones (all primers in *Supplementary file 1a*). Coordinates for all regions are included in *Supplementary file 1b*.

For individual constructs, Olig2-NR1 or Olig2-NR2 CRMs were inserted upstream of the Olig2 minimal promoter and GFPint, whereas Olig2-NR3 CRM was assembled downstream of the GFPint+ polyA sequence. The Olig2 composite construct was assembled accordingly, placing the NR1 and NR2 fragments adjacent to each other and upstream of the Olig2 minimal promoter, GFPint+ polyA sequence, and NR3 fragment. Backbone plasmids containing GFPint alone or GFPint under the Olig2 minimal promoter in the absence of CRM fragments were also assembled (primers in *Supplementary file 1a*).

The mouse Ngn2-CRM constructs were assembled by placing the Ngn2-CRM1, CRM2, or CRM3 fragments upstream of the TATAA minimal promoter and GFPint, while the CRM4 fragment was

placed downstream of the GFPint+ polyA sequence. The chick OLIG2-CRM constructs were similarly constructed, with OLIG2-CRM1 or CRM2 fragments assembled upstream of the cOLIG2 minimal promoter, while the OLIG2-CRM3 fragment was assembled downstream of the GFPint+ polyA sequence.

## ddPCR

All ddPCR assays were performed on the BioRad ddPCR system, including the automated droplet generator (BioRad, cat. #1864101). *Olig2* gene expression was determined using Evagreen-based ddPCR with *Hprt* as a normalization control (primers in *Supplementary file 1a*). cDNA was generated from isolated RNA with the LunaScript RT SuperMix Kit (NEB, cat. #E3010) from ~2–3 ng of total RNA and purified with the Zymo Oligo Clean & Concentrator kit. 1 ng of cDNA was added to each 22 µl ddPCR reaction; no RT controls confirmed the absence of genomic DNA amplification (all assays spanned exon-exon junctions).

## Sequencing, quantification, and statistical analyses

### LS-MPRA barcode association and quantification analysis

Paired-end 150 bp sequencing reads were processed to establish barcode-fragment associations and quantify barcode abundances. Two custom shell scripts were used for sequencing data analysis.

### Barcode-fragment association

For barcode-fragment association, sequencing reads were aligned to the mm10 reference genome using Minimap2 version 2.28 (*Li, 2018*) and Bowtie2 version 2.5.4 (*Langmead and Salzberg, 2012*). Adapter sequences were trimmed using bbduk from the BBTools suite version 39.18 (*Bushnell et al., 2017*), and barcode sequences were extracted from Read 1 using UMI-tools extract version 1.1.6 (*Smith et al., 2017*). Valid barcode-fragment pairs were identified using UMI-tools group and filtered for library-specific sequences. Genomic coordinates were corrected based on barcode positions, and results were output as BigWig tracks using deepTools bamCoverage version 3.5.5 (*Ramírez et al., 2016*). Genomic coordinate manipulation, barcode overlap correction, and file conversions were performed using Bedtools version 2.31.1 (*Quinlan and Hall, 2010*), Samtools version 1.21 (*Danecek et al., 2021*; *Li et al., 2009*), and SeqKit version 2.6.1 (*Shen et al., 2016a*).

An earlier version of the pipeline differed primarily in its alignment and filtering steps. Instead of Minimap2, it used BLAT version 35 (*Kent, 2002*) to align BAC sequences to the genome, extending flanking regions by 1 kb. Additionally, overlapping barcode-associated reads were collapsed using R packages GenomicRanges version 1.50.2 (*Lawrence et al., 2013*), regioneR version 1.30.0 (*Gel et al., 2016*), and data.table version 1.16.4 (*Barrett et al., 2025*). Barcode extraction followed a similar approach but lacked the library-specific filtering step included in the updated version. By replacing BLAT with Minimap2 and incorporating stricter filtering, the updated pipeline improved alignment efficiency and accuracy.

### Barcode abundance quantification

For barcode abundance quantification, extracted barcode sequences were filtered against a whitelist to exclude sequencing errors and ensure accuracy. The whitelist was generated by extracting reverse-complement barcode sequences from a reference FASTA file containing known barcode-fragment associations using the Biostrings R package version 2.66.0 (*Pagès et al., 2025*). Pairwise Hamming distances were computed using the stringdist R package version 0.9.15 (*van der Loo, 2014*) to define a mutation threshold. All possible barcode variants within this threshold were enumerated and stored in an RDS file for downstream filtering. Sequencing reads were matched against this whitelist using UMI-tools extract, and barcode counts were normalized to library input using deepTools bamCompare. Normalized BigWig tracks were generated for visualization and analysis.

An earlier version of the pipeline extracted barcodes using bbduk and SeqKit, without applying Hamming distance-based error correction. Unique barcode counts were merged with barcode-fragment associations, and fragment counts were expanded to match observed barcode frequencies before normalization with bamCompare. While both methods yielded comparable results, the whitelist-based approach provided better error correction and filtering, enhancing reliability in downstream analyses.

All LS-MPRA plots were displayed on the Integrative Genomics Viewer (IGV) (*Robinson et al., 2011*), and aligned with analyses of genetic conservation from the UCSC Genome Browser database (*Perez et al., 2025*) as well as with RefSeq genes (*O'Leary et al., 2016*).

## ATAC-seq analysis

Open chromatin regions were determined based on pseudo-bulk re-analysis of published single-cell ATAC-seq datasets (*Lyu et al., 2021*), and deposited under accession number GSE181251. Deposited bam files were converted back to FASTQ format using bamtofastq version 1.3.2 from 10X Genomics (*Github, 2022*). Single-cell FASTQ files for E14 and P8 mouse retinas were analyzed using cellranger atac count version 2.0.0 (*Satpathy et al., 2019*), where reads were filtered and aligned to the mm10 mouse genome, barcode counted, and transposase cut sites identified. Regions of accessible chromatin were detected on a single-cell basis. Bam files contained all the aligned paired-end reads were used to determine pseudo-bulk genome coverage using Deeptools version 3.5 (*Ramírez et al., 2016*), normalized using RPKM, and filtered for blacklisted (*Amemiya et al., 2019*), gap and repeat regions, and visualized using IGV (*Robinson et al., 2011*; *Thorvaldsdóttir et al., 2013*).

## CUT&RUN analysis

Analysis of sequenced samples from the CUT&RUN experiments were analyzed as previously described (*Catta-Preta et al., 2025*). FASTQ files containing the reads were controlled for quality using FASTQC version 0.11.9 (*Andrews, 2010*), and further cleaned up of remaining adapter sequences using Trim Galore version 0.6.6 (*Krueger, 2015*). The resulting pair-ended reads were aligned to the mm10 mouse genome using BWA mem version 0.7.17 (*Li and Durbin, 2009*), and duplicates removed with Samtools version 1.15.1 (*Li et al., 2009*). Peaks of enriched immunoprecipitated regions against IgG (negative) control were called using MACS version 2.1.1 (*Zhang et al., 2008*) with a p-value of 0.05, extension size 200, no lambda baseline correction. Genome coverage from the bam files for all samples was determined using Deeptools version 3.5 (*Ramírez et al., 2016*), normalized using RPKM, and filtered for blacklisted (*Amemiya et al., 2019*), gap, and repeat regions. Coverage was visualized using IGV (*Robinson et al., 2011*; *Thorvaldsdóttir et al., 2013*). Motif enrichment and specific motif presence was determined using Homer version 4.11.1 (*Heinz et al., 2010*).

## D-MPRA plasmid and experimental library processing

To analyze barcode-fragment associations and quantify changes in mutated fragment (i.e. barcode) frequency in d-MPRA plasmid and experimental libraries, sequencing data were processed through a pipeline involving read merging, alignment, barcode extraction, and statistical analysis of barcode frequencies across CRM sequences.

### Plasmid library processing

Plasmid control libraries were processed by merging paired-end reads and aligning them to reference sequences using Minimap2 and pysam version 0.22.1 (*Bonfield et al., 2021*). Barcode and fragment associations were then analyzed by parsing Minimap2's cs tags, with mutations identified and quantified using custom Python scripts that leveraged NumPy version 1.26.3 (*Harris et al., 2020*) and pandas version 2.1.4 (*McKinney, 2010*) for data handling. Reads containing deletions within homopolymer regions were excluded based on positions identified from reference FASTA sequences, and reads failing to align fully across the contig were discarded. Monte Carlo simulations were implemented using multiprocessing to perform 10 independent subsampling replicates, and barcode mutation frequencies were normalized to wild-type sequence counts for accurate mutation rate estimation.

### Experimental library processing

Reads from experimental d-MPRA libraries were merged and aligned similarly to the plasmid workflow, and barcode-fragment associations were identified. An additional filtering step was applied to remove reads that failed to align fully across the contig, reducing false-positive mutation calls.

### Mutation filtering and statistical analysis

Both plasmid and experimental workflows applied stringent mutation filtering, discarding deletions occurring in homopolymer regions. Substitutions and deletions were quantified using pysam to parse

cs tags from alignments produced by Minimap2 and filtered with logic implemented in numpy and pandas. Barcode frequencies were normalized relative to plasmid control counts using data structures and sampling routines implemented in pandas, numpy, and Python's multiprocessing library. Monte Carlo sampling was used to refine barcode frequency estimates while accounting for sequencing depth. Final mutation statistics, including mean and standard deviation across positions and replicates, were compiled using pandas and json, ensuring accurate and robust quantification of sequence variation. This workflow ensured precise quantification of barcode frequency in both plasmid and experimental d-MPRA libraries while minimizing sequencing artifacts.

## Visualization of d-MPRA profiles

To visualize barcode frequency distributions across CRMs, normalized mutation rates from Monte Carlo-sampled plasmid controls and experimental d-MPRA libraries were compared and plotted. Barcode frequencies corresponding to substitutions and deletions in the experimental datasets were extracted from position-level JSON summaries. These values were normalized to corresponding wild-type-corrected rates from plasmid controls and $\log_2$-transformed to assess relative mutation enrichment or depletion.

To reduce noise and account for local sequence context, the data were smoothed using a five-base sliding window moving average. A baseline correction was applied to each dataset by subtracting the median $\log_2$-transformed value prior to smoothing. For each genomic position, only sites present in both control and experimental datasets were retained for comparison. Edge regions flanking the CRM, specifically, the first 18 and last 14 positions, were excluded from visualization to eliminate non-informative regions required for construct assembly.

For each replicate, baseline-corrected mutation rates were plotted individually alongside the average profile across all replicates. All visualizations were generated in Python using matplotlib version 3.8.2 (*Hunter, 2007*) and numpy.

## Quantification of *Olig2* transcript levels using ddPCR

ddPCR was used to quantify the concentration of *Olig2* transcripts in treated and control retinas across multiple time points. Normalized transcript concentrations were determined from ddPCR reactions and analyzed in GraphPad Prism. Statistical comparisons between treated and control groups were performed using unpaired t-tests assuming equal standard deviations, with multiple testing correction applied using the Holm-Šídák method.

## Quantification of barcode enrichment using AUC analysis

CRM barcode enrichment was quantified by calculating the area under the curve (AUC) using BigWig sequencing files. Normalized barcode count values were extracted for the CRM genomic regions, and AUC was computed using the trapezoidal rule. Statistical analysis was performed in GraphPad Prism using Brown-Forsythe and Welch's ANOVA tests, followed by Dunnett's T3 multiple comparisons test to assess significance.

## Quantification of cell populations using a semi-automatic colocalization analysis toolbox

Fluorescent cell counting was performed using a semi-automatic object-based colocalization analysis (OBCA) toolbox developed by *Lunde and Glover, 2020*. This toolbox comprises two ImageJ plugins, a Microsoft Excel macro, and a MATLAB script, facilitating efficient analysis of complex imaging data. The first ImageJ plugin was employed to enhance visualization of multichannel 3D images, optimizing features relevant to OBCA. The second plugin enabled semi-automatic quantification of colocalization across multiple fluorescence channels. In this study, cells expressing a specific gene transcript or protein, GFP driven by CRM activity, and BFP from an electroporation control plasmid were identified. The toolbox provided counts of cells co-expressing the gene transcript, GFP protein, and/or GFP transcript. These data were used to calculate the percentage of electroporated (BFP+) cells expressing the specific gene transcript that also expressed GFP protein, GFP transcript, or both, for each analyzed section. Additionally, the percentage of GFP+ cells co-expressing the gene transcript was determined.

## Motif analysis

De novo motif analysis was performed on CRM sequences using the HOMER tool (findMotifsGenome. pl) (*Heinz et al., 2010*). The motif search was performed on CRM coordinates in the mm10 genome assembly using a fixed size for each CRM region and a randomly generated reference genome directory. HOMER's built-in TF binding site database, derived from JASPAR, was used to detect known TF binding sites within the discovered motifs. The analysis output was used to identify potential regulatory factors enriched in these enhancer regions. The comprehensive list of potential TF binding sites within each predicted motif of the Olig2 CRMs is available at the online repository (*Tulloch, 2025*).

## Visualization of PFMs

Position frequency matrices (PFMs) for selected TFBSs were parsed and visualized using a custom Python script. PFMs were read into a pandas DataFrame and transposed to organize nucleotide frequencies by position. Rows were normalized to ensure that nucleotide frequencies summed to one at each position. The information content at each position was calculated using Shannon entropy, incorporating a small pseudocount to avoid undefined logarithmic values. Sequence logos were generated using Logomaker (*Tareen and Kinney, 2020*), with bit scores computed by multiplying nucleotide frequencies by their respective information content.

## Single-cell RNA-seq data acquisition and processing

scRNA-seq data for retinal development across multiple ages were obtained from the published data repository at https://github.com/gofflab/developing_mouse_retina_scRNASeq (*Goff, 2021*; *Clark et al., 2019*). The dataset included:

1. a gene expression counts matrix (*10x_mouse_retina_development.mtx*)
2. gene feature annotations (*10x_mouse_retina_development_feature.csv*)
3. sample-level metadata (*10x_mouse_retina_development_phenotype.csv*).

The raw counts matrix was processed and analyzed using Seurat v4.0 (*Butler et al., 2018*).

### Data preprocessing

The scRNA-seq data were processed in R using Seurat (*Hao et al., 2024*). The raw counts matrix was loaded with `Read10X()` and converted into a Seurat object. Genes detected in fewer than three cells were removed, and cells with fewer than 200 detected genes or >5% mitochondrial gene content were excluded. To normalize for sequencing depth, the SCTransform workflow was applied, followed by the identification of highly variable genes.

### Subset analysis by developmental age

To focus on specific developmental stages, cells were subsetted based on 'age metadata.' Two ages were subsetted: E14 and P8. Dimensionality reduction, clustering, and Uniform Manifold Approximation and Projection (UMAP) visualization (*McInnes et al., 2018*) were performed on each subset independently.

### Dimensionality reduction and clustering

Principal component analysis (PCA) was performed on the top 2000 variable genes. UMAP was used for visualization, and clustering was conducted using `FindNeighbors()` and `FindClusters()` with a resolution of 0.5 to delineate distinct retinal cell populations. UMAP plots were generated using `DimPlot()`, with cells color-coded by cell type, developmental stage, or gene expression levels.

### Gene expression and co-expression analysis

The expression of individual genes of interest was visualized using `FeaturePlot()`. A continuous color gradient (light gray to blue) was applied to indicate expression levels. To visualize gene co-expression, `FeaturePlot()` was used with blend =TRUE, where cells expressing Gene1 appeared in blue, Gene2 in red, and co-expressing cells in pink. The third panel of the blended plot, which represents co-expression, was extracted and saved using `ggsave()`. All visualizations were generated in R using ggplot2 (*Wickham, 2016*).

## Quantification of co-expressing cells

Expression data for genes of interest were extracted from the RNA assay slot of the Seurat object using `GetAssayData()`. Cells were classified as expressing a gene if their count was greater than zero. The number of cells expressing each gene individually and both genes concurrently was computed. The percentage of Gene1-expressing cells that also expressed Gene2 was calculated as:

$$\text{Percent Co-Expression} = (\text{Cells expressing both genes} \text{Total cells expressing Gene1}) \times 100$$

$$\text{Percent Co-Expression} = \left( \frac{\text{Cells expressing both genes}}{\text{Total cells expressing Gene1}} \right) \times 100$$

$$\text{Percent Co-Expression} = (\text{Total cells expressing Gene1} \text{Cells expressing both genes}) \times 100$$

## Statistical analysis of gene co-expression with Olig2 from scRNAseq

To assess whether specific genes were significantly enriched or depleted in *Olig2*-expressing cells, Fisher's exact test for each gene of interest was performed. Gene expression data were obtained from the RNA count matrix of the retinal single-cell dataset. Cells were classified as expressing a given gene if the raw count value was greater than zero.

For each gene, a 2×2 contingency table was constructed, categorizing cells as *Olig2*+ or *Olig2*− and as expressing or not expressing the target gene. Fisher's exact test was then applied to each contingency table to determine whether the presence of a given gene was significantly associated with *Olig2* expression. To account for multiple hypothesis testing, the Benjamini-Hochberg false discovery rate (FDR) correction was applied to the resulting p-values. Genes with an adjusted p-value (FDR q-value) below 0.05 were considered significantly enriched or depleted in *Olig2*-expressing cells. An odds ratio greater than 1 indicated that the gene was enriched in *Olig2*+ cells, whereas an odds ratio less than 1 indicated depletion in *Olig2*+ cells.

## Acknowledgements

First, we thank Dr. Brian Rabe for developing the original LS-MPRA and d-MPRA methods that formed the basis of this study. We would like to thank Dr. Changhee Lee for discussions related to the study design and for expertise in methodology. As well, we thank Dr. Ryoji Amamoto and Henry Bushnell for expertise in methodology. We would also like to thank Dr. Paula Montero Llopis, Dr. Praju Vikas Anekal, and Dr. Adrienne Wells from the MicRoN imaging core at Harvard Medical School for providing expert advice concerning imaging and analysis. We thank BioRender for assisting in the creation of several schematics (https://www.biorender.com/). We are grateful to the Howard Hughes Medical Institute (CLC and AJT), the National Eye Institute (5K99EY034603, RND and 5R01EY029771 RC-P) for support.

## Additional information

### Funding

| Funder | Grant reference number | Author |
|---|---|---|
| Howard Hughes Medical Institute | | Alastair J Tulloch Constance L Cepko |
| National Eye Institute | 5K99EY034603 | Ryan Nicholas Delgado |
| National Eye Institute | 5R01EY029771 | Rinaldo Catta-Preta |

The funders had no role in study design, data collection and interpretation, or the decision to submit the work for publication.

## Author contributions

Alastair J Tulloch, Conceptualization, Data curation, Software, Formal analysis, Validation, Investigation, Visualization, Methodology, Writing – original draft, Writing – review and editing; Ryan Nicholas Delgado, Data curation, Software, Formal analysis, Funding acquisition, Validation, Visualization, Methodology, Writing – review and editing; Rinaldo Catta-Preta, Data curation, Software, Writing – review and editing; Constance L Cepko, Conceptualization, Formal analysis, Supervision, Funding acquisition, Investigation, Methodology, Writing – original draft, Writing – review and editing

## Author ORCIDs

Alastair J Tulloch ⓘ https://orcid.org/0000-0002-7631-4300
Ryan Nicholas Delgado ⓘ https://orcid.org/0000-0002-7551-0182
Rinaldo Catta-Preta ⓘ https://orcid.org/0000-0001-9833-1278
Constance L Cepko ⓘ https://orcid.org/0000-0002-9945-6387

## Ethics

All experimental procedures were preapproved by the Harvard Medical Area Standing Committee on Animals (HMA IACUC protocol #IS00001679-6). Timed and untimed pregnant CD-1 mice used for sequencing and histology were purchased from Charles River Laboratories and were housed in a climate-controlled pathogen-free facility. Mouse tissues were collected for sequencing experiments at E14, P3, or P10 and for histological experiments at E14 or P1 as noted within the text. Chick tissues were collected for sequencing experiments at E3, E5, and E6 and for histological experiments at E3 and E5, as noted within the text.

Reviewer #1 (Public review): https://doi.org/10.7554/eLife.107565.3.sa1
Reviewer #2 (Public review): https://doi.org/10.7554/eLife.107565.3.sa2
Reviewer #3 (Public review): https://doi.org/10.7554/eLife.107565.3.sa3
Author response https://doi.org/10.7554/eLife.107565.3.sa4

---

# Additional files

## Supplementary files

Supplementary file 1. PCR primers, genomic coordinates, and cell type specificity quantification. (a) PCR primers used in all experiments. (b) Genomic coordinates for all regions of interest. (c–g) Quantification of EGFP expression from electroporated plasmid constructs and colocalization with cell-type-specific gene expression.

MDAR checklist

Source data 1. LS-MPRA library characteristics and supplemental QC plots.

Source data 2. D-MPRA library supplemental QC plots.

## Data availability

All original code and unprocessed sequencing files are available on the Harvard Dataverse (DOI:https://doi.org/10.7910/DVN/TW0ZQL). All processed LS-MPRA genome tracks are available to view on a Track Hub in the UCSC Genome Browser (https://github.com/cattapre/ALAS00 copy archived at *Catta-Preta, 2025*).

The following dataset was generated:

| Author(s) | Year | Dataset title | Dataset URL | Database and Identifier |
|---|---|---|---|---|
| Tulloch A | 2025 | LS-MPRA / d-MPRA Data Repository | https://doi.org/10.7910/DVN/TW0ZQL | Harvard Dataverse, 10.7910/DVN/TW0ZQL |

The following previously published dataset was used:

| Author(s) | Year | Dataset title | Dataset URL | Database and Identifier |
|---|---|---|---|---|
| Lyu P, Hoang T, Santiago CP, Thomas ED, Timms AE, Appel H, Gimmen M, Le N, Jiang L, Kim DW, Chen S, Espinoza D, Telger AE, Weir K, Clark BS, Cherry TJ, Qian J, Blackshaw S | 2021 | Gene regulatory networks controlling temporal patterning, neurogenesis, and cell fate specification in the mammalian retina | https://www.ncbi.nlm.nih.gov/geo/query/acc.cgi?acc=GSE181251 | NCBI Gene Expression Omnibus, GSE181251 |

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
