## [Editor Report · eLife Assessment]

This manuscript presents a **valuable** methodological approach to investigating context-dependent activity of cis-regulatory activity within defined genomic loci. The authors combine a locus-specific massively parallel reporter assay, enabling unbiased and high-coverage profiling of enhancer activity across large genomic regions, with a degenerate reporter assay to identify nucleotides critical for enhancer function. The data supporting the conclusions are **solid**, highlighted by successful identification and characterization of both previously known and new regulatory elements across multiple developmental stages, cell types, and species. While the approach has inherent limitations in sensitivity, and indirect assignment of regulatory elements to target genes, it provides a flexible platform for nominating candidate cis-regulatory elements across defined loci.

---

## [Referee Report · Reviewer #1 (Public review)]

MPRAs are a high-throughput and powerful tool for assaying the regulatory potential of genomic sequences. However, linking MPRA-nominated regulatory sequences to their endogenous target genes, and identifying the more specific functional regions within these sequences can be challenging. MPRAs that tile a genomic region, and saturation mutagenesis-based MRPAs can help to address these challenges. In this work, Tulloch et al. describe a streamlined MPRA system for the identification and investigation of the regulatory elements surrounding a gene of interest with high resolution. The use of BACs covering a locus of interest to generate MPRA libraries allows for an unbiased, and high-coverage assessment of a particular region. Follow up degenerate MPRAs, where each nucleotide in the nominated sequences are systematically mutated, then can point to key motifs driving their regulatory activity. The authors present this MPRA platform as straightforward, easily customizable, and less time- and resource-intensive than traditional MPRA designs. They demonstrate the utility of their design in the context of the developing mouse retina, where they first use the LS-MPRA to identify active regulatory elements for select retinal genes, followed by d-MPRA which allowed them to dissect the functional regions within those elements and nominate important regulatory motifs. These assays were able to recapitulate some previously known cis-regulatory modules (CRMs), as well as identify some new potential regulatory regions. Follow up experiments assessing co-localization of the gene of interest with the CRM-linked GFP reporter in the target cells, and CUT&RUN assays to confirm transcription factor binding to nominated motifs provided support linking these CRMs to the genes of interest. Overall, this method appears flexible and could be an easy to implement tool for other investigators aiming to study their locus of interest with high resolution.

Strengths:

(1) The method of fragmenting BACs allows for high, overlapping coverage of the region of interest.

(2) The d-MPRA method was an efficient way to identify key functional transcription factor motifs, and nominate specific transcription factor-driven regulatory pathways that could be studied further.

(3) Additional assays like co-expression analyses using the endogenous gene promoter, and use of the Notch inhibitor in the case of Olig2, helped correlate the activity of the CRMs to the expression of the gene of interest, and distinguish false positives from the initial MPRA.

(4) The use of these assays across different time points, tissues, and even species demonstrated that they can be used across many contexts to identify both common and divergent regulatory mechanisms for the same gene.

Weaknesses:

(1) The LS-MPRA assay most strongly identified promoters, which are not usually novel regulatory elements you would try to discover, and the signal to noise ratio for more TSS-distal, non-promoter regulatory elements was usually high, making it difficult to discriminate lower activity CRMs, like enhancers, from the background. For example, NR2 and NR3 in Figure 3 have very minimal activity peaks (NR3 seems non-existent). The ex vivo data in Figure 2 is similarly noisy. Is there a particular metric or calculation that was or could be used to quantitatively or statistically call a peak above the background? The authors mention in the discussion some adjustments that could reduce the noise, such as increased sequencing depth, which I think is needed to make these initial LS-MPRA results and the benchmarking of this assay more convincing and impactful.

---

## [Referee Report · Reviewer #2 (Public review)]

Summary:

In this study, Tulloch et al. developed two modified massively parallel reporter assays (MPRAs) and applied them to identify cis-regulatory modules (CRMs) - genomic regions that activate gene expression - controlling retinal gene expression. These CRMs usually function at specific developmental stages and in distinct cell types to orchestrate retinal development. Studying them provides insights into how retinal progenitor cells give rise to various retinal cell types.

The first assay, named locus-specific MPRA (LS-MPRA), tests all genomic regions within 150-300 kb of the gene of interest, rather than relying on previously predicted candidate regulatory elements. This approach reduces potential bias introduced during candidate selection, lowers the cost of synthesizing a library of candidate sequences, and simplifies library preparation. The LS-MPRA libraries were electroporated into mouse retinas in vivo or ex vivo. To benchmark the method, the authors first applied LS-MPRA near stably expressed retinal genes (e.g., Rho, Cabp5, Grm6, and Vsx2), and successfully identified both known and novel CRMs. They then used LS-MPRA to identify CRMs in embryonic mouse retinas, near Olig2 and Ngn2, genes expressed in subsets of retinal progenitor cells. Similar experiments were conducted in chick retinas and postnatal mouse retinas, revealing some CRMs with conserved activity across species and developmental stages.

Although the study identified CRMs with robust reporter activity in Olig2+ or Ngn2+ cells, the data do not provide sufficient evidence to support the claims that these CRMs regulate Olig2 or Ngn2, rather than other nearby genes, in a cell type-specific manner. For example, the authors propose that three regions (NR1/2/3) regulate Olig2 specifically in retinal progenitor cells based on: (1) the three regions are close to Olig2, (2) increased Olig2 expression and NR1/2/3 activity upon Notch inhibition, and (3) reporter activity observed in Olig2+ cells (though also present in many Olig2- cells). While these are promising findings, they do not directly support the claims.

The second assay, called degenerate MPRA (d-MPRA), introduces random point mutations into CRMs via error-prone PCR to assess the impact of sequence variations on regulatory activity. This approach was used on NR1/2/3 to identify mutations that alter CRM activity, potentially by influencing transcription factor binding. The authors inferred candidate transcription factors, such as Mybl1 and Otx2, through motif analysis, co-expression with Olig2 (based on single-cell RNA-seq), and CUR&RUN profiling. While some transcription factors identified in this way overlapped with the d-MPRA results, others did not. This raises questions about how well d-MPRA complements other methods for identifying TF binding sites.

Strengths:

The study introduces two technically robust MPRA protocols that offer advantages over standard methods, such as avoiding reliance on predefined candidate regions, reducing cost and labor, and minimizing selection bias.

The identified regulatory elements and transcription factors contribute to our understanding of gene regulation in retinal development and may have translational potential for cell type-specific gene delivery into developing retinas.

Weakness:

Like other MPRA-based approaches, LS-MPRA mainly tests whether a sequence can drive expression of a reporter gene in given cell type(s). However, this type of assay generally does not show which endogenous gene the sequence regulates. In this study, the evidence supporting gene-specific CRMs is largely correlative. The evidence for cell-type-specific CRMs is also not fully supported (e.g., reporter expression is observed in the intended cell type as well as additional cell types). If further validation in the native genomic context (e.g., CRISPRi of the candidate element followed by RNA-seq across relevant cell types) is out of scope, the manuscript should avoid wording that implies definitive target gene assignment or cell-type specificity.

---

## [Referee Report · Reviewer #3 (Public review)]

Summary:

Use of reporter assays to understand the regulatory mechanisms controlling gene expression moves beyond simple correlations of cis-regulatory sequence accessibility, evolutionary sequence conservation, and epigenetic status with gene expression, instead quantifying regulatory sequence activity for individual elements. Tulloch et al., provide systematic characterization of two new reporter assay techniques (LS-MPRA and d-MPRA) to comprehensively identify cis-regulatory sequences contained within genomic loci of interest during retinal development. The authors then apply LS-MPRA and d-MPRA to identify putative cis-regulatory sequences controlling Olig2 and Ngn2 expression, including potential regulatory motifs that known retinal transcription factors may bind. Transcription factor binding to regulatory sequences is then assessed via CUT&RUN. The broader utility of the techniques are then highlighted by performing the assays across development, across species, and across tissues.

Strengths:

The authors validate the reporter assays on retinal loci for which the regulatory sequences are known (Rho, Vsx2, Grm6, Cabp5) mostly confirming known regulatory sequence activity but highlighting either limitations of the current technology or discrepancies of previous reporter assays and known biology. The techniques are then applied to loci of interest (Olig2 and Ngn2) to better understand the regulatory sequences driving expression of these transcription factors across retinal development within subsets of retinal progenitor cells, identifying novel regulatory sequences through comprehensive profiling of the region.

LS-MPRA provides broad coverage of loci of interest

d-MPRA identifies sequence features that are important for cis-regulatory sequence activity.

The authors take into account transcript and protein stability when determining the correlation of putative enhancer sequence activity with target gene expression.

Overall, the manuscript highlights the utility of the techniques to identify novel cis-regulatory sequence contributions to gene expression, including systematic characterizations of sequence motifs conferring activating or repressive functions.

Limitations:

Barcoding strategies have the potential to induce high collision rates (see Table S3) that may lead to misinterpretation of the data and/or high false positive/negative rates.

There are limited robust methods to distinguish differentially active versus inactive CRMs in the LS-MPRA.

---

## [Author Response]

The following is the authors’ response to the original reviews.

**Public Reviews:**

**Reviewer #1 (Public review):**
MPRAs are a high-throughput and powerful tool for assaying the regulatory potential of genomic sequences. However, linking MPRA-nominated regulatory sequences to their endogenous target genes and identifying the more specific functional regions within these sequences can be challenging. MPRAs that tile a genomic region, and saturation mutagenesis-based MPRAs, can help to address these challenges. In this work, Tulloch et al. describe a streamlined MPRA system for the identification and investigation of the regulatory elements surrounding a gene of interest with high resolution. The use of BACs covering a locus of interest to generate MPRA libraries allows for an unbiased and high-coverage assessment of a particular region. Follow-up degenerate MPRAs, where each nucleotide in the nominated sequences is systematically mutated, can then point to key motifs driving their regulatory activity. The authors present this MPRA platform as straightforward, easily customizable, and less time- and resource-intensive than traditional MPRA designs. They demonstrate the utility of their design in the context of the developing mouse retina, where they first use the LS-MPRA to identify active regulatory elements for select retinal genes, followed by d-MPRA, which allowed them to dissect the functional regions within those elements and nominate important regulatory motifs. These assays were able to recapitulate some previously known cis-regulatory modules (CRMs), as well as identify some new potential regulatory regions. Follow-up experiments assessing co-localization of the gene of interest with the CRM-linked GFP reporter in the target cells, and CUT&RUN assays to confirm transcription factor binding to nominated motifs, provided support linking these CRMs to the genes of interest. Overall, this method appears flexible and could be an easy-to-implement tool for other investigators aiming to study their locus of interest with high resolution.Strengths:(1) The method of fragmenting BACs allows for high, overlapping coverage of the region of interest.(2) The d-MPRA method was an efficient way to identify key functional transcription factor motifs and nominate specific transcription factor-driven regulatory pathways that could be studied further.(3) Additional assays like co-expression analyses using the endogenous gene promoter, and use of the Notch inhibitor in the case of Olig2, helped correlate the activity of the CRMs to the expression of the gene of interest, and distinguish false positives from the initial MPRA.(4) The use of these assays across different time points, tissues, and even species demonstrated that they can be used across many contexts to identify both common and divergent regulatory mechanisms for the same gene.Weaknesses:The LS-MPRA assay most strongly identified promoters, which are not usually novel regulatory elements you would try to discover, and the signal-to-noise ratio for more TSS-distal, non-promoter regulatory elements was usually high, making it difficult to discriminate lower activity CRMs, like enhancers, from the background. For example, NR2 and NR3 in Figure 3 have very minimal activity peaks (NR3 seems non-existent). The ex vivo data in Figure 2 are similarly noisy. Is there a particular metric or calculation that was or could be used to quantitatively or statistically call a peak above the background? The authors mention in the discussion some adjustments that could reduce the noise, such as increased sequencing depth, which I think is needed to make these initial LS-MPRA results and the benchmarking of this assay more convincing and impactful.

Much of the statistical and quantitative data asked for by the Reviewers have been provided in the Revision. However, it is important to note that the types of statistics using peak callers asked for regarding candidate choice will be of limited value. If one is testing a library in a single cell type in vitro, and/or running genome-wide assays, these statistics could aid in the choice of candidates. However, here we are electroporating a complex and dynamic set of cells, with each cell type constituting what can be very different frequencies (e.g. Olig2-expressing cells are <2.4% of cells). This fact alone will give different apparent signal to noise values. In addition, at least for Olig2 and Ngn2, their expression is very transient, suggesting dynamic regulation by what is likely multiple positive and negative CRMs. An additional confound is that the level of expression of each gene that one might test is variable. All of these variables render a statistical prediction of candidates to be less valuable than one might hope, and might lead one to miss those CRMs of interest, particularly those in a small subset of cells. Instead, we suggest that one use one’s own level of interest and knowledge in choosing CRM candidates. We provide several examples of experimental, rather than purely statistical, approaches that might help in one’s choice of candidates. We used a functional read-out of CRM activity (Notch perturbation), carried out in the context of the entire LS-MPRA library, as one method. Co-expression in single cells of candidate regulators identified by the d-MPRA is another. One can of course use chromatin structure and sequence conservation, as used in many studies of regulatory regions, as other ways to narrow down candidates. The d-MPRA predictions also can be viewed in light of previous genetic studies, i.e. mutations in TFs that effect the cell type of interest or the regulation of the gene of interest, as we were able to do here for CRMs predicted to be regulated by Otx2.

**Reviewer #2 (Public review):**
Summary:In this study, Tulloch et al. developed two modified massively parallel reporter assays (MPRAs) and applied them to identify cis-regulatory modules (CRMs) - genomic regions that activate gene expression, controlling retinal gene expression. These CRMs usually function at specific developmental stages and in distinct cell types to orchestrate retinal development. Studying them provides insights into how retinal progenitor cells give rise to various retinal cell types.The first assay, named locus-specific MPRA (LS-MPRA), tests all genomic regions within 150-300 kb of the gene of interest, rather than relying on previously predicted candidate regulatory elements. This approach reduces potential bias introduced during candidate selection, lowers the cost of synthesizing a library of candidate sequences, and simplifies library preparation. The LS-MPRA libraries were electroporated into mouse retinas in vivo or ex vivo. To benchmark the method, the authors first applied LS-MPRA near stably expressed retinal genes (e.g., Rho, Cabp5, Grm6, and Vsx2), and successfully identified both known and novel CRMs. They then used LS-MPRA to identify CRMs in embryonic mouse retinas, near Olig2 and Ngn2, genes expressed in subsets of retinal progenitor cells. Similar experiments were conducted in chick retinas and postnatal mouse retinas, revealing some CRMs with conserved activity across species and developmental stages.Although the study identified CRMs with robust reporter activity in Olig2+ or Ngn2+ cells, the data do not provide sufficient evidence to support the claims that these CRMs regulate Olig2 or Ngn2, rather than other nearby genes, in a cell-type-specific manner. For example, the authors propose that three regions (NR1/2/3) regulate Olig2 specifically in retinal progenitor cells based on: (1) the three regions are close to Olig2, (2) increased Olig2 expression and NR1/2/3 activity upon Notch inhibition, and (3) reporter activity observed in Olig2+ cells (though also present in many Olig2- cells). While these are promising findings, they do not directly support the claims.The second assay, called degenerate MPRA (d-MPRA), introduces random point mutations into CRMs via error-prone PCR to assess the impact of sequence variations on regulatory activity. This approach was used on NR1/2/3 to identify mutations that alter CRM activity, potentially by influencing transcription factor binding. The authors inferred candidate transcription factors, such as Mybl1 and Otx2, through motif analysis, co-expression with Olig2 (based on single-cell RNA-seq), and CUR&RUN profiling. While some transcription factors identified in this way overlapped with the d-MPRA results, others did not. This raises questions about how well d-MPRA complements other methods for identifying transcriptional regulators.Strengths:(1) The study introduces two technically robust MPRA protocols that offer advantages over standard methods, such as avoiding reliance on predefined candidate regions, reducing cost and labor, and minimizing selection bias.(2) The identified regulatory elements and transcription factors contribute to our understanding of gene regulation in retinal development and may have translational potential for cell-type-specific gene delivery into developing retinas.Weaknesses:(1) The claims for gene-specific and cell type-specific CRMs would benefit from further validation using complementary approaches, such as CRISPR interference or Prime editing.

The methods that we developed were meant to provide candidates for regulatory elements for a gene of interest. These candidates could be used to further understand the regulation of a gene, a complex and difficult task, especially for dynamically regulated genes in the context of development. These candidates could also, or instead, be used to drive gene expression specifically in a target cell of interest for applications such as gene therapy or perturbations that need this type of specificity. In the first case, to use the candidates to understand the regulation of a gene, one would need to validate the candidates using the types of methods typically employed for this purpose, most rigorously in the in vivo genomic context. We did not pursue this level of validation as it would encompass a great deal of work outside the scope of the current study. However, by initially testing loci which have been studied by several groups (as cited in the manuscript, Rho, Grm6, Vsx2, and Cabp5), we were able to show that LS-MPRA can identify known CRMs. In the cases of Rho and Vsx2, previous data have shown the CRMs to be relevant in the genomic context in vivo. In addition, two Vsx2 CRM’s identified by LS-MPRA are located at -37 Kb and -17Kb, and the Grm6 CRM identified by LS-MPRA is at -8Kb. These are the same CRM locations identified previously using classical methods. These data show that the method is capable of identifying distal elements. When one has only one or a few loci of interest, i.e. one does not need to use genome-wide approaches, LS-MPRA is accurate enough to be worth the relatively small effort to identify potential CRMs, even those at some distance from the TSS. However, it is apparent that our methods are not perfect and that the LS-MPRA does not pick up all CRMs. We do not know of a method that has been shown to do so.

**Reviewer #3 (Public review):**
Summary:Use of reporter assays to understand the regulatory mechanisms controlling gene expression moves beyond simple correlations of cis-regulatory sequence accessibility, evolutionary sequence conservation, and epigenetic status with gene expression, instead quantifying regulatory sequence activity for individual elements. Tulloch et al., provide a systematic characterization of two new reporter assay techniques (LS-MPRA and d-MPRA) to comprehensively identify cis-regulatory sequences contained within genomic loci of interest during retinal development. The authors then apply LS-MPRA and d-MPRA to identify putative cis-regulatory sequences controlling Olig2 and Ngn2 expression, including potential regulatory motifs that known retinal transcription factors may bind. Transcription factor binding to regulatory sequences is then assessed via CUT&RUN. The broader utility of the techniques is then highlighted by performing the assays across development, across species, and across tissues.Strengths:(1) The authors validate the reporter assays on retinal loci for which the regulatory sequences are known (Rho, Vsx2, Grm6, Cabp5) mostly confirming known regulatory sequence activity but highlighting either limitations of the current technology or discrepancies of previous reporter assays and known biology. The techniques are then applied to loci of interest (Olig2 and Ngn2) to better understand the regulatory sequences driving expression of these transcription factors across retinal development within subsets of retinal progenitor cells, identifying novel regulatory sequences through comprehensive profiling of the region.(2) LS-MPRA provides broad coverage of loci of interest.(3) d-MPRA identifies sequence features that are important for cis-regulatory sequence activity.(4) The authors take into account transcript and protein stability when determining the correlation of putative enhancer sequence activity with target gene expression.Weaknesses:(1) In its current form, the many important controls that are standard for other MPRA experiments are not shown or not performed, limiting the interpretations of the utility of the techniques. This includes limited controls for basal-promoter activity, limited information about sequence saturation and reproducibility of individual fragments across different barcode sequences, limitations in cloning and assay delivery, and sequencing requirements. Additional quantitative metrics, including locus coverage and number of barcodes/fragments, would be beneficial throughout the manuscript.

We thank the reviewer for these comments and have provided detailed responses to the additional analyses in the subsequent Recommendations section.

(2) There are no statistical metrics for calling a region/sequence 'active'. This is especially important given that NR3 for Olig2 seems to have a small 'peak' and has non-significant activity in Figure 4.

See comments about peak calling in our response to Reviewer #1.

(3) The authors present correlational data for identified cis-regulatory sequences with target gene expression. Additionally, the significance of transcription factor binding to the putative regulatory sequences is not currently tested, only correlated based on previous single-cell RNA-sequencing data. While putative regulatory sequences with potential mechanisms of regulation are identified/proposed, the lack of validation (and discrepancies with previous literature) makes it hard to decipher the utility of the techniques.

See comments about further validation in our response to Reviewer #2.

(4) While the interpretations that Olig2 mRNA/protein expression is dynamically regulated improved the proportions of cells that co-expressed CRM-regulated GFP and Olig2, alternate explanations (some noted) are just as likely. First, the electroporation isn't specific to Olig2+ progenitors. Also, the tested, short CRM fragments may have activating signals outside of Olig2 neurogenic cells because chromatin conformation, histone modifications, and DNA methylation are not present on plasmids to precisely control plasmid activity. Alternatively, repressive elements that control Olig2 expression are not contained in the reporter vectors.

The electroporation of Olig2 minus and plus cells is an excellent way to determine if a CRM is active in all cells, or only a specific subset, and we therefore consider this the best way to answer the question of specificity. We agree that we were unable to show that all CRM active cells were indeed Olig2-expressing cells. As noted by the Reviewer, we went to some lengths to quantify RNA and protein co-expression, including of endogenous Olig2 protein and RNA. Even with the endogenous RNA and protein, there was a mismatch wherein one infrequently saw the two together in the same cell, which could be predicted from the short half-lives of these molecules. Regarding chromatin, etc., we are intrigued by the proper regulation that we have observed for CRMs that we have previously discovered by plasmid electroporation (e.g. Kim et al. 2008, Matsuda and Cepko, 2004, Wang et al. 2014, Emerson et al. 2013). It is indeed interesting that plasmids can recapitulate proper regulation, without the proper genomic context or chromatin modifications. We have expanded our discussion of these points in the Discussion.

(5) It is unclear as to why the d-MPRA uses a different barcoding strategy, placing a second copy of the cis-regulatory sequence in the 3' UTR. As acknowledged by the author, this will change the transcript stability by changing the 3' UTR sequence. Because of this, comparisons of sequence activity between the LS-MPRA and d-MPRA should not be performed as the experiments are not equivalent.

We had provided a rationale for the different strategies of barcoding in the original submission, and believe it is at the discretion of the experimenter to utilize either strategy for their specific purposes. We agree that comparing activity between different techniques would not be appropriate. The analysis of mutated CRMs using d-MPRA does not utilize data from the LS-MPRA, but is an analysis of relative activity among all mutated d-MPRA constructs.

(6) Furthermore, details of the mutational burden in d-MPRA experiments are not provided, limiting the interpretations of these results.

We have provided detailed responses to the additional analyses in the subsequent Recommendations section and included details of the mutational burden in Supplemental Document A.

(7) Many figures are IGV screenshots that suffer from low resolution. Many figures could be consolidated.

We have increased the resolution of all IGV genome tracks, but believe the content within all figures remains appropriate.

**Recommendations for the authors:**

**Reviewer #1 (Recommendations for the authors):**
Suggestions for improving the clarity of the results in the figures:(1) The pie charts used the show the percentage of overlapping cells in the colocalization analyses were not especially intuitive to read, and although the percentages and any statistical significance were often written in the text, it would've been helpful to have them written in the figures. I would suggest displaying the results in stacked bar plots, possibly like the one shown in Figure 6A, to demonstrate the data more clearly.

We thank the reviewer for the suggestions. Though adding the percentages directly to the pie charts would make the relevant panels too confusing to interpret, we added supplemental tables (Tables S5-S9) with the percentages displayed in all pie charts for readers interested in the precise quantifications.

(2) The scRNA-seq UMAPs showing co-expression of Olig2 with the TFS of interest - it is very hard to see the cells that co-express. I would recommend either having a window zoomed in on the Olig2-expressing cell population to be able to see the co-expression more clearly visually, and/or including a graph demonstrating the percentages of co-expressing cells. These numbers were written in the text, but would be useful to see in the figure.

The resolution of the scRNA-Seq plot has been improved for the visualization of co-expressing cells, which were also brought forward in all UMAP plots to improve clarity. Because of the higher quality images, insets should no longer be necessary. We have also included percentages of co-expression in the figures (Figs. 8 and 8S) and thank the reviewer for the suggestion.

Other minor suggestions/corrections:(3) Figures 6B and 10S are missing the overlap quantification (in bar or pie charts) like in the other figures.

The quantification for the image in 6B (i.e., GFP fluorescence and GFP RNA) is displayed in 6D for the four Olig2 CRM plasmid constructs. In Fig. 10S, the experiments in early chick ventral neural tube delivered constructs to a very limited number of cells, and quantification of cells would not necessarily represent an accurate number of cells with CRM activity. We therefore decided to show only representative images of CRM activity in this population of cells rather than present a biased count or increase the number of experiments/samples to obtain a robust quantification.

(4) On the second-to-last line of page 10, in the sentence "The d-MPRA approach provided a robust, high resolution method for functionally relevant TF binding sites....", I think you're missing a word between "for" and "functionally". For example, it might be "for identifying..." or "for nominating...".

We have revised the sentence accordingly.

**Reviewer #2 (Recommendations for the authors):**
Minor suggestions:(1) Please indicate which mouse reference genome (e.g., mm10) was used in plots such as Figure 2.

We have added text to the relevant sections in the Results (the reference genome was already mentioned in Methods).

(2) In Figures 2 and 2S, the CRMs discussed in the text are not labeled or highlighted, making it unclear which regions are being referenced.

We have labeled peaks with roman numerals in both the figures, legends, and text for clarity and thank the reviewer for the suggestion.

(3) Consider listing the genomic coordinates for the CRMs mentioned in the text, as this information would be especially useful for readers interested in exploring these regions further.

This information was included in Table 2S in the original submission, with all relevant coordinates provided therein.

(4) The d-MPRA plots (e.g., Figure 7C-E) do not clearly show the effects of different nucleotide substitutions. A more informative visualization style can be found in Kircher et al (PMID: 31395865, Fig. 1D) or Deng et al (PMID: 38781390, Fig. 5F).

The precise nucleotide substitutions would be informative to visualize the effects of specific changes. However, we were more interested in how any nucleotide substitution influenced the CRM activity to hone in on relevant TFBS. We therefore believe the current visualization is the most appropriate to accomplish this. However, for some types of future applications, a more informative visualization as noted would be a valuable addition.

(5) It would be extremely helpful to the community if the LS-MPRA data were uploaded to the UCSC genome browser and made accessible via a link.

We have uploaded all LS-MPRA genome tracks to a Track Hub in the UCSC genome browser and provided the appropriate link to access the Hub (https://github.com/cattapre/ALAS00) in the methods section.

**Reviewer #3 (Recommendations for the authors):**
(1) The authors should address the following metrics to showcase the utility of the techniques:

We thank the reviewer for requesting the detailed metrics outlined below. We have addressed all inquiries and included the majority of metrics in the resubmission.

(a) Library sizeThis should be shown for each library that is generated. It is acknowledged that the complete size of the library is limited by sequencing, and the comprehensiveness of the library will change every time the library is re-prepped. However, metrics of this are not currently provided in a robust manner for each library. "Libraries of at least 7x10^6 and as many as 9x10^7 fragments are made" - vague - how was library complexity established since this seems to be an estimation, how many reads were utilized to estimate library complexity?

We created a new supplemental table (Table S3) that displays the complexity based on sequencing rather than the estimated complexity based on the serial dilutions prior to 3D culture (which was used for the estimates listed in the results). We updated the complexity range in the text as well and thank the reviewer for the suggestion.

Does library size scale proportionally to the BACs of different sizes?

The fragmentation of different BACs with differing sizes does not necessarily alter the size of the library. Library size is primarily determined by the library creation pipeline, with the size selection step of the fragmented BAC and the cloning step that inserts adapter-ligated fragments into the barcoded expression vector being the primary determinants of complexity of plasmid libraries.

(b) Sequence saturationCan the authors please provide evidence that the libraries have been sequenced to saturation or estimates of the degree of under-sequencing? How many reads does it take to discover a new barcode associated with a new regulatory sequence?

We have provided library characteristics for this in Table S3 and have also generated Sequence Saturation Curves for each association library in Supplemental Document A.

(c) Barcode saturationHow many barcodes are present for each fragment in the libraries? Are most fragments only covered by 1 barcode? The barcoding strategy doesn't prevent the same barcode from being assigned to multiple different fragments, as barcodes are random. What is the incidence of barcode collisions?

We have provided library characteristics for this in Table S3 and have also generated Barcode Saturation Curves for each association library in Supplemental Document A.

Additionally, we tested whether the omission of barcode collisions would affect the output of our LS-MPRA. We reanalyzed one barcode abundance library (one replicate following 12h Notch inhibitor) and filtered the barcodes so that only unique barcodes were analyzed. We were able to replicate all previously identified peaks. Though it is not necessary to filter out barcode collisions, there may be an improvement in signal-to-noise if the sequencing depth of libraries was sufficient (see Supplemental Document B).

(d) NormalizationAs performed, fragment activity is normalized by RNA expression compared to the presence of fragments in the library. While this is done for small libraries, for large libraries, this may not be appropriate. For large libraries, every sequence in the library will not be delivered to each cell, and many fragments contained in the library may not be electroporated at all. Ideally, the authors would have sequenced both the RNA and DNA from the electroporations to (i) identify the fragment distribution of the library that was successfully electroporated and (ii) provide an internal normalization factor across replicate samples. This is especially important if the libraries were ever re-prepped, as the jack-potting or asymmetries in fragment recovery can occur every time the library is re-derived.

We agree with the reviewer’s comments about the variability in fragments delivered experimentally, though we also believe the normalization of the libraries is still appropriate. We never needed to re-prep the libraries as there was sufficient material for many more experiments than were performed. However, should one ever need to re-prep an LS-MPRA library, all experimental sequencing should be normalized to the respective sequenced association library to account for biased distributions, as the reviewer mentions.

In the absence of these metrics (this would likely require the authors to repeat all experiments and is acknowledged to be outside the scope of revisions), the authors should provide information on the percentage of the library that is profiled in the RNA for each library.

We have provided RNA profiles of all abundance libraries in Table S4. The overall fraction of fragments represented in the RNA pools was lower than that observed in other published MPRAs. This difference is expected given that most MPRA studies preselect fragments based on chromatin accessibility, transcription factor binding, sequence conservation, or bioinformatically predicted CRMs, thereby enriching for regulatory elements with high activity potential. Our locus-specific MPRA libraries, by contrast, include all fragments across the targeted genomic region, many of which are likely to be inactive in the tested context. Consequently, only a smaller proportion of fragments show measurable RNA expression.

(e) Fragment sizesPlease provide a density plot or something similar showcasing the size distribution of the libraries generated. Is there any correlation between sequence activity and the size of fragments?

We have generated size distribution plots and correlations between fragment size and activity of all libraries and have included them in Supplemental Document A.

(2) Questions about the statistical validity of results:(a) What threshold is utilized for calling a sequence as active? This is important as NR3 does not seem to be an element that has significant activity.

See comments about peak calling in prior responses.

(b) A Fisher's exact test using cells from single-cell RNA-sequencing as replicate samples is inappropriate as the cells are (i) not from replicate experiments and (ii) potentially in different cell states. The proportions of cells across replicate scRNA-seq datasets would be more appropriate.

We thank the reviewer for raising this important point. While we agree that individual cells do not substitute for biological replicates, we believe Fisher’s exact test remains appropriate for testing whether gene expression is associated with Olig2 expression within a single scRNA-seq dataset. The test assesses co-occurrence at the level of individual cells, which is valid under the assumption that each cell represents an independent sampling of transcriptional states, even when it is possible that cells are in different states. We use this method as an exploratory tool to identify candidate genes associated with Olig2 expression in this dataset, and in the future, this could also be further validated by comparing the proportions of cells across replicate datasets, as the reviewer mentions.

(3) Discussion of the reporter/Olig2/Ngn2 RNA/protein disconnect needs to be expanded. Some simpler explanations for the presence of GFP in Olig2- and Ngn2- cells, as well as the presence of Olig2 or Ngn2 in GFP- cells, is that (i) these putative CRMs are being introduced to cells in plasmids, taking them out of their native genomic context where they may be inaccessible or repressed and allowing them to drive reporter expression even if their candidate target gene is not endogenously expressed, (ii) these putative CRMs may regulate genes besides just Olig2 or Ngn2, and (iii) Olig2 and Ngn2 are regulated by far more regulatory elements than the 3 or 4 being tested in each reporter assay, so their expression likely does not rely solely on the activity of the few putative CRMs tested.

We have added these points in an expanded discussion in the text.

(4) Problems with figures: Low resolution of many IGV genome tracks, pink 'co-expression' dots are completely indiscernible. Numbers should be listed with the pie charts. BFP expression should be shown since this is being quantified, especially since electroporation efficiency can change across age and/or tissue samples.

We have reconfigured the IGV tracks so that they are higher resolution and have included supplemental tables for the numbers pertaining to the pie charts. For electroporation controls (BFP and RFP), BFP expression is shown in Figs 5S, 6, and 10S and the RFP electroporation control is shown in Fig. 11. Though BFP is sometimes used as a qualifier in the denominator of some of the quantification, displaying its expression, particularly in combination with three other signals that are already included in most images, provides limited utility.

(5) More information is required to understand the utility of the d-MPRA. Detailed quantification of the number of mutations/fragments needs to be ascertained. When multiple mutations are present, how are the authors controlling for which mutation is affecting activity? What is the coverage of the loci of interest for mutational burden (ie, is every base pair mutated in at least one fragment?). For mutations that increase the activity of the element, are there specific sequence features that increase activity (new motifs generated)?

The d-MPRA platform is a high-throughput assay that seeks to identity putative sub-regions within CRMs nominated by the LS-MPRA, or any other assay. It relies on deep mutational coverage to determine positive and negative regulatory sub-regions of the CRMs. While many reads have multiple mutations, they are broadly co-occurring across the entire fragment (see Supplemental Document A) so as not to create a false linkage between the sites. Every individual site is mutated many times with roughly even coverage across each fragment (see Supplemental Document A), thus allowing us to assess the requirement of each base in contributing to a putative CRM’s activity. Comparing d-MPRA plots using bulk fragments or fragments with singleton mutations (Supplemental Document A) yielded almost identical plots for two libraries, and a similar analysis of the third library. Any differences between analysis of fragments with one or more mutations is likely a result of either sequencing depth or the requirement of multiple bases for binding or CRM activation. Follow-up experiments investigating intra-CRM interactions would elucidate such variability. Whether new motifs are generated for any specific substitution is an interesting question, which could be followed up for a CRM of interest. The d-MPRA data that we provide would provide the starting point for such follow-up experiments.

(6) Transcription factors as regulators of CRM-activity.It is appreciated that the authors validated the binding of transcription factors to NR2. However, this correlative analysis should be further tested in follow-up experiments to highlight novel biology using systems already in place. Potential experiments that could be performed include the following (reagents in hand, or performed in a manner similar to experiments performed by the lab in previous publications):(a) over-expression of TF using LS-MPRA library.(b) over-expression of TF using d-MPRA library, showing that mutations in the putative TF binding site disrupt activity compared to non-mutated sequences.(c) performing TF over-expression using target CRMs, including sequences where the TF binding site is mutated (similar to a small MPRA).(d) the quantification of target gene expression when (i) TF is over-expressed, (ii) CRM is activated using CRISPRa, or (iii) CRM is inhibited using CRISPRi.

These are all valid follow-up experiments. Please see prior responses we have provided regarding further validation.

Minor points(1) Please acknowledge that some distal regulatory sequences may be contained outside of the BAC regions. Also, the authors should emphasize the point that the assay is NOT cell-type-specific or specific to regulatory sequences for the gene of interest, but ALL regulatory sequences contained within the locus. The discussion of this with respect to Ift122 and Rpl32 is somewhat confusing.

We have added a sentence in the Discussion addressing possible CRMs outside the BAC coverage. We believe it is implicitly understood that the assay only screens regulatory activity in the BAC, and believe we have addressed this in the manuscript.

If one wishes to use a candidate CRM to drive gene expression in a targeted cell type, one needs to establish specificity. In particular, specificity needs to be established in the context of the vector that is being used. Non-integrated vs integrated vectors, different types of viral vectors with their own confounding regulatory sequences, different types of plasmids and methods of delivery, and copy number can all affect specificity. We provided a double in situ hybridization method for the examination of specificity for some of the novel candidate CRMs. It was quite difficult in the case of Olig2 and Ngn2 as their RNAs and proteins are unstable. We would need to provide further evidence should we wish to use these candidate CRMs for directing expression specifically in Olig2- or Ngn2-expressing cells. We suggest that an investigator can choose the vector and method for establishing specificity depending upon the goals of the application.

(2) I am curious as to why low-resolution, pseudo-bulked single-nucleus ATAC was utilized instead of more comprehensive retina ATAC samples at similar time-points (for example, as available in Al Diri et al., 2017 (E14, E17, P0, P3, P7, P10)) samples are all available.

The use of pseudo-bulked single-nucleus ATAC-seq data provided a convenient and consistent comparison to our LS-MPRA results. We agree that incorporating higher-resolution datasets such as those from Al Diri et al. would be valuable for future analyses aimed at linking CRM activity with broader chromatin accessibility dynamics.